# Ocean-driven millennial-scale variability of the Eurasian Ice Sheet during the Last Glacial Period simulated with a hybrid ice-sheet-shelf model

Jorge Alvarez-Solas[1,2], Rubén Banderas[1,2], Alexander Robinson[1,2], and Marisa Montoya[1,2]

[1]Dpto. Física de la Tierra y Astrofísica; Facultad de Ciencias Físicas; Universidad Complutense de Madrid (UCM)
[2]Instituto de Geociencias (UCM-CSIC), Madrid, Spain
*Correspondence to:* Jorge Alvarez-Solas (jorge.alvarez.solas@fis.ucm.es)

**Abstract.**

The last glacial period (LGP; ca.110-10 ka BP) was marked by the existence of two types of abrupt climatic changes, Dansgaard-Oeschger (DO) and Heinrich (H) events. Although the mechanisms behind these are not fully understood, it is generally accepted that the presence of ice sheets played an important role in their occurrence. While an important effort has been made to investigate the dynamics and evolution of the Laurentide Ice Sheet (LIS) during this period, the Eurasian Ice Sheet (EIS) has not received much attention, in particular from a modeling perspective. However, meltwater discharge from this and other ice sheets surrounding the Nordic Seas is often implied as a potential cause of ocean instabilities that lead to glacial abrupt climate changes. Thus, a better understanding of the evolution of the EIS during the LGP is important to understand its role in glacial abrupt climate changes. Here we investigate the response of the EIS to millennial-scale climate variability during the LGP. We use a hybrid, three-dimensional, thermomechanical ice-sheet model that includes ice shelves and ice streams. The model is forced offline through a novel perturbative approach that, as opposed to conventional methods, clearly differentiates between the spatial patterns of millennial-scale and orbital-scale climate variability. Thus, it provides a more realistic treatment of the forcing at millennial timescales. The effect of both atmospheric and oceanic variations are included. Our results show that the EIS responds with enhanced ice discharge in phase with interstadial warming in the North Atlantic when forced with surface ocean temperatures. Conversely, when subsurface ocean temperatures are used, enhanced ice discharge occurs both during stadials and at the beginning of the interstadials. Separating the atmospheric and oceanic effects demonstrates the major role of the ocean in controlling the dynamics of the EIS on millennial time scales. While the atmospheric forcing alone is only able to produce modest iceberg discharges, warming of the ocean leads to higher rates of iceberg discharges as a result of relatively strong basal melting at the margins of the ice sheet. Our results clearly show the capability of the EIS to react to glacial abrupt climate changes, and highlight the need for stronger constraints on the ice sheet's glacial dynamics and climate-ocean interactions.

# 1 Introduction

The last glacial period (LGP; ca. 110-10 ka before present, BP) was marked by the existence of two types of abrupt climatic changes: Dansgaard-Oeschger (DO) and Heinrich (H) events (e.g. Alley et al., 1999). DO-events are identified in Greenland ice-core records as regional abrupt warmings by up to 16°C (Huber et al., 2006; Kindler et al., 2014) from cold (stadial) to relatively warm (interstadial) conditions within decades (Dansgaard et al., 1993) followed by a gradual cooling interval lasting from centuries to millennia and an ultimate phase of rapid cooling back to stadial conditions (Steffensen et al., 2008). Superimposed on the millennial-scale variability associated with DO-events, an additional lower-frequency climatic cycle is identified. So-called Bond cycles are flanked by prolonged stadials ending with prominent DO-events within about 7-10 ka (Bond et al., 1993). Preceding these, and concomitant with the culmination of the prolonged stadials, H-events are registered in North Atlantic marine sediments as layers of remarkably high concentrations of ice-rafted debris (IRD) (Heinrich, 1988) as a result of massive iceberg discharges from the Laurentide ice-sheet (LIS) (Hemming, 2004).

While significant effort has been invested in understanding the role of the LIS in glacial abrupt climate changes, the dynamics of the Eurasian Ice Sheet (EIS) during the LGP has received comparatively less attention from a modeling perspective. However, improving our understanding of the evolution of the EIS and its response to past climate changes is important for a number of reasons. First, constraining freshwater inputs into the North Atlantic Ocean is crucial for a better understanding of the driving mechanisms of glacial abrupt climate changes (Rasmussen and Thomsen, 2013), since meltwater discharge from the ice sheets surrounding the Nordic Seas is often implied as a cause of ocean instabilities. Precursor events could possibly have originated from the European and Icelandic ice sheets (Grousset et al., 2000; Scourse et al., 2000). Meltwater peaks in the Norwegian Sea as well as in the southern border of the EIS during Marine Isotopic Stage 3 (MIS 3; ca. 60-25 ka BP) have been associated with H events and millennial-scale climate variability (Lekens et al., 2006; Toucanne et al., 2015). From a broader perspective, the EIS, consisting of the Fennoscandian, the British Isles and the Barents-Kara ice sheets (FIS, BIIS and BKIS, respectively) contained a large marine-based sector at its maximum extension (Hughes et al., 2016) that was exposed to oceanic variations. The BKIS, in particular, was predominantly marine-based for much of the LGP. For this reason, and because it had a similar size as the West Antarctic ice sheet (WAIS) during the Last Glacial Maximum (LGM) (Anderson et al., 2002; Bentley et al., 2014; Denton and Hughes, 2002; Evans et al., 2006; Hillenbrand et al., 2012; Svendsen et al., 2004; Whitehouse et al., 2012), it is sometimes considered as a geological analog of the current WAIS (e.g. Gudlaugsson et al., 2013). However, while the WAIS endured the deglaciation, the BKIS completely disappeared (Andreassen and Winsborrow, 2009). Mechanisms contributing to the deglaciation of the BKIS include ice stream surging (Andreassen and Winsborrow, 2009); subglacial meltwater (Esteves et al., 2017) and subsurface melting through ocean warming (Ivanovic et al., 2018; Rasmussen and Thomsen, 2004). An improved understanding of these would provide important insights into the future evolution of the WAIS (Gudlaugsson et al., 2013, 2017).

Reconstructing the evolution of glacial ice sheets prior to the LGM has been difficult, in part because, in reaching their maximum extent, ice sheets eroded and removed nearly all older deposits. This has hampered, in particular, the reconstruction of the EIS response to past glacial abrupt climate changes. Nevertheless, the available paleodata indicate that during MIS 3

the EIS was highly dynamic, with its advance and retreat closely linked to stadials and interstadials (Toucanne et al., 2015). In this line, records from Norway (Mangerud et al., 2003, 2010; Olsen et al., 2002), Finland (Helmens and Engels, 2010) and Sweden (Wohlfarth, 2010) indicate rapid and rythmic ice-sheet variations in Scandinavia, with advances and retreats during stadials and interstadials, respectively. Recent records also indicate enhanced meltwater discharges during interstadials

from the Svalbard-Barents Sea ice sheet and probably also from the Scandinavian ice sheet (Rasmussen and Thomsen, 2013). The resolution and quality of geophysical data across marine sectors has improved considerably in the past decade (Hughes et al. (2016) and references therein). These data confirm substantial variations of the EIS extent, with the largest uncertainties in marine sectors of the ice sheets; as a consequence trying to estimate its limits prior to 32 ka BP was not attempted by Hughes et al. (2016). Strong variations in the deposition of IRD suggest high co-variability of BIIS-sourced calving events

with changes in ocean sea surface temperature (Hall et al., 2011; Scourse et al., 2009) and variations in EIS ice streams (Becker et al., 2017). North Atlantic marine sediment records register widespread variations of IRD input throughout the LGP indicating variations of iceberg rafting from virtually all surrounding ice sheets. Sources and timing differ among different sites. A dominant periodicity equal to that of DO-events was identified in sediment records from the Irminger Sea, with the largest IRD peaks at the end of stadials originating in the Iceland and Greenland ice sheets (von Kreveld et al., 2000). Strong

millennial-scale iceberg rafting variability of the BIIS has been documented in sediment records from the North Sea (Hall et al., 2011; Peck et al., 2007; Scourse et al., 2009), but enhanced IRD seems to occur both during interstadials and stadials. For the FIS, IRD records in the Norwegian Sea show the characteristic DO periodicity, with IRD discharge occurring just before stadial-to-interstadial transitions (Lekens et al., 2006). More recently, however, an increase in IRDs from Fennoscandia during interstadials has been reported (Dokken et al., 2013; Becker et al., 2017). Correlating IRD occurrence with temperature

changes registered in Greenland remains difficult, however, because it requires an extremely well dated chronology to assess the phasing between ocean sediments and ice cores.

Progress has been achieved also in the past decade using ice-sheet models. Siegert and Dowdeswell (2004) used inverse modelling to simulate the EIS evolution during the second part of the LGP, matching the geological evidence presented by optimizing the fit with data. Forsström and Greve (2004) used subsequent versions of a three-dimensional, polythermal ice-

sheet model to simulate the EIS evolution throughout the LGP. Important variations in the EIS ice volume in response to temperature and precipitation variations were simulated. Clason et al. (2014) additionally included a parameterisation of surface meltwater enhanced sliding. In both cases too much ice was simulated in the northeastern EIS. Gudlaugsson et al. (2017) used the same model but introducing a simple representation of the subglacial hydrological system, focusing on its role in the temporal evolution of the EIS. Recently, an ice-sheet model constrained by data has been used to simulate the EIS evolution

throughout part of the LGP, from 37-19 ka BP (Patton et al., 2016). This study was subsequently extended throughout the last deglaciation until 8 ka BP (Patton et al., 2017). The model targets the most probable EIS distribution at different time slices and reproduces substantial ice-volume variations. However, all of these models suffer from limitations, such as the use of the shallow-ice approximation (SIA) and its associated lack of an explicit treatment of the oceanic forcing. Marshall and Koutnik (2006) investigated the production of icebergs from all the North American ice sheets with a parameterized

calving model. They found different behaviors on millennial time-scales depending on the local glaciological and climatic

characteristic, with increased iceberg production both during stadials (e.g. from Iceland) and during interstadials (e.g. from Barents Sea). Nonetheless, sub-marine melting at the grounding line has not been explicitly considered until now and its impacts on millennial-scale variability have not been investigated up to now from a modelling perspective. Notable exceptions are the recent studies by Petrini et al. (2018) and Åkesson et al. (2018). The latter used a high-resolution ice-sheet model with an accurate representation of the grounding-line dynamics to study the deglaciation of the marine-based southwestern section of the Scandinavian Ice sheet; however the model domain was limited to a very small region within southwest Norway.

Here, we investigate the response of the EIS to millennial-scale climate variability during MIS 3 using a three-dimensional ice-sheet model. To this end, a novel offline approach is used that provides a better representation of millennial-scale climate variability (Banderas et al., 2018). In addition, for the first time, both the atmospheric and oceanic effects of millennial scale climate variability associated with glacial abrupt climate changes are considered. This facilitates the quantification of the relative contribution of surface (ablation) and dynamic processes related to ice-ocean interactions.

The paper is organized as follows: in Section 2 the ice-sheet model, the forcing method and the experimental setup are described. In Section 3 the response of the EIS to the imposed forcing is shown, the focus being the evolution of its ice volume, its impact on sea level and the mechanisms behind meltwater and ice discharge. In section 4 the implications of our study for glacial and future climate changes are discussed. Finally, the main conclusions are summarised in Section 5.

## 2 Model and experimental setup

### 2.1 Model

The model used in this study is the ice-sheet model GRISLI-UCM, an extension of the original model GRISLI developed by Ritz et al. (2001). GRISLI-UCM is a hybrid three-dimensional thermomechanical ice-sheet model. Inland ice flows through deformation under the Shallow Ice Approximation (SIA, Hutter, 1983). The underlying assumption is that for grounded ice the flow is dominated by bed-parallel vertical shear (i.e., shear or deformational flow).

A nonlinear viscous flow law (Glen's flow law) is used with an exponent $n = 3$. Viscosity depends on temperature through an Arrhenius law. A traditional enhancement factor, $E_f$, that decreases viscosity and accelerates inland flow is used in most ice-sheet models as a tuning parameter, in order to improve the agreement between modelled and measured ice thicknesses; here $E_f = 3$. Further details can be found in Ritz et al. (2001). Thermomechanical coupling is extended to the ice shelves and ice streams. Ice viscosity, dependent on the temperature field, is integrated over the thickness, as in Peyaud et al. (2007). Ice shelves and ice streams are described following the Shallow Shelf Approximation (SSA, MacAyeal, 1989). In such fast flow areas bed-parallel shear is no longer dominant; instead, longitudinal and lateral stresses become important in such a way that the horizontal velocity is independent of depth (plug flow). Both approximations are valid when the spatial scales is much smaller in the vertical direction than in the horizontal one, as is the case in large-scale ice-sheet modelling. Ice streams (areas of fast flow, typically faster than $10^2 \ m \ a^{-1}$) are considered as dragging ice shelves, allowing for basal movement of the ice (Bueler and Brown, 2009). Basal stress under ice streams ($\boldsymbol{\tau}_b$) is proportional to the ice velocity $\mathbf{u}_b$ and to the effective pressure

of ice $N_{\text{eff}}$ representing the balance between ice and water pressure:

$$\boldsymbol{\tau}_b = -f\mathbf{u}_b \tag{1}$$

where

$$f = c_f N_{\text{eff}}. \tag{2}$$

Here $c_f$ is an adjustable basal friction coefficient related to the bedrock topography that accounts for the basal type of material. The effects of varying this proportionality factor on the simulated ice streams are discussed in Alvarez-Solas et al. (2011b). In this study, $c_f$ values of 20 and $2 \times 10^{-5}$ a m$^{-1}$ were used for ice streams over bedrock and sediments, respectively, accounting for the lower basal friction in the latter case. For comparison, absolute values up to $7 \times 10^{-4}$ a m$^{-1}$ were inferred by Morlighem et al. (2013) in Antarctica, with a very heterogeneous distribution, with low coefficient values in areas of fast motion dominated
by sliding. $N_{\text{eff}}$ is calculated as

$$N_{\text{eff}} = \rho g H - \rho_w g (h - b), \tag{3}$$

where $\rho$ and $\rho_w$ are the densities of ice and water, $H$ is the ice-sheet thickness, and $h$ is the hydraulic head, which corresponds to the height that would be attained by water if it were not subject to confining pressure, calculated within the basal hydrology scheme implemented by Peyaud (2006). The first term on the right hand side thus represents the pressure due to the ice load;
the second one, the subglacial water pressure. At the base of the ice shelves, friction and thus basal drag, is set to zero. The locations of the ice streams are determined by the presence of basal water within areas where the sediment layer is saturated. The criterion to activate SSA inland relies on the presence of water above 1 meter in places of soft sediments (Laske, 1997) and above 400 meters in absence of such sediments. Setting these thresholds ensures that ice streams are activated in regions that are robustly temperate. The presence of water at the base of the ice sheet implies that it is not frozen to the bedrock, i.e.,
sliding is physically possible. More water at the base facilitates sliding more by reducing the effective pressure, and sediments also facilitate sliding because they are deformable. The criteria of 1 m water thickness over sediments reduces noise in the SSA activation. The 400 m criterion over hard bedrock is a tunable parameter, also allowing for a more numerically robust calculation of velocities within the SSA. The grounding line position dynamically evolves following the flotation criterion after the mass conservation equation is solved.

Calving takes place at the ice-shelf front when two conditions are met. First, the ice-shelf thickness must fall below a threshold $H_{calv}$. This is a semiempirical parameter reflecting the fact that this is the typical thickness of ice-shelf fronts currently observed in Antarctica (Griggs and Bamber, 2011). Second, the upstream advection must fail to maintain the ice thickness above this threshold following a semi-Lagrangian approach (Peyaud et al., 2007) to account for the fact that ice-flux divergence fosters the formation of crevasses (Levermann et al., 2012). This method is standard in the GRISLI model. It was
introduced after recognising that a systematic cutoff of ice shelves below a given threshold led to a realistic simulation of the present-day ice shelves in Antarctica, as is the case in many models, but prevents any development of new ice shelves (Peyaud, 2006; Peyaud et al., 2007). When focusing on past climates, ice sheets should be able to evolve in response to climate

changes, and in particular to allow the advance of ice shelves in cold climates. To this end, before calving ice in a certain point, we test whether advection allows for the growth of ice at the front, and therefore the ice-shelf advance. $H_{calv}$ was set to 150 m, in the standard setup. To assess the sensitivity of the ice dynamics to the value of the thickness threshold $H_{calv}$ below which the ice is calved we have performed a new ensemble exploring a wide value range of this parameter's values, from 10 to 800 meters (Fig. S4). Values of this threshold above 400 m produce a drastic disintegration of the Barents-Kara complex due to its relative shallow bed. The overall effect of this sensitivity test around the preferred value is to modulate the amplitude of the response to the oceanic perturbations. GRISLI-UCM thus explicitly calculates grounding line migration, ice-stream and ice-shelf velocities. This allows the model to properly represent both grounded and floating ice. Note that there is no ambiguity in the model between calving and basal melt, which are two distinct processes in the model. Calving is the result of the threshold criterion described above; the calving rate at a given time is thus given by the amount of ice lost to the ocean through this process by unit of time, converted to mass-water equivalent. Basal melt is dependent on the applied ocean temperature anomaly. GRISLI-UCM uses finite differences on a staggered Cartesian grid at a 40 km resolution, corresponding to $224\times208$ grid points for the Northern Hemisphere domain, including the EIS, with 21 vertical levels. By default, initial topographic conditions are provided by surface and bedrock elevations built from the ETOPO1 dataset (Amante and Eakins, 2009) and ice thickness (Bamber et al., 2001). Note there are more recent datasets for Greenland topographic features (e.g. Bamber et al., 2013; Morlighem et al., 2014, 2017). However, since Greenland is not the focus of our study this does not affect our results. The glacial isostatic adjustment (GIA) is described by the elastic lithosphere – relaxed asthenosphere method (Le Meur and Huybrechts, 1996), for which the viscous asthenosphere responds to the ice load with a characteristic relaxation time for the lithosphere of 3000 years. For the sake of simplicity, the isostatic adjustment is assumed here to be only due to local ice mass variations, as other works have done in the past (Greve and Blatter, 2009; Helsen et al., 2013; Huybrechts, 2002; Langebroek and Nisancioglu, 2016; Stone et al., 2010, e.g.). The surface mass balance (SMB) is given by the sum of accumulation and ablation, both of which are calculated from monthly surface air temperatures (SATs) and monthly total precipitation. Accumulation is calculated by assuming that the fraction of solid precipitation is proportional to the fraction of the year with mean daily temperature below 2°C. The daily temperature is computed from monthly SATs assuming that the annual temperature cycle follows a cosine function. Ablation is calculated using the positive-degree-day (PDD) method (Reeh, 1989). Its main parameters are the standard deviation of daily temperature, $\sigma$, and the conversion factors from PDDs to melt for snow and ice, $f_{PDD_{snow}}$ and $f_{PDD_{ice}}$. Here, $\sigma = 5$ K, $f_{PDD_{snow}} = 0.003$ mwe PDD$^{-1}$ and $f_{PDD_{ice}} = 0.008$ mwe PDD$^{-1}$. Refreezing is considered, with a value of $C_{si} = 60\%$ (see Section 2 in the Supplementary Information). This melting scheme is admittedly too simple for fully transient paleo simulations, as it omits the contribution of insolation-induced effects on surface melting (Robinson and Goelzer, 2014). Nevertheless, insolation changes are most relevant in long-term simulations including variations at orbital timescales, especially in past warmer periods such as the Eemian. Since this study focuses on abrupt climate changes within a fixed glacial background climate, insolation changes are not important and the PDD melt model should be sufficient to give a good approximation of surface melt in response to interstadials in a reasonable manner. GRISLI-UCM accounts for changes in elevation at each time step considering a linear atmospheric vertical profile for temperature with

different lapse rates in summer and in the annual mean (0.0065 and 0.0080 K m$^{-1}$, respectively) to account for the smaller summer atmospheric vertical stability.

Basal melting for grounded ice depends on pressure and water content at the base of the ice sheet (Ritz et al., 2001) as well as on the geothermal heat flux, which is prescribed from the reconstruction by Shapiro and Ritzwoller (2004). Basal melting for floating ice is computed using a linear temperature anomaly with respect to the freezing point. The details of the implementation of the boundary conditions (SMB and oceanic basal melting) in this particular study are given below (Section 2.2). Finally, ice flow was calculated with a 1 year timestep while thermodynamics and boundary conditions (including PDD) were updated every 5 years. The model parameters, together with the range of their values explored here are shown in table 1.

## 2.2 Offline forcing method

SMB and oceanic basal melting are obtained through a time-varying synthetic climatology built through a novel method that is found to provide a more realistic offline forcing for ice-sheet models than classical offline methods (Banderas et al., 2018). The method follows a perturbative approach in the sense that the forcing combines the present-day climatology, obtained from observational data, together with simulated anomalies. But in contrast to usual offline forcing methods, orbital and millennial scale variabilities are not lumped in a sole anomaly pattern but differentiated. The method thus combines present-day obser­vations, simulated LGM anomalies relative to present, scaled by an orbital-timescale index, and simulated stadial-interstadial anomalies, scaled by a millennial-timescale index:

$$\boldsymbol{T}^{\mathrm{atm}}(t) \;=\; \boldsymbol{T}_0^{\mathrm{atm}} + (1-\alpha^\star(t))\,\boldsymbol{\Delta T}_{\mathrm{orb}}^{\mathrm{atm}} + \beta^\star(t)\,\boldsymbol{\Delta T}_{\mathrm{mil}}^{\mathrm{atm}} \tag{4}$$

$$\boldsymbol{P}(t) \;=\; \boldsymbol{P}_0\left\{\alpha^\star(t) + (1-\alpha^\star(t))\,\delta\boldsymbol{P}_{\mathrm{orb}}\left[(1-\beta^\star(t)) + \beta^\star(t)\delta\boldsymbol{P}_{\mathrm{mil}}\right]\right\} \tag{5}$$

Here, $\boldsymbol{T}^{\mathrm{atm}}(t)$ and $\boldsymbol{P}(t)$ are the SAT and precipitation fields at time $t$. $\boldsymbol{T}_0^{\mathrm{atm}}$ and $\boldsymbol{P}_0$ are the ERA-INTERIM present-day SAT and precipitation climatologies (Dee et al., 2011). $\boldsymbol{\Delta T}_{\mathrm{orb}}^{\mathrm{atm}} = \boldsymbol{T}_{\mathrm{lgm}}^{\mathrm{atm}} - \boldsymbol{T}_{\mathrm{pd}}^{\mathrm{atm}}$ and $\delta\boldsymbol{P}_{\mathrm{orb}} = \boldsymbol{P}_{\mathrm{lgm}}/\boldsymbol{P}_{\mathrm{pd}}$ are the orbital temperature anomaly and precipitation ratio relative to the present day (not shown, see Banderas et al. (2018)), respectively, obtained from previous equilibrium simulations for the preindustrial and LGM climates performed with the CLIMBER-3$\alpha$ model (Montoya and Levermann, 2008). $\boldsymbol{\Delta T}_{\mathrm{mil}}^{\mathrm{atm}} = \boldsymbol{T}_{\mathrm{is}}^{\mathrm{atm}} - \boldsymbol{T}_{\mathrm{st}}^{\mathrm{atm}}$ and $\delta\boldsymbol{P}_{\mathrm{mil}} = \boldsymbol{P}_{\mathrm{is}}/\boldsymbol{P}_{\mathrm{st}}$ are the millennial temperature anomaly and precipita­tion ratio, respectively, for the interstadial relative to the stadial state (Section 2.3). The key differences between these climate modes as simulated by Montoya and Levermann (2008) with the CLIMBER-3$\alpha$ model are that in the stadial, North Atlantic Deep Water (NADW) formation is relatively weak and takes place south of Iceland. Accordingly the sea-ice front in the North Atlantic reaches 40°N. In the interstadial state there is a northward shift and intensification of NADW formation. Northward oceanic heat transport increases, and the North Atlantic and surrounding areas warm relative to the stadial state, in particu­lar the Nordic Seas. The simulated interstadial state is thus characterised by a more vigorous NADW formation and Atlantic meridional overturning circulation (AMOC) together with reduced sea ice in the Nordic Seas, and a temperature increase of up to 10 K in the North Atlantic relative to the stadial state, with a maximum anomaly in the Nordic Seas. Note bold symbols indicate two-dimensional spatial fields. The stadial mode in our study is represented by a climate simulation of the LGM with CLIMBER-3$\alpha$ (Montoya and Levermann, 2008). The interstadial mode is taken from a recent glacial transient simulation per-

formed with the same model under glacial climatic conditions, but with intensified NADW formation (Banderas et al., 2015). $\alpha^\star$ and $\beta^\star$ are two indices that separately modulate the contribution of the orbital and millennial anomalies. Both were built based on two recent complementary temperature reconstructions over Greenland, one from the NGRIP ice-core record for the LGP (Kindler et al., 2014), and the other one from several ice-core records for the Holocene (Vinther et al., 2009). Their combination (hereafter, the KV reconstruction) results in a continuous temperature reconstruction for Greenland for the past 120 ka (Banderas et al., 2018). $\alpha^\star$ is obtained after applying a low-pass frequency filter ($f_c = 1/18\ \mathrm{ka}^{-1}$) to the original KV reconstruction based on a spectral decomposition; $\beta^\star$ is obtained following a similar procedure but retaining the high frequency signal. Both indices are tuned in such a way that the resulting synthetic temperature time series at the NGRIP site exactly matches the KV reconstruction (this distinguishes $\alpha^\star$ and $\beta^\star$ from the raw $\alpha$ and $\beta$ indices previous to this tuning; Banderas et al. (2018)).

The net basal melting rate for floating ice, $\boldsymbol{B}$ is assumed to follow a linear relation:

$$\boldsymbol{B} = \kappa\left(\boldsymbol{T}^{\mathrm{ocn}} - \boldsymbol{T}_f\right) \tag{6}$$

where $\boldsymbol{T}^{\mathrm{ocn}}$ is the oceanic temperature close to the grounding line, $\boldsymbol{T}_f$ is the temperature at the ice base, assumed to be at the freezing point, and $\kappa$ is the heat flux exchange coefficient between ocean water and ice at the ice-ocean interface; its standard value in the present study is $\kappa = 5\ m\ a^{-1}\ K^{-1}$. Several marine-shelf basal melting parameterisations can be found in the literature as recently reviewed by Asay-Davis et al. (2017). The submarine melt rate is thought to be directly influenced by the oceanic temperature variations below the ice shelves. Accordingly, most basal melting parameterizations are built as a function of the difference between the oceanic temperature at the ice–ocean boundary layer and the temperature at the ice-shelf base, generally assumed to be at the freezing point. The dependence on this temperature difference can be linear (Beckmann and Goosse, 2003) or quadratic (Holland et al., 2008; Pollard and DeConto, 2012; DeConto and Pollard, 2016; Pattyn, 2017). The linear marine-shelf basal melting parameterization used in this study is the simplest case that allows testing of the ice-sheet sensitivity to past oceanic temperature changes. Nevertheless, it accounts separately for basal melting below the ice shelves (away from the grounding line) and at the grounding line. The basal melting rate of the ice shelves ($B_{\mathrm{sh}}$) is given by the grounding-line basal melt ($B_{\mathrm{gl}}$) scaled by a constant factor ($\gamma$)

$$B_{\mathrm{sh}} = \gamma B_{\mathrm{gl}}(t) \tag{7}$$

In this study, $\gamma$ is set to 0.1. Thus, we consider that the submarine melting rate for ice shelves is 10 times lower than that close to the grounding zone, which is in qualitative agreement with observations in some Greenland glaciers with floating tongues (Münchow et al., 2014; Wilson et al., 2017) as well as in Antarctic ice shelves (Rignot and Jacobs, 2002; Marsh et al., 2016). Note that this value is subject to uncertainty. Although we did not explore any other values different from $\gamma = 0.1$, we did consider a range of $\kappa$ values between 1-10 m a$^{-1}$ K$^{-1}$, which accounts for a wide range of oceanic sensitivities (see section 2.3).

As in Peyaud et al. (2007), in regions with ocean depths above 750 m, an artificially large melting rate (20 m a$^{-1}$ is prescribed to avoid unrealistic growth of ice shelves beyond the continental-shelf break, where they would likely be subject to high melt rates in reality because of high heat exchanges with the ocean.

Following the approach described above, $\boldsymbol{T}^{\mathrm{ocn}}(t)$ is assumed to be given by an expression analogous to Eq. 4. Thus Eq. 6 can be rewritten as:

$$\boldsymbol{B} = \boldsymbol{B}_0 + \kappa \left[ (1 - \alpha^\star(t)) \boldsymbol{\Delta T}^{\mathrm{ocn}}_{\mathrm{orb}} + \beta^\star(t) \boldsymbol{\Delta T}^{\mathrm{ocn}}_{\mathrm{mil}} \right] \tag{8}$$

where $\boldsymbol{B}_0 = \kappa(\boldsymbol{T}^{\mathrm{ocn}}_0 - \boldsymbol{T}_f)$ represents the present-day oceanic basal melting rate.

Finally, millennial-scale sea-level variations are prescribed according to the reconstruction by Grant et al. (2012, Section 2.3). The specific details of the experimental setup used are described below.

## 2.3 Experimental setup

We herein investigate the response of the EIS to millennial-scale climate variability during MIS 3. The starting point of our experiments is a control-run ice-sheet simulation with constant boundary conditions for MIS 3 that provides a representative

configuration of the EIS for that time period (Figure 1). To this end, $\alpha^\star$ was set to its value at 40 ka BP, that is, $\alpha^\star = \alpha^\star_{40K} = -0.1$, and $\beta^\star = 0$ to preclude millennial-scale variations. Note however these values are to a certain extent arbitrary; they are intended to provide a stable mean background state similar but not neccessarily identical to background MIS 3 conditions. Thus:

$$\boldsymbol{T}^{\mathrm{atm}}_{40K} = \boldsymbol{T}^{\mathrm{atm}}_0 + (1 - \alpha^\star_{40K}) \boldsymbol{\Delta T}^{\mathrm{atm}}_{\mathrm{orb}} \tag{9}$$

$$\boldsymbol{P}_{40K} = \boldsymbol{P}_0 \left[ \alpha^\star_{40K} + (1 - \alpha^\star_{40K}) \delta \boldsymbol{P}_{\mathrm{orb}} \right] \tag{10}$$

$$\boldsymbol{B}_{40K} = \boldsymbol{B}_0 + \kappa (1 - \alpha^\star_{40K}) \boldsymbol{\Delta T}^{\mathrm{ocn}}_{\mathrm{orb}} \tag{11}$$

Note that although Eq. 11 is formally correct and consistent with the scheme used, in contrast to the present-day SAT or precipitation the present-day rate of oceanic basal melting cannot be determined. Thus, in practice we replace this equation by directly tuning the value of $\boldsymbol{B}_{40K}$ to obtain a reasonable ice-sheet configuration at 40 ka BP (Figure 2) given the atmospheric

forcing fields expressed by equations 9-10. To this end, a constant basal melting rate of 0.1 m a$^{-1}$ is assumed. The ice sheet was forced with the resulting climatologies for 100 ka previous to the starting of the perturbations described below. This allows the vertical temperature profile within the ice sheet to be equilibrated with the climate. This procedure was found to facilitate the growth of European ice-sheets within the reconstructed limits for 60 ka BP and 20 ka BP (Svendsen et al., 2004; Kleman et al., 2013) (Figure 2).

Our forcing method allows to investigate the response of the EIS solely to millennial-scale climate variability at MIS 3 by keeping constant the orbital component of the forcing ($\alpha^\star = \alpha^\star_{40K}$) and letting $\beta^\star$ vary throughout the LGP (eqs. 4, 5 and 8). In order to assess the relative roles of the atmosphere and the ocean, three independent experiments have been carried out. First, an atmospheric-only forced simulation (ATM) in which the time evolution of SAT and precipitation on millennial time scales

is considered, while the oceanic forcing is kept constant to MIS 3 (i.e., 40 ka BP) background climatic conditions. Thus:

$$\boldsymbol{T}^{\mathrm{atm}}(t) = \boldsymbol{T}^{\mathrm{atm}}_{40K} + \beta^{\star}(t)\,\boldsymbol{\Delta T}^{\mathrm{atm}}_{\mathrm{mil}} \tag{12}$$

$$\boldsymbol{P}(t) = \boldsymbol{P}_{40K}\left[(1 - \beta^{\star}(t)) + \beta^{\star}(t)\,\delta\boldsymbol{P}_{\mathrm{mil}}\right] \tag{13}$$

$$\boldsymbol{B}(t) = \boldsymbol{B}_{40K} \tag{14}$$

Second, an oceanic-only forced simulation OCN in which the atmospheric forcing is kept constant while the oceanic basal melting is allowed to vary at millennial timescales around its background MIS 3 value:

$$\boldsymbol{T}^{\mathrm{atm}}(t) = \boldsymbol{T}^{\mathrm{atm}}_{40K} \tag{15}$$

$$\boldsymbol{P}(t) = \boldsymbol{P}_{40K} \tag{16}$$

$$\boldsymbol{B}(t) = \boldsymbol{B}_{40K} + \kappa\,\beta^{\star}(t)\,\boldsymbol{\Delta T}^{\mathrm{ocn}}_{\mathrm{mil}} \tag{17}$$

The magnitude and sign of oceanic temperature anomalies $\boldsymbol{\Delta T}^{\mathrm{ocn}}$ depends on the depth at which $\boldsymbol{T}^{\mathrm{ocn}}$ is considered. In our simulations, a large part of the NE sector of the EIS is marine based with shallow bedrock depths between 500 m and less than 100 m in several locations further south. It is therefore unclear whether this marine ice sheet should be more susceptible to changes in the surface or the subsurface of the ocean. To investigate the effect of this uncertainty, we decided to perform two different simulations considering different depths: one corresponding to the surface (OCNsrf) and the other one considering

deeper (subsurface) oceanic waters by averaging temperatures within the range of 400-600 m depth (OCNsub). Therefore we hereafter distinguish between $\boldsymbol{\Delta T}^{\mathrm{ocn}}_{\mathrm{mil}}$ for surface or subsurface millennial-scale temperature anomalies, respectively (Figure 3). The realism and convenience of applying one or the other is addressed in section 4.

Finally, a simulation ALL combining both the atmospheric and the oceanic forcings:

$$\boldsymbol{T}^{\mathrm{atm}}(t) = \boldsymbol{T}^{\mathrm{atm}}_{40K} + \beta^{\star}(t)\,\boldsymbol{\Delta T}^{\mathrm{atm}}_{\mathrm{mil}} \tag{18}$$

$$\boldsymbol{P}(t) = \boldsymbol{P}_{40K}\left[(1 - \beta^{\star}(t)) + \beta^{\star}(t)\,\delta\boldsymbol{P}_{\mathrm{mil}}\right] \tag{19}$$

$$\boldsymbol{B}(t) = \boldsymbol{B}_{40K} + \kappa\,\beta^{\star}(t)\,\boldsymbol{\Delta T}^{\mathrm{ocn}}_{\mathrm{mil}} \tag{20}$$

In all experiments $\beta^{\star}(t)$ dictates the millennial-scale variability of the forcings (Figure 4, a). Because our simulated stadial-to-interstadial transition results from an intensification of the AMOC, positive $\beta^{\star}$ values imply an increase in $\boldsymbol{T}^{\mathrm{atm}}$ relative to its background MIS 3 value (e.g., Eq. 18 and Figures 3 and 4). As a consequence, the atmosphere warms at interstadials relative to

stadial periods, as reflected by the $\boldsymbol{\Delta T}^{\mathrm{atm}}_{\mathrm{mil}}$ millennial-scale anomaly field (Figure 3 a, b). Note that refreezing is not allowed to occur in our model in the current setup. If $\kappa\beta^{\star}\,\boldsymbol{\Delta T}^{\mathrm{ocn}}_{\mathrm{mil}} < -\boldsymbol{B}_{40K}$ (which would imply $\boldsymbol{B}(t) < 0$) we simply impose the value $\boldsymbol{B}(t) = 0$.

An ensemble of simulations for different values of $\kappa$ have been considered to evaluate the sensitivity of the EIS to the forcing. Finally, varying sea-level forcing is considered (Figure 4b), both alone (SL run) and in combination with the previous forcings

(ALL).

## 3 Results

We analyse the response in terms of the ice volume evolution, the mass balance and the grounding-line dynamics. The different simulations analysed here are summarized in table 2.

### 3.1 Ice volume evolution

Substantial differences are found in the response of the EIS to the forcing scenarios. Under constant forcing, the CTRL run shows negligible millennial-scale sea-level equivalent (SLE) variations, although a lower frequency SLE fluctuation is found as a result of internal ice-sheet variability (Figure 4) through a thermomechanical feedback. This slow variability appears only in the southernmost parts of the Eurasian ice sheet where ablation exists. It is due to an interplay between the available basal water favoring sliding and the EIS associated thinning due to an increase in velocities. Since this phenomenon concerns

only the ablative borders of the ice sheet and its frequency corresponds to more than 20 kyr, its governing dynamics is not detailed here. When the model is forced only by changes in sea level (SL run), a small response of approximately 0.5 m SLE is observed on millennial-scales. These changes appear not be sufficient to cause a substantial migration of the grounding line, thus not affecting ice velocities (not shown). In ATM, the atmospheric forcing alone causes a sequence of enhanced ablation episodes resulting in modest ice volume variations (up to 1.5 m SLE) during the most prominent stadial-interstadial transitions;

this represents a change of approximately 7% with respect to the initial ice-sheet volume. In contrast, the oceanic forcing in OCNsrf induces pronounced changes in the dynamics of the EIS on millennial time scales (see below), with episodes of large volume reduction occurring during interstadials. The combination of sea level, atmospheric and oceanic forcings (ALLsrf) results in a very similar response of the EIS to that obtained in OCNsrf (Figure 4) as a consequence of the larger effect of the oceanic forcing in OCNsrf with respect to ATM. OCNsub shows an anti-phase relationship with respect to OCNsrf, with the

largest reductions in ice volume occurring during prolonged stadial periods and regrowth during interstadials. This behavior can be explained by the fact that ocean waters at the subsurface warm (cool) during episodes of reduced (enhanced) convection at the Nordic Seas as a result of variations in the AMOC strength (Figure 3d-e). Note that the anti-phase relationship is, however, not perfect. At the surface, the largest anomalies are found off the North Atlantic, the British Isles and the Norwegian coast, and result from the intensification of Atlantic northward heat transport associated to the enhanced AMOC during interstadials;

at the subsurface the concomitant cooling is largest in the Nordic Seas as a result of enhanced heat loss to the atmosphere associated with enhanced convection.

  Thus, the out-of-phase relationship found in the dynamic response of the EIS between these two oceanic experiments results from the opposed sign of their spatial forcing patterns (Figure 3). When considering the forcing at the subsurface of the ocean together with the atmosphere (ALLsub), slight reductions of the EIS volume (less than 1 m of s.l.e) during interstadials are

superimposed onto the previous behavior (Figure 4).

  As a consequence of the millennial-scale forcing, a trend in ice volume from its initial value of $8.3 \times 10^{15} km^3$ (about 21 m SLE) leading to a loss of 8-12 m SLE is found. This is a consequence of the fact that no refreezing is allowed and that a positive constant (and spatially uniform) basal melting of 0.1 m/a was imposed, As a consequence accumulation is not able

to compensate for ice loss through basal melt and calving after each ice-mass loss event. Note, however, that background conditions are fixed at 40 ka BP; in a more realistic setup, as time proceeds forward, orbital forcing leading to gradually colder conditions would be expected to aid in the ice regrowth, thereby helping to its growth throughout the LGP. Spatially non-uniform background melting is also conceivable. However, we have no information on what this background value would

have been. Because our focus was the response of the EIS to millennial-scale climatic variability, we opted for the simplest experimental setup possible, meaning a spatially uniform and fixed-in-time background value perturbed by a millennial-scale index.

The magnitude of these changes in terms of sea-level rise rate and discharge, specifically for the MIS 3 period, is illustrated in Figure 5. The simulations forced with the surface of the ocean (OCNsrf and ALLsrf) show the largest amplitudes, with

peaks of sea-level rise above 4 mm $a^{-1}$ during DO-events and sustained contributions well above 1 mm $a^{-1}$ during entire interstadial periods. In ATM, a decline of the EIS during stadial-to-interstadial transitions is still observed but presents a smaller amplitude of 1-2 mm $a^{-1}$. The simulations in which the ice sheet is forced with the subsurface of the ocean (OCNsub and ALLsub) present a decline of their volume during stadial periods and regrowth during interstadials as a consequence of the inverted spatial pattern of temperature anomalies with respect to the surface. In OCNsub (and ALLsub) the amplitude of

these changes is smaller than in OCNsrf (and ALLsrf), on the order of 0.5-1 mm $a^{-1}$, reaching more than 1 mm $a^{-1}$ during pronounced stadials (as ca. at 44 ka BP). The ALLsrf and ALLsub simulations show a similar or slightly larger volume loss during interstadials, as a consequence of the additional atmospheric forcing, that is superimposed onto the OCNsrf and OCNsub behaviour.

## 3.2 Mass-balance response

The response of the EIS has been analyzed in terms of its mass balance decomposition for the all-forcing runs (Figure 6). In ALLsrf the surface ocean temperature varies in phase with the atmosphere (Figure 3). Thus, during stadial-to-interstadial transitions, the high negative values of dV/dt can be explained by the conjunction of an initial sharp increase in ablation together with pronounced increases in basal melting and calving, which allow for a large grounding line retreat in the Bjørnøyrenna basin (Figure 6 b). The rate of ice loss by basal melting is similar to that resulting from the increase in ablation (as reflected in

the SMB) during the peak of a stadial-to-interstadial period. However, basal melting is much more efficient than surface mass balance in decreasing volume along the whole duration of an interstadial. This is due to the fact that ablation is restricted to the southern borders of the EIS. Thus, when the ice sheet has retreated to areas of no ablation, in spite of a slight further loss provided by the elevation feedback it rapidly equilibrates and a negative surface mass balance cannot propagate further inland. In contrast, when enhanced basal melting from higher oceanic temperatures is applied, the associated retreat can propagate

further inland occupying a large proportion of the Bjørnøyrenna basin and facilitating high rates of volume loss (although similar in amplitude with respect to SMB) during the whole interstadial period (see the animation corresponding to ALLsrf in the Supplementary Information). Note that basal melting together with calving is a very efficient method to remove ice; basal melting leads to thinning of the ice shelf which can subsequently undergo calving. During stadial periods, both the enhanced positive mass balance and the absence of basal melting (favored by the negative oceanic anomalies) favor the regrowth of the

EIS. Subsurface ocean temperatures evolve also in phase with the atmosphere in the SW part of the EIS but in anti-phase in its NE part. In other words, when forcing with the subsurface of the ocean, a slight warming (cooling) is observed around the Britain -Ireland ice sheet while cooling (warming) of the Bjørnøyrenna basin is simulated during interstadial (stadial) periods (see Figure 3). Therefore, the ALLsub simulation presents volume declines during stadial-to-interstadial transitions

due to an increase in ablation and basal melting in the SW part. The corresponding mass fluxes reach up to about 0.05 Sv; of these, approximately 0.025, 0.02 and 0.005 Sv originate in the Barents-Kara, Scandinavia and the British Islands, respectively. Subsequently, reduced basal melting in the NE part of the EIS favors regrowth of the Bjørnøyrenna basin during interstadial periods. Finally, shifting to pronounced stadial periods (as in ca. 44 ka BP) favors the penetration of warm subsurface waters that increase basal melting enough to produce an ice-sheet retreat in the NE part in spite of the enhanced positive surface

mass balance (Figures 5 and 6). When considering the atmosphere and the subsurface ocean forcing together in ALLsub, these competing processes translate into a smaller amplitude of millennial-scale EIS changes as compared to the case with surface ocean forcing (ALLsrf). Furthermore, declines of the EIS can be observed both during the beginning of interstadial periods and during pronounced stadial periods in ALLsub (Figures 5 and 6).

Focusing on the OCN and ATM simulations separately facilitates isolating the effects of the ocean on this complex pattern.

To this end, the simulated ice-sheet distribution and velocities of OCNsrf, OCNsub and ATM are shown in Figure 7 for the period around DO-event 12, at ca. 47 ka BP. As expected, OCNsrf shows a widespread retreat both in the NE and the SW of the EIS from the stadial to the interstadial period (Figure 7, d). This is accompanied by an acceleration of the Bjørnøyrenna basin due to its grounding line thinning and retreat (Figure 7, a,d). OCNsub presents a collapsed Bjørnøyrenna basin during the stadial period previous to DO-event 12 due to enhanced basal melting from warmer subsurface waters. The transition to

the interstadial period favors a slight regrowth of this NE part of the EIS due to decreased basal melting, while its SW section slightly retreats (Figure 7, b, c)

Concerning ATM, only in the southwestern (SW) part of the EIS is the atmospheric forcing capable of generating an important reduction in the EIS volume in response to the stadial-interstadial transition (Figure 7, c,f) right bottom panel). This is a result of the spatial pattern of the forcing, with the largest SAT anomalies located around the Nordic seas (Figure 3).

Therefore, the ice volume reduction of the EIS in ATM during interstadials is due to the positive SAT anomaly, which leads to enhanced ablation in the SW part of the EIS. In turn, reduced SATs during stadials allow the regrowth of the ice sheet up to the continental margin of the Nordic seas. The more active dynamic response of the EIS in the OCN simulations can be attributed to the increase in oceanic temperatures by 2-4°C (Figure 3) within the margins of the ice sheet during interstadial (in the case of OCNsrf) and stadial (OCNsub case) periods, which translates into enhanced basal melting at the margins of the EIS. The

SW sector of the EIS also responds to the warmer SSTs, actually with a larger reduction of ice volume than in ATM (Figure 7).

The spatial patterns shown in Figure 7 are representative of the ice-sheet response during all other stadial-to-interstadial transitions. In OCNsrf, the EIS reacts to every abrupt surface warming with a substantial ice-flow acceleration, especially in the Bjørnøyrenna basin (Figures 7 and 8). Ice shelves that are present during stadial periods suddenly retreat during DO-events and together with enhanced basal melting favor thinning and retreat of the grounding line that translate in large iceberg

discharges up to ca. 0.06 Sv. In OCNsub, ice velocities in the Bjørnøyrenna basin increase during stadials, when enhanced

basal melting erodes the grounding line and favors its retreat. Peaks in calving are recorded accordingly during pronounced stadial periods. These peaks are however of smaller amplitude than in OCNsrf. This can be explained by the fact that

along the coast of Eurasia, the amplitude of the simulated SST anomalies used to compute basal melting in OCNsrf is larger than the subsurface temperature anomalies in OCNsub, since the basal melt was calculated by using ocean temperature at a fixed depth, either at the surface or at the subsurface. Also, transitions to stadials are usually more gradual than transitions to interstadials, thus the incursion of warmer (subsurface) waters happens in this case in a smoother manner. High velocities reach their maxima at the end of the stadial and beginning of the interstadials. The latter are however not accompanied by an increase in calving due to the fact that ice shelves are expanding and thickening during this period thanks to reduced basal melting (Figure 9). In general, the extension of ice shelves is greatly reduced during periods of enhanced basal melting (Figures 8, 9), with no large unconfined ice shelves surviving during these episodes. Some thinner ice shelves remain, in spite of the enhanced basal melting, thanks to an increase in advection from the Bjørnøyrenna ice stream triggered by a grounding line retreat (Figure 7).

## 3.3  Grounding-line dynamics

Changes in the position of the calving front are usually accompanied by a grounding line displacement (not shown). For some minor ice-shelf breakups this close relationship can be broken, but with almost no effects upstream inland. Thus we consider that the grounding line position is the best indicator for characterizing the dynamic behavior of the marine part of the EIS. Inspection of the temporal evolution of the grounding line position in OCN simulations confirms that ice dynamics control the majority of ice-volume variations in the EIS as opposed to the SMB processes involved in ATM (Figure 11). The migration of the grounding line through time has been characterized by means of an index ($\mu$) that weighs the proportion of non-grounded points in the region of the Bjørnøyrenna basin:

$$\mu(t) = \left(1 - \frac{N_g(t)}{N}\right) \cdot 100 \tag{21}$$

where $N_g(t)$ represents the evolution of the number of points of grounded ice within a fixed area of $N$ points in the Barents Sea region defined over the black square highlighting the Bjørnøyrenna basin shown in Fig. 2. Note that other metrics are also possible; the same metric has been used in other studies in different domains such as Antarctica (Blasco et al., 2018).

An increase (decrease) in $\mu$ thus indicates a retreat (advance) of the grounding line. While in ATM $\mu$ barely changes Figure 11), OCN runs show a large dynamic behavior of the basin. In OCNsrf, $\mu$ reflects a synchronous evolution of the grounding line position and the oceanic forcing, with major retreats coinciding with interstadial states (Figure 11). Conversely, the Bjørnøyrenna basin is generally much closer to a full retreat in OCNsub during stadials due to a larger penetration of warm subsurface waters (Figure 3; OCNsub) compared to the surface waters (Figure 3; OCNsrf). However, the grounding line is able to advance and reach Svalbard during episodes of reduced basal melting at the interstadials.

The direct coupling between the oceanic forcing and the response of the Bjørnøyrenna ice stream is also evident from the relatively high negative correlation ($r \simeq -0.9$) found between $\mu$ and ice thickness in this area (Figure 11). A local thinning of the grounding line produced by a warmer ocean triggers its retreat and starts the propagation of the dynamic imbalance of

the ice stream. The propagation of a change in the surface slope happens almost instantaneously at these time scales (with a typical propagation speed of about 10 $km\ a^{-1}$). This chain of processes explains the tightened linear relationship between the Bjørnøyrenna basin thickness and the grounding line position, $\mu$. Although a grounding line retreat (advance) of the grounding line in this region produces an acceleration (deceleration) of the ice streams, its linear relationship is less obvious than regarding ice thickness (Figure 11, c)). This is explained by the fact that ice-stream velocities lag the grounding line imbalance due to the characteristic time for the kinematic wave to propagate along the ice streams of the whole basin (typically of 1 $km\ a^{-1}$).

As a consequence of the destabilization of the ice sheet, important ice-volume variations are observed in the NE part of the EIS during millennial-scale climatic transitions, which added to the minor contribution of the SW retreat (Figure 8, result in fluctuations of more than 4 m SLE in OCNsrf, up to 2.5 m in OCNsub and ca. 1 m in ATM (Figure 4).

In order to investigate the sensitivity of the results to the model parameters, eight additional OCN simulations, both for the surface and the subsurface, have been carried out with different $\kappa$ parameters between 1-10 $m\ a^{-1}\ K^{-1}$, i.e., bracketing our standard case of $\kappa = 5\ m\ a^{-1}\ K^{-1}$. This choice reflects the inferences based on measurements made on Antarctic ice shelves that a variation of 1 K in the effective oceanic temperature changes the melt rate by ca. 10 $m\ a^{-1}$ (Rignot and Jacobs, 2002; Shepherd et al., 2004). A robust response of the EIS is found, with a more reactive EIS response for increasing $\kappa$ values (see Section 1 in the Supplementary Information). The sensitivity of our results to the values of the atmospheric mass balance model has also been explored. In spite of largely exploring the values of the parameters that determine the sensitivity to surface mass balance, the EIS variability induced by the ocean is always found to be of greater amplitude than the one induced by the atmosphere provided that $\kappa > 2\ m\ a^{-1}\ K^{-1}$ (see Fig. S3 of the Supplementary Information).

## 4 Discussion

Our results suggest a highly dynamic Eurasian ice sheet at millennial time-scales largely responding to changes in the ocean temperatures. Some authors (e.g. Gudlaugsson et al., 2013) present the marine based Kara-Barents complex as an analogue for present-day West Antarctic ice sheet for which bedrock topography is a major control for stability. We have shown, in this sense, that the Bjørnøyrenna basin is highly susceptible to changes in the oceanic temperatures.

Our results indicate that the timing of the response with respect to changes registered in Greenland (i.e., their ocurrence during stadials or interstadial) depends, however, on whether the surface or the subsurface of the ocean is considered as the relevant forcing of the ice sheet. Recently, IRD peaks of Fennoscandian origin reported from a high-resolution marine sediment core from the Norwegian Sea indicate the presence of more frequent IRD deposition and thus calving during interstadials than during stadials (Dokken et al., 2013). This result has been corroborated in a compilation of new and previously published data (Becker et al., 2017) clearly showing that within MIS 3, the IRD deposition increases within interstadials. The coeval deposition of carbonate-rich, sorted fine sands and near-surface warming suggests the presence of Atlantic water along the margin, and is interpreted by the authors as the effects of winnowing due to an intensified AMOC during interstadials. This interpretation results in concordance with our results when considering the surface waters as the oceanic forcing. Thus, this

agreement would play in favor of considering that the EIS was primarily responding to changes in the surface of the ocean along the southwest EIS (Irish/Scottish margin) at least.

An out-of-phase relationship is found in the dynamic response of the EIS when forcing the ice-sheet model with the millennial-scale simulated surface and subsurface temperature anomalies. This behaviour results from the roughly opposite sign of their spatial forcing patterns in the Nordic Seas. This pattern has been found to be robust in a number of models but its details could well be model dependent, and, in particular, dependent on the precise location of the convection sites affected (e.g. Brady and Otto-Bliesner, 2011; Mignot et al., 2007; Montoya and Levermann, 2008; Shaffer et al., 2004; Flückiger et al., 2006).

Our results also provide a mechanism to explain the pervasive presence of IRD in the North Atlantic during MIS 3, both during stadials and interstadials, and originating both in the LIS and the EIS. During stadials, the simultaneous appearance of IRD across the wider North Atlantic Ocean can be explained through the build-up of subsurface heat in the high-latitude North Atlantic leading to increased iceberg calving in the presence of large, thick ice shelves, together with lower surface temperatures allowing for wider dispersal of icebergs (Barker et al., 2015). According to our results interstadials could lead to enhanced calving of the EIS through oceanic surface subglacial melting as a result of the warmer surface conditions and relatively shallow grounding lines of this ice sheet.

The identification of IRD layers with increased calving through ice-sheet instabilities must be taken with caution, since it is based on several untested assumptions (Clark and Pisias, 2000): (i) delivery of IRD to a specific site is caused solely by iceberg calving, versus transport by sea-ice; (ii) an increase in IRD represents an increase in the iceberg flux, versus a greater amount of debris incorporated at the base of the ice sheet that delivers the icebergs, or a greater distance of iceberg transport; (iii) the amount of IRD carried by all the icebergs is similar, assuming therefore a direct relationship between IRD concentration and iceberg flux. However, the former assumptions have not been confirmed and, thus, the calving-IRD relationship might not be so direct. In addition, ocean temperatures affect melting of icebergs and thus their release of IRD. Variations in ocean temperatures can alter the IRD released by an iceberg at a certain site, causing variations in IRD deposition even for a constant amount of icebergs produced at the source.

Given the conclusion that the ocean plays a major role in abrupt ice sheet changes, the model's treatment of grounding-line dynamics is a key issue. Several studies have shown that for many applications, a resolution of around 1 km is needed to accurately determine the grounding-line position. In addition, it has been shown (e.g. Gladstone et al., 2017) that the grounding-line behaviour is sensitive to the choice of friction law and the physics of submarine melting, and that these determine model-resolution requirements. In our case, the dependence of basal drag on effective pressure allows for the desirable property of basal drag going to zero at the grounding line. However, our basal-melt parameterisation does not provide a smooth transition from grounded to floating ice. Thus our results regarding the key role of the ocean on the grounding line position can be affected by the coarse model resolution. Computational constraints do not allow for the required high model resolution, especially with a 3D finite difference model on these long time scales. However, the potential inaccuracy of the grounding line position introduced by the coarse resolution, typically of 100 km (Vieli and Payne, 2005; Gladstone et al., 2017) is one order of

magnitude smaller than the grounding line migrations simulated here (more than 1000 km). This issue should be investigated in the future both at much higher resolution, as well as including different formulations of friction and submarine melting.

Furthermore, it has been suggested (Pollard et al., 2015) that the ice front can suffer dramatic calving in vertical termini glaciers due to the so-called cliff instability mechanism. This process is not parameterized in our model. We believe its inclusion would, if something, amplify the simulated response of the EIS to the ocean forcing. Nonetheless, the necessity of including this phenomenon in ice-sheet models has recently been contested (Edwards et al., 2019).

Our experimental setup is not intended to match the paleorecord, but to provide insight into the response of the EIS to millennial-scale variability. The EIS variations simulated here represent the upper-end amplitude of potential responses during the whole glacial cycle, due to its large size. Extending the study to cover the whole LGP would require the consideration of orbital variability as part of the forcing. In this case, the EIS would likely be smaller during the mildest phase of MIS 3, thus limiting its contact with the ocean and the production of iceberg discharges.

Also, our results depend somewhat on the particular SAT and oceanic temperature anomaly patterns simulated by our climate model, the magnitudes of the resulting forcing, and the initial size of the simulated EIS. Since the response to the ocean has been found to be dominant, a larger ice sheet, with more developed ice shelves and thus more exposed to the ocean would be prone to suffer stronger basal melting; destabilisation of ice shelves could therefore result in a more dynamic ice sheet with larger calving peaks. A smaller ice sheet would therefore only be affected by atmospheric forcing. The use of different atmospheric realisations is subject to the availability of climate simulations with different models for the three climate states needed: glacial (stadial), present, and interstadial. The latter is only available for a reduced number of models. This makes the assessment of this issue difficult in the present study. Assessing the sensitivity to these features should be in the scope of future work, and illustrates the need for carrying out new simulations of both the interstadial and the stadial states with more sophisticated climate models. Nonetheless, our results indicate that the ocean is the major driver of the EIS ice-volume changes during MIS-3. Note the temporal index used is the same for the atmosphere and the ocean and the amplitude is given by an OGCM simulation of two different oceanic states mimicking stadial and interstadial periods. We then translate those fields into ablation (through PDD, whose uncertainty has been extensively explored) and into basal melting (through a linear equation). It is conceivable that in certain locations our synthetic oceanic temperature forcing is larger than that deduced from reconstructions, which range from 4-10 K (Dokken et al., 2013; Martrat et al., 2004; Rasmussen et al., 2016). However, the possible uncertainty in the temperature forcing is subsumed in the $\kappa$ index which in our case varies between 1-10 m a$^{-1}$ K$^{-1}$. These values are in the range (or even below in most cases) of those suggested by data in Antarctica (Rignot and Jacobs, 2002). Note, in particular, that even from mid values of $\kappa$ of 5 m a$^{-1}$ K$^{-1}$ the response to the ocean is already of greater amplitude than that to the atmosphere, making our main conclusions robust.

For the sake of simplicity, and following up from previous work (Alvarez-Solas et al., 2011a, 2013) we herein calculated the basal melt by using ocean temperature at a fixed depth, either at the surface or at the subsurface. Using the three-dimensional temperature provided by the climate model at the local ice-shelf depth that can evolve in time as the ice-shelf thickness varies would have been more realistic and should be in the scope of future work.

Finally, our study lacks bi-directional coupling between the ice sheet, the atmosphere and the ocean. Eventually the goal is to investigate this matter with fully coupled climate-ice sheet models.

Our results have implications not only for the study of past glacial abrupt climate changes, but also currently ongoing and future climate change. In Greenland, warmer North Atlantic waters penetrating into Greenland's fjords are currently thought to contribute to the recently enhanced discharge of ice into the ocean (e.g. Straneo and Heimbach, 2013). Warmer ocean temperatures enhance submarine melting at the calving front of tidewater glaciers, contributing to accelerate them, increasing the discharge of ice mass into the ocean and potentially leading to a retreat of their grounding lines. This mechanism has been observed in several of Greenland's marine-terminating glaciers (Hill et al., 2017; Wood et al., 2018, e.g.). In Antarctica, the WAIS is losing mass at an accelerated rate as a consequence of the enhanced submarine melting of floating ice shelves and calving processes at the ice front (Paolo et al., 2015; Rignot et al., 2013). The most rapid thinning and mass loss has occurred in the ice shelves of the Amundsen and Bellingshausen seas, in regions where Antarctic Continental Shelf Bottom Water have warmed through the intrusion of Circumpolar Deep Water onto the Amundsen and Bellingshausen Seas continental shelves (Schmidtko et al., 2014). Under future climate change, many climate models project a weakening of the AMOC and a regional cooling or minimum atmospheric warming around Greenland during the 21st century that constitutes a negative feedback that could reduce melting of the GrIS in a warming climate. However, a maximum in warming has also been found to occur in the subsurface ocean layer around Greenland as a consequence of AMOC reorganisations that could induce a year-round melting of polar ice sheets (Yin et al., 2011). Projections indeed indicate enhanced subsurface warming will lead to enhanced submarine melt rates of Greenland's outlet glaciers (Nick et al., 2013; Peano et al., 2017; Calov et al., 2018), even though models do not generally account for the dynamic response of these glaciers. In Antarctica, although processes that regulate ocean heat transport to the sub-ice-shelf cavities and their sensitivity to changes in forcing need to be understood (Rintoul, 2018), climate projections indicate that changes in stratification of the water column will enhance the intrusion of CDW in Antarctic ice-shelf cavities, and thereby submarine melting (Naughten et al., 2018). This mechanism is also found in a coupled climate model including an eddying ocean component (Goddard et al., 2017). Thus, changes in ocean water temperatures appear to be key in driving ice-sheet changes both in the past and in the future.

Meltwater discharge from the EIS and other ice sheets surrounding the Nordic Seas is often implied as a cause of ocean instabilities (e.g. Broecker et al., 1985; Clark et al., 2002; Ganopolski and Rahmstorf, 2001; Rasmussen and Thomsen, 2013; Schmittner et al., 2002). The same would be the case for iceberg discharges. This issue is beyond the scope of this study; its assessment would require investigating the impact of these freshwater perturbations in deep water formation and the AMOC. Again, proper assessment requires the use of a coupled climate-ice sheet model.

## 5 Conclusions

We have investigated the response of the EIS to millennial-scale climate variability associated with DO-events through a series of simulations with a three-dimensional, hybrid ice-sheet model that represents inland ice flow under the SIA and floating ice shelves and ice streams through the SSA. The model includes as well an explicit grounding-line treatment, a simple basal

melting parameterisation that depends linearly of the ocean temperature anomalies and calving through a double criterion on ice thickness and advection at the ice front. The model makes use of an offline forcing method that separately accounts for orbital and millennial-scale climate variability during the LGP, improving the representation of the latter (Banderas et al., 2018). Atmospheric and ocean forcings associated with millennial-scale variability were considered both separately and together.

Oceanic forcing was considered both at the surface and at the subsurface. The timing of the response with respect to changes registered in Greenland depends on whether the surface or the subsurface of the ocean is considered as the relevant forcing of the ice sheet. A quasi-anti-phase relationship is found in these two cases. This behavior can be explained by the fact that ocean waters at the subsurface warm (cool) during episodes of reduced (enhanced) convection at the Nordic Seas as a result of variations in the AMOC strength.

Separating the effects of atmospheric and oceanic forcing during the glacial period has allowed us to quantify the contribution of each to EIS variability. Atmospheric forcing during stadial-interstadial transitions has a modest effect on the ice sheet, which is a consequence of the largest SMB changes being confined to SW sector of the EIS, where the forcing is strongest. In contrast, the oceanic forcing has a larger effect, through changes in the ice dynamics in the Bjørnøyrenna basin of the EIS. Ocean warming is able to induce a retreat of grounded ice in this part of the EIS through dynamic processes. As a consequence,

significant ice-volume variations result during millennial-scale climatic transitions. Added to the smaller contribution of the SW retreat, this results in sea-level changes on the order of several meters. Sensitivity experiments for different values of the oceanic heat coefficient parameter show that this is a robust response of the model.

Our results thus support the existence of a highly dynamic EIS during the LGP. They suggest an important role of oceanic melt forcing through changes in the ocean circulation in controlling the ice-stream activity. A number of studies have consid-

ered the interaction between ocean circulation changes and ice-sheet dynamics as a plausible mechanism to explain iceberg discharges from the LIS associated with H events. For example, subsurface oceanic warming during stadials in response to reduced North Atlantic deep water formation (Alvarez-Solas et al., 2010; Flückiger et al., 2006; Mignot et al., 2007; Shaffer et al., 2004) has been shown to be capable of producing large discharges from the LIS, induced by enhanced basal melting rates (Marcott et al., 2011; Alvarez-Solas et al., 2011b). The satisfactory agreement between the simulated calving and North

Atlantic marine IRD records provides strong support for this mechanism (Alvarez-Solas et al., 2013), recently proposed to be modulated by isostatic adjustment (Bassis et al., 2017). The evaluation of the impact of these NH discharges on the oceanic circulation and their effects on the triggering mechanism of DO events require the use of a coupled climate-ice-sheet model. Nonetheless, it has recently been shown that the typical oceanic cooling registered in sediment cores of the North Atlantic during stadials occurs before the arrival of the icebergs to these same cores (Barker et al., 2015). In this sense, iceberg discharges

from the Laurentide and the Eurasian ice sheets are seen as potential amplifiers but not as active elements in the triggering of millennial-scale variability. Taken together with these studies, our results support the potential of NH ice sheets to react to glacial abrupt climate changes. Additionally, our results highlight the need for stronger constraints on the local North Atlantic behavior in order to shed light on the NH ice sheet's glacial dynamics. Since the ocean plays a major role during abrupt ice sheet changes, the model's treatment of grounding line dynamics is a key issue. Finally, this represents one of the first attempts

to simulate both oceanic and atmospheric impacts on ice sheets associated to abrupt climate changes. Investigating this is-

sue further with higher resolution in and exploring the effect of the underlying uncertainties in ice-sheet and grounding line dynamics is of uttermost interest and in the scope of future work.

*Competing interests.* The authors declare no competing interests

*Acknowledgements.* This work was funded by the Spanish Ministerio de Economía y Competitividad (MINECO) through project MOCCA (Modelling Abrupt Climate Change, Grant CGL2014-59384-R). R. Banderas was funded by a PhD Thesis grant of the Universidad Complutense de Madrid. A. Robinson is funded by the Marie Curie Horizon2020 project CONCLIMA (Grant 703251). Part of the computations of this work were performed in EOLO, the HPC of Climate Change of the International Campus of Excellence of Moncloa, funded by MECD and MICINN. This is a contribution to CEI Moncloa.

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

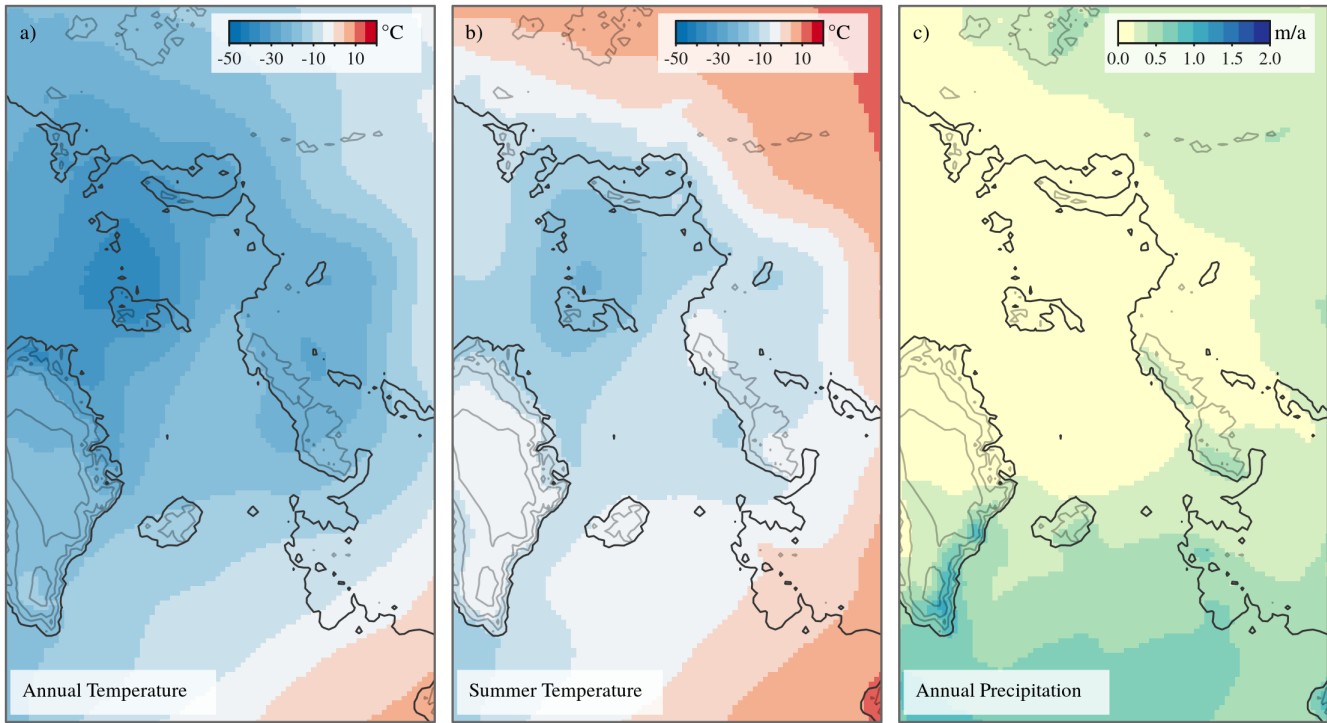

**Figure 1.** Background climatic forcing for the control run (CTRL). MIS 3 ($\sim$40 ka BP) reference annual mean SAT (**a**) and summer mean SAT (**b**) in °C and annual mean precipitation in m a$^{-1}$ (**c**). Present-day countour lines with the land boundary delineated at a depth of -80 m are added for reference.

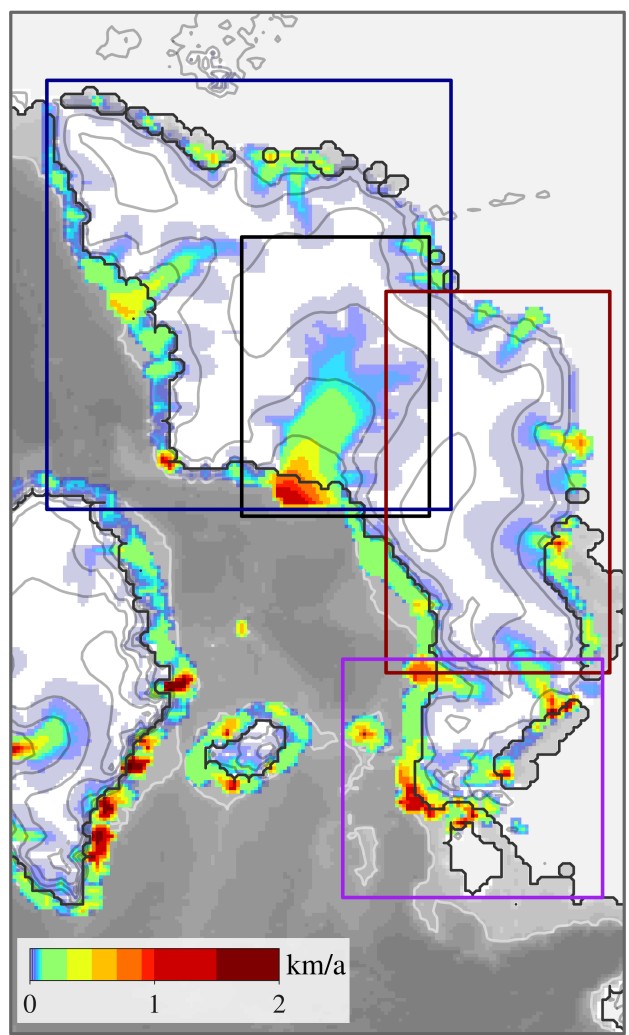

**Figure 2.** Resulting ice sheet of the MIS 3 control run (CTRL). Simulated ice thickness with contours plotted for every 500 m. The grounding line position is shown by a black line, the 500 m depth contour shown by the white line, and velocities (shaded colors, in km a$^{-1}$) after the spinup was completed. This ice sheet represents the initial state previous to the applied perturbations. Bjørnøyrenna basin, as referenced in the text, is shown by the black rectangle. The Barents-Kara, Scandinavia and Brithis Islands regions are highlighted by the blue, red and purple rectangles respectively

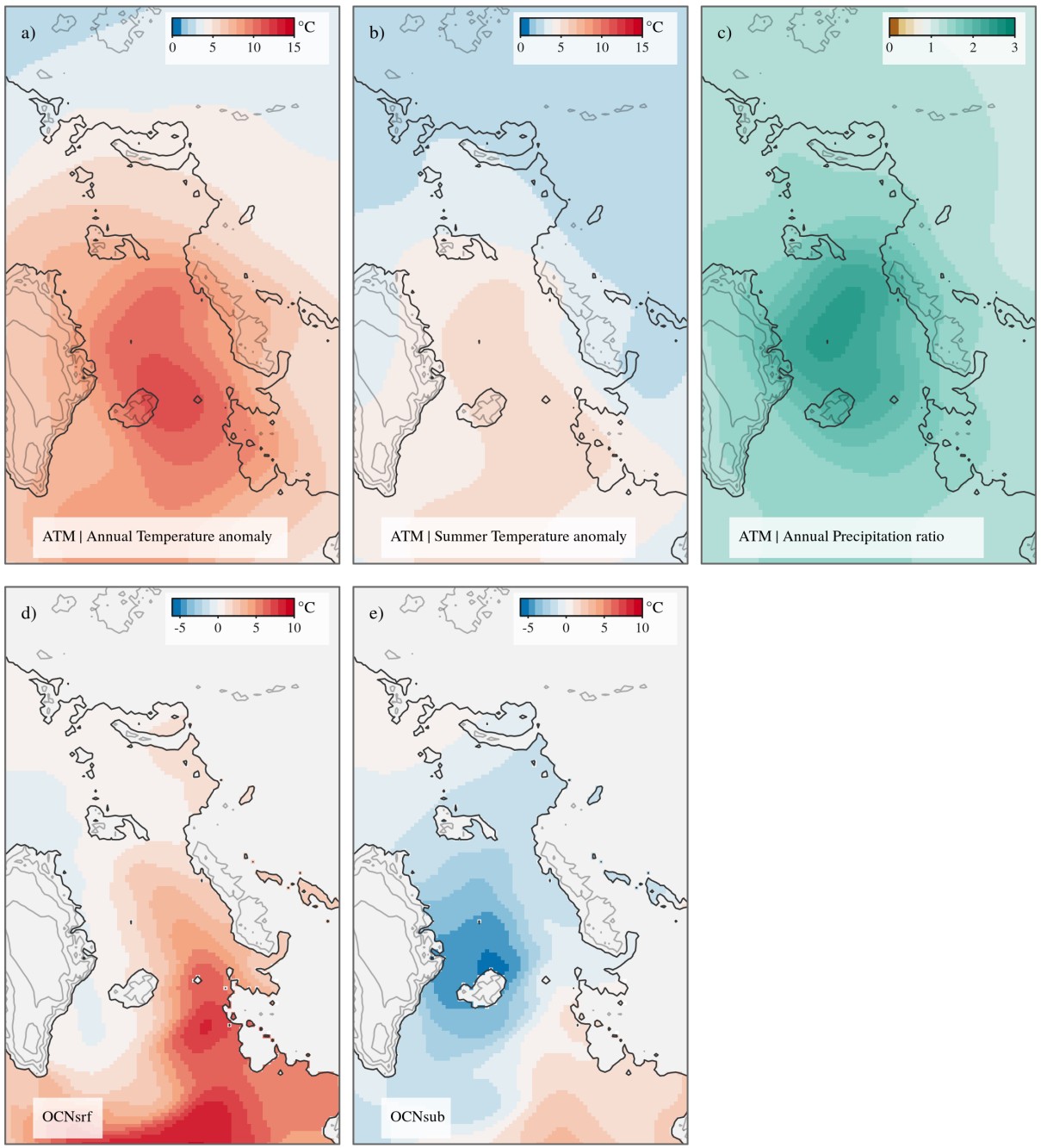

**Figure 3.** Millennial-scale components of the boundary forcing. **a)** SAT anomalies (interstadial minus stadial) in °C. **b)** Summer SAT anomalies (interstadial minus stadial) in °C. **c)** Precipitation ratio (interstadial to stadial). **d)** Anomalies of SST and **e)** subsurface ocean temperature (at 500 m depth) in °C. Present-day countour lines with the land boundary delineated at a depth of -80 m are added for reference. Note that to force the ice-sheet model these fields are scaled to reproduce the NGRIP interstadial minus stadial temperature change.

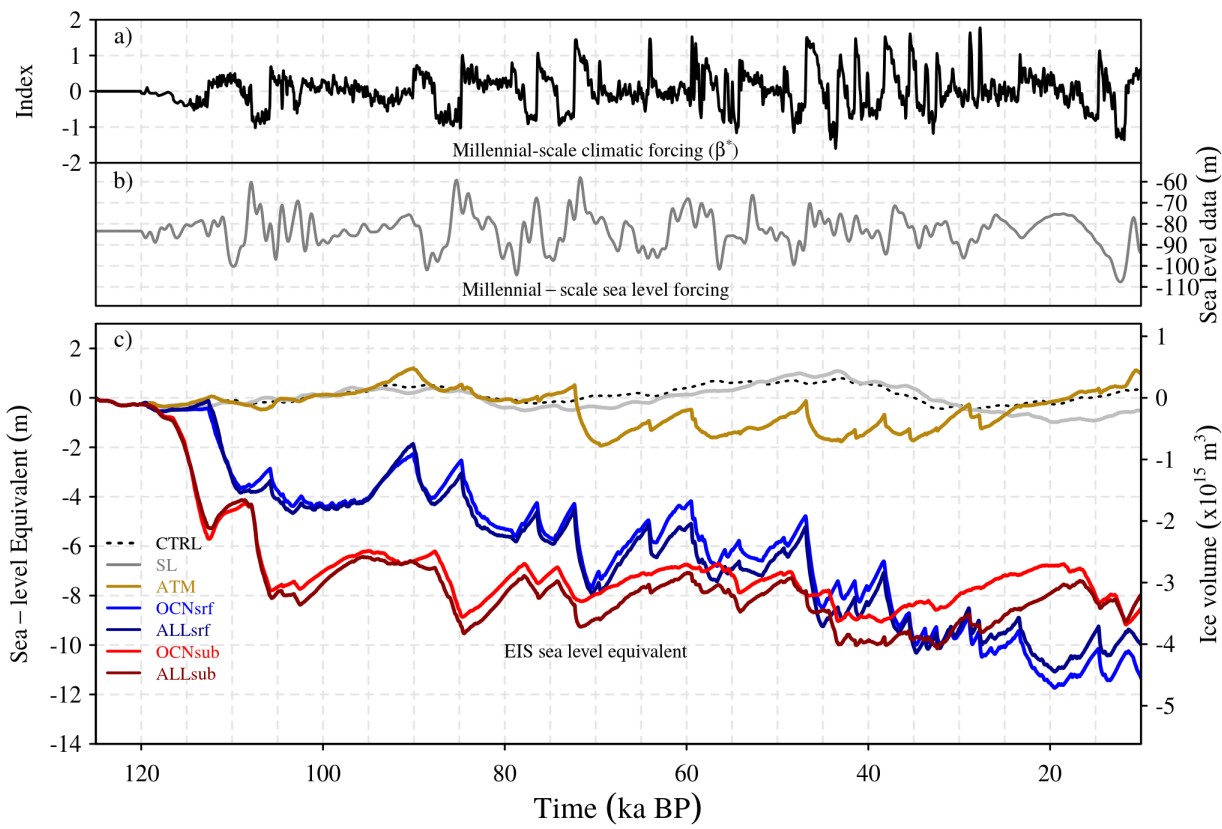

**Figure 4. a)** Temporal component of the millennial-scale climatic forcing ($\beta^\star$ index). **b)** Millennial-scale sea-level forcing (Grant et al., 2012). **c)** EIS sea-level equivalent (m) related to ice volume variations (m$^3$) with respect to initial conditions for the CTRL run (black) and for the SL (gray), ATM (gold), OCNsrf (blue), OCNsub (red), ALLsrf (dark blue), and ALLsub (dark red) forcing experiments.

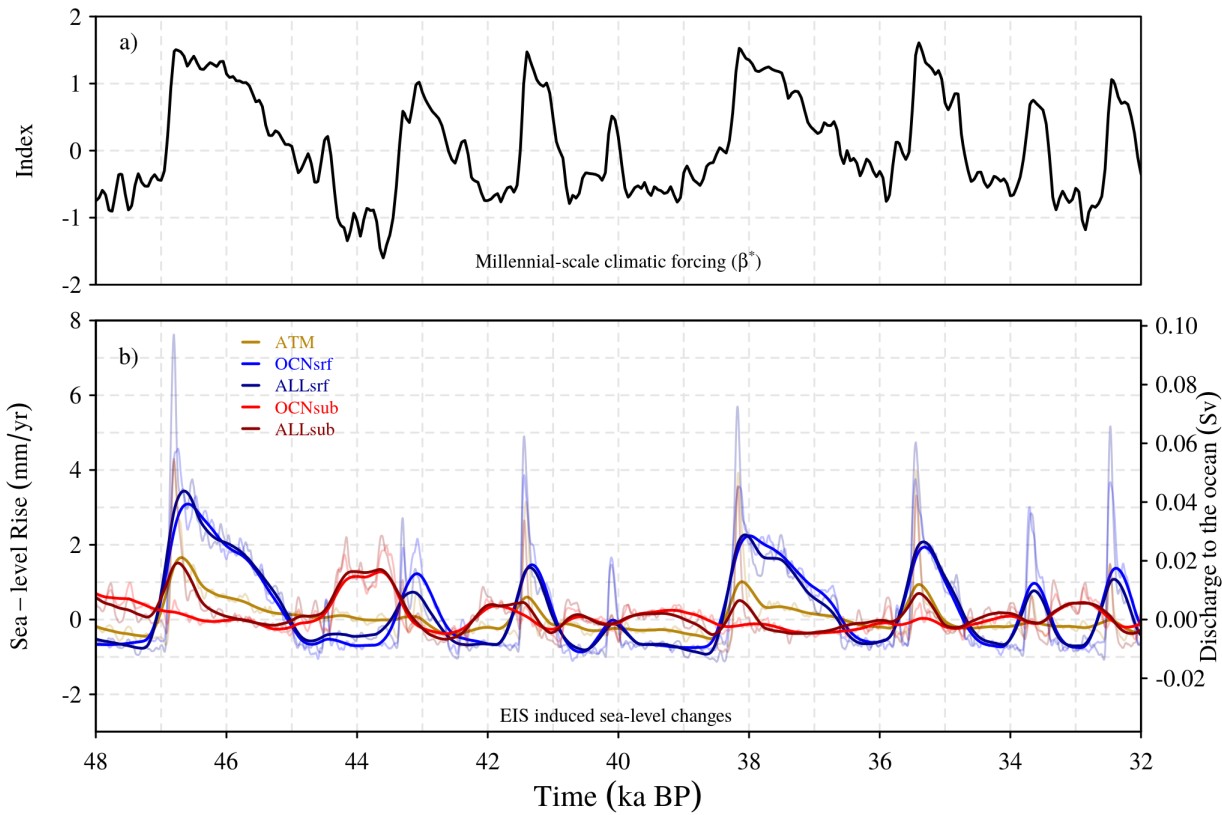

**Figure 5.** MIS3 Period. **a)** Temporal component of the millennial-scale climatic forcing ($\beta^\star$ index), and **b)** EIS changes (mm a$^{-1}$ and Sv) related to ice volume variations with respect to initial conditions for the CTRL run (black) and for the SL (gray) ATM (gold), OCNsrf (blue), OCNsub (red), ALLsrf (dark blue), and ALLsub (dark red) forcing experiments. Thick lines show the variables after applying a low pass filter of 100 years.

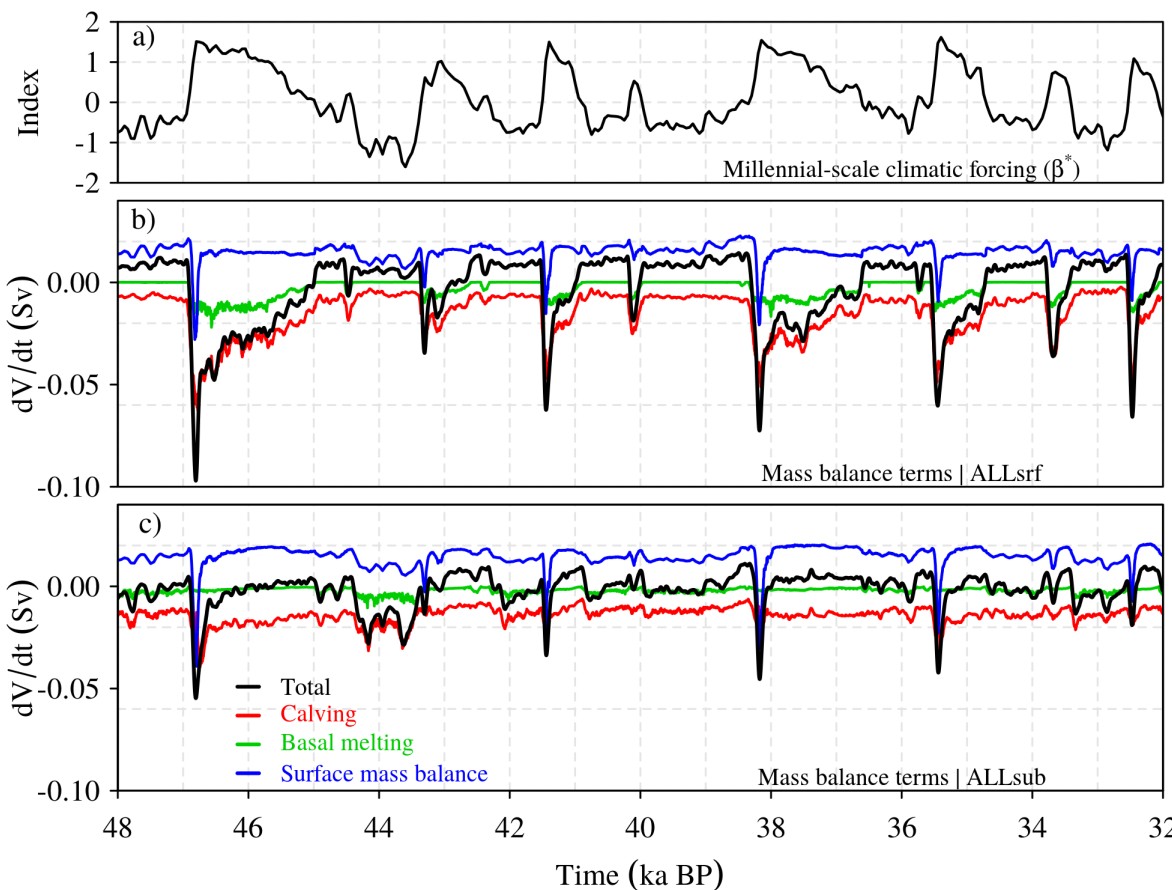

**Figure 6.** MIS3 Period. **a)** Temporal component of the millennial-scale climatic forcing ($\beta^\star$ index), and contribution of the different terms of the EIS mass balance to ice volume variations ($Sv$) in the simulations considering all forcings, with **b)** corresponding to the surface oceanic forcing (ALLsrf) and **c)** to the subsurface oceanic forcing (ALLsub). The calving and basal-melt rates are given by the amount of ice lost to the ocean through the calving and basal-melting parameterisation per unit of time, converted to water-equivalent volume.

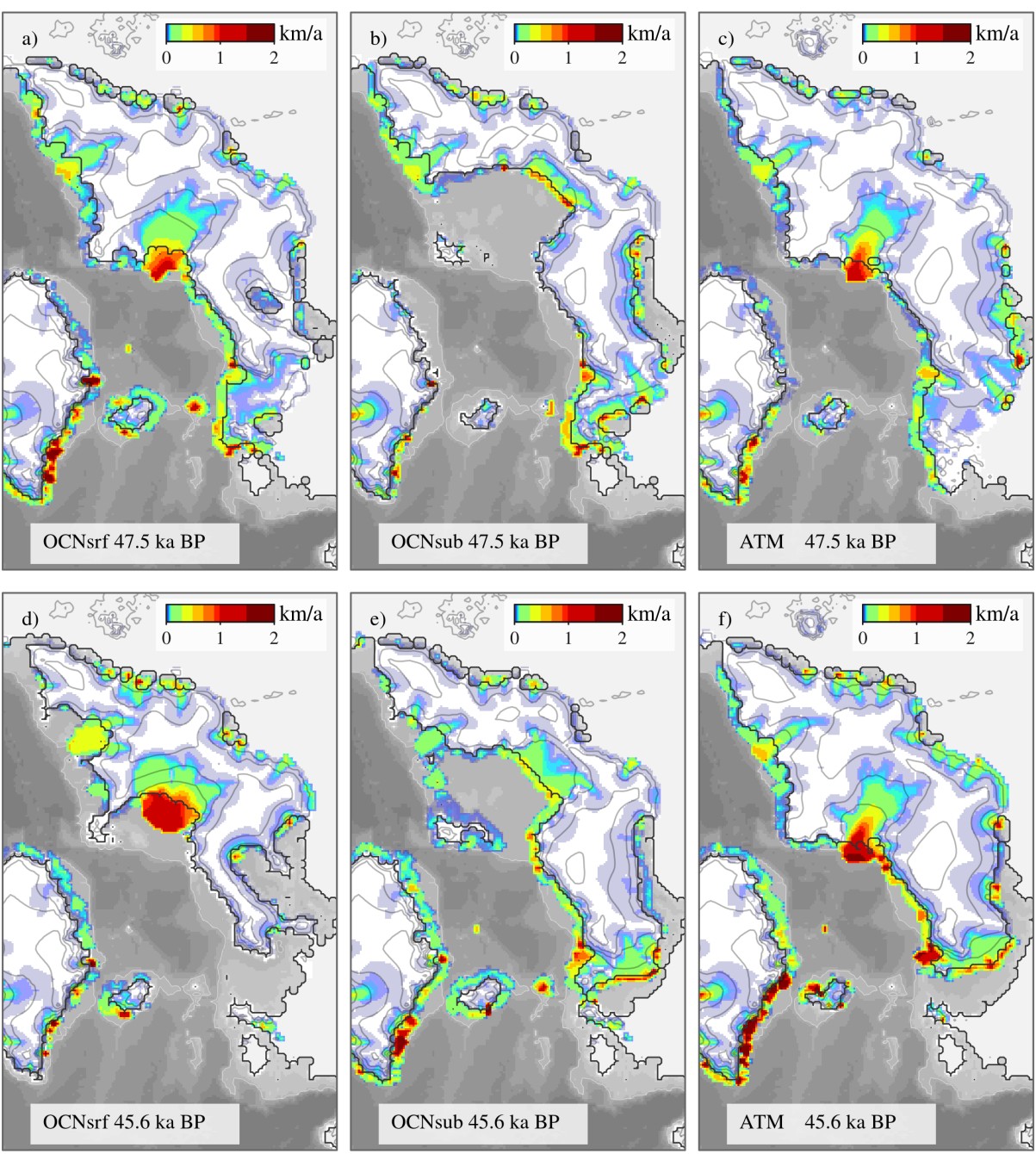

**Figure 7.** Simulated EIS at the end of a stadial period (a-c) and at the end of an interstadial period (d-f) for the experiments: OCNsrf (a, d); OCNsub (b, e) and ATM (c, f). Shaded colors show ice velocities (km a$^{-1}$). The ice thickness contours are plotted for every 500 m with the grounding line position shown by a black line. The 500 m depth contour is shown by the white line. The periods represented here corresponds to the stadial and interstadial periods prior and posterior to DO 12 (ca. 47 ka BP), respectively.

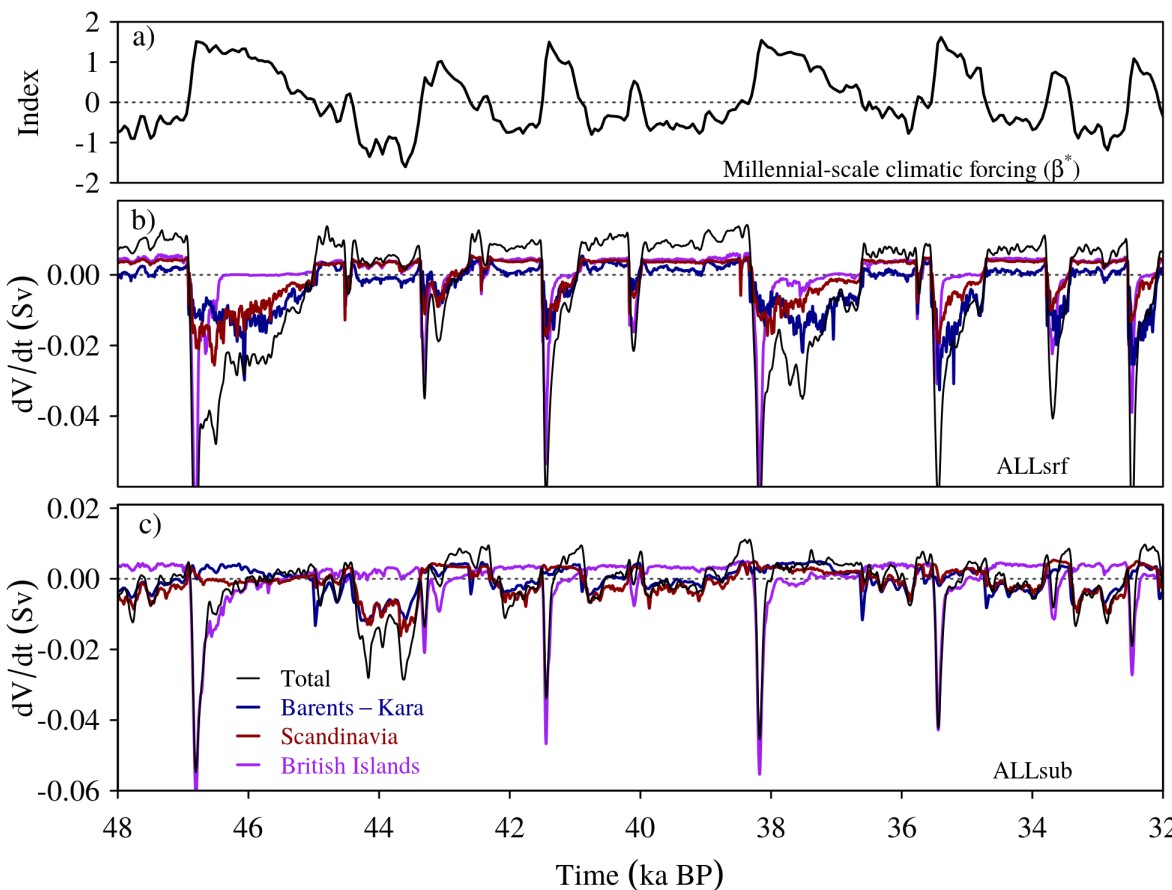

**Figure 8.** MIS3 Period. **a)** Temporal component of the millennial-scale climatic forcing ($\beta^\star$ index), and contribution of the different regions to ice volume variations ($Sv$) in the simulations considering all forcings, with **b)** corresponding to the surface oceanic forcing (ALLsrf) and **c)** to the subsurface oceanic forcing (ALLsub). The geographical domain of the different regions is highlighted in Figure 2.

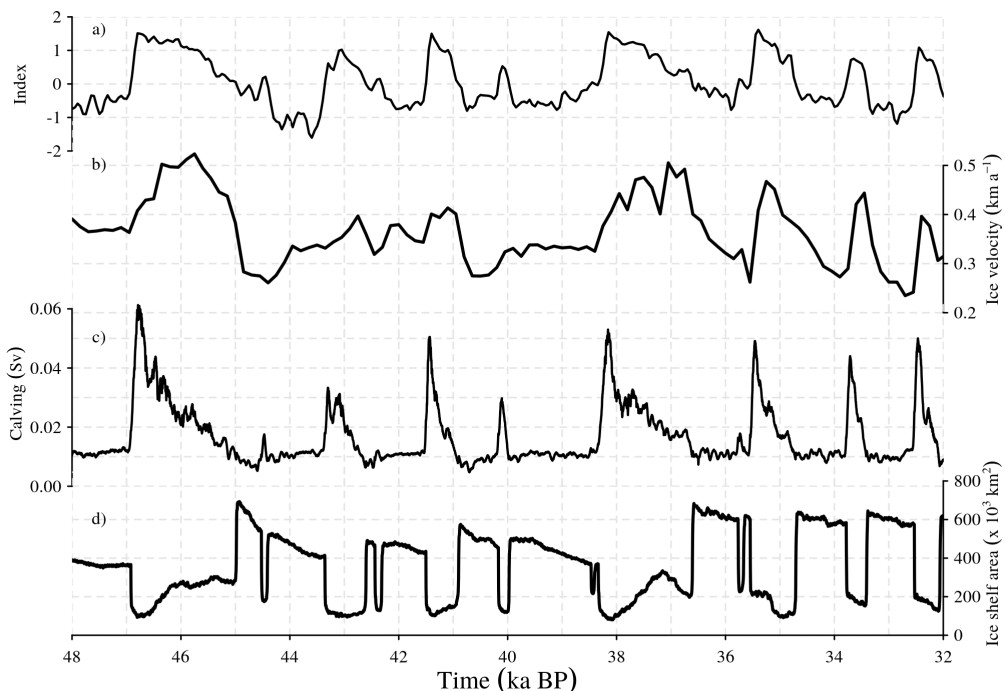

**Figure 9.** MIS3 Period. Temporal component of the millennial-scale climatic forcing ($\beta^\star$ index), ice velocities in the Bjørnøyrenna basin, calculated as mean values over the entire basin, (km a$^{-1}$), calving rate (Sv) and ice-shelf area ($10^3\ km^2$) in the OCNsrf simulation.

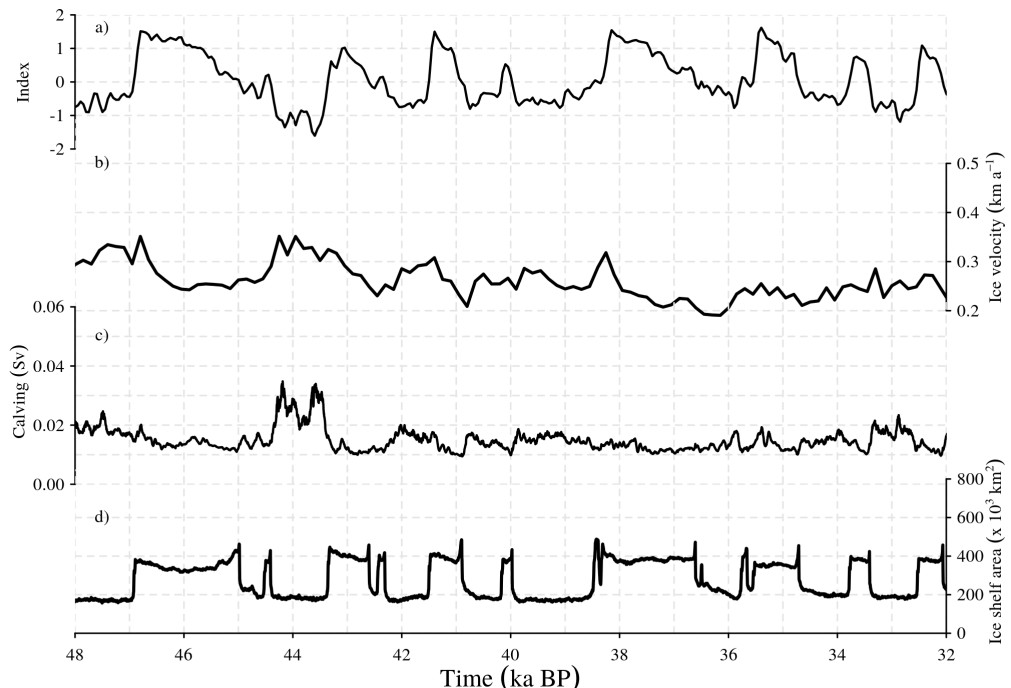

**Figure 10.** MIS3 Period. Temporal component of the millennial-scale climatic forcing ($\beta^\star$ index), ice velocities in the Bjørnøyrenna basin calculated as mean values over the entire basin ($km\ a^{-1}$), calving rate (Sv) and ice-shelf area ($10^3\ km^2$) in the OCNsub simulation.

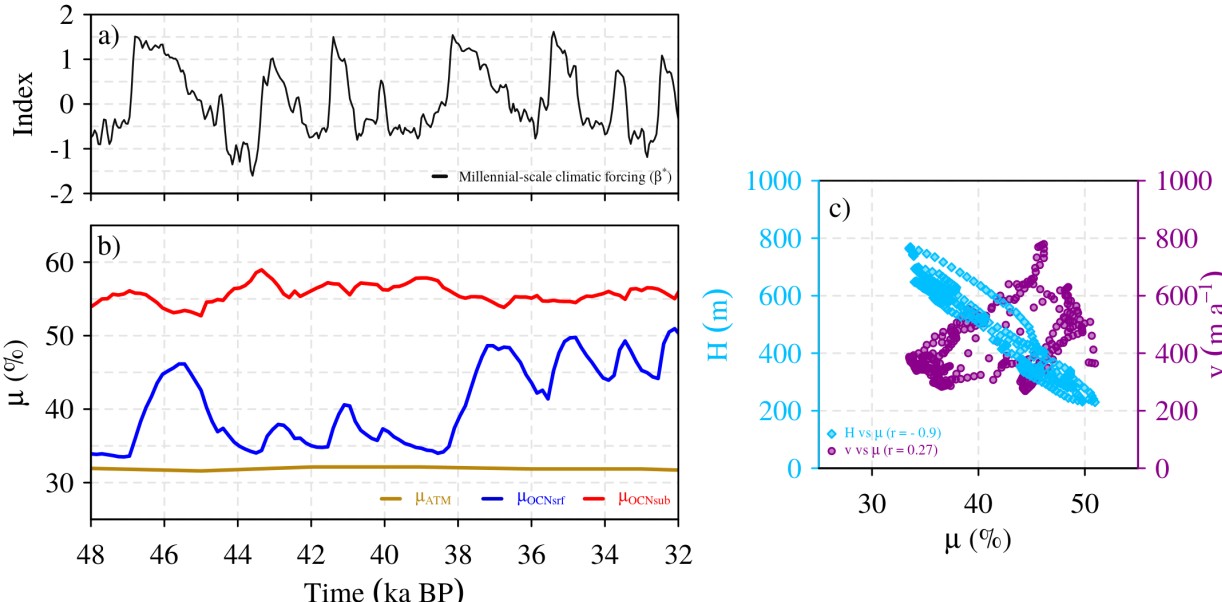

**Figure 11.** Dynamic behavior of the EIS during millennial-scale climatic transitions for the OCNsrf, OCNsub and ATM experiments. Displacement of the grounding line in the Bjørnøyrenna basin (**b**) in response to the climatic $\beta^\star$ forcing (**a**). The evolution of the grounding line position is shown for OCNsrf (blue), OCNsub (red) and ATM (gold). The migration of the grounding line has been characterized as an index $\mu(t)$ that represents the evolution of the number of points of grounded ice $N_g(t)$ over a fixed area of $N$ points in the Barents Sea region, defined over the black square highlighting the jørnøyrenna basin shown in Figure 2. Increasing values of $\mu$ indicate grounding line retreat. **b)** OCNsrf scatter plot diagram showing the relationship between mean ice thickness $H$ in the region of the Bjørnøyrenna basin and $\mu$ (light blue diamonds) as well as the relationship between ice-stream velocities $v$ in the same region and $\mu$ (purple circles).

| Variable \| Parameter | Identifier name | Standard value | Explored range | Units |
|---|---|---|---|---|
| Basal friction coefficient on sediments | $c_f$ | $2\times 10^{-5}$ | – | $\text{a m}^{-1}$ |
| Basal friction coefficient on bedrock | $c_f$ | $20\times 10^{-5}$ | – | $\text{a m}^{-1}$ |
| Standard deviation of daily temperature | $\sigma$ | 5 | [4 - 6] | K |
| Snow conversion factor from PDDs to melt | $f_{PDD_{snow}}$ | 0.003 | [0.0015 - 0.006] | $\text{mwe PDD}^{-1}$ |
| Ice conversion factor from PDDs to melt | $f_{PDD_{ice}}$ | 0.008 | [0.004 - 0.016] | $\text{mwe PDD}^{-1}$ |
| Ice thickness threshold for calving | $H_{calv}$ | 150 | [10 - 500] | m |
| Oceanic sensitivty for ice-shelf melting | $\kappa$ | 5 | [0 - 10] | $\text{m a}^{-1}K^{-1}$ |

**Table 1.** Model parameters used in this study with their standard and explored values

| | Millennial-scale forcing component | | | |
|---|---|---|---|---|
| Experiment name | Atmosphere | Surface ocean | Subsurface ocean | Sea level |
| $CTRL$ | · | · | · | · |
| $SL$ | · | · | · | ✓ |
| $ATM$ | ✓ | · | · | · |
| $OCN_{srf}$ | · | ✓ | · | · |
| $OCN_{sub}$ | · | · | ✓ | · |
| $ALL_{srf}$ | ✓ | ✓ | · | ✓ |
| $ALL_{sub}$ | ✓ | · | ✓ | ✓ |

**Table 2.** Millennial-scale components used to force the ice-sheet model in the different experiments shown in this study.