# Peer review of "Ocean-driven millennial-scale variability of the Eurasian Ice Sheet during the Last Glacial Period simulated with a hybrid ice-sheet-shelf model"

_Climate of the Past, 2018_

## Referee Comment (RC1) · Anonymous Referee #1 · 22 Oct 2018

**General comments**

Alvarez-Solas et al. investigate the behaviour of the Eurasian Ice Sheet (EIS) during the Last Glacial Period (LGP), with a particular focus on the Marine Isotope Stage 3 (MIS3). They run a hybrid 3-D ice sheet model with explicit grounding line treatment and calving, using an offline climate forcing which separates orbital- and millennial-scale climate variability. After an initial control simulation providing the initial state for MIS3, the authors carry out a number of transient model experiments comparing the relative importance of the atmosphere and the ocean in driving ice sheet change over the MIS3 period. Particular attention is paid to the atmospheric and oceanic role in forcing ice sheet change during transitions from stadial (cold) to interstadial (warm) periods. Separate experiments are also carried out for how temperature changes in the surface and subsurface ocean affect the EIS.

The authors find a highly dynamic EIS during the LGP, and that ocean forcing dominates ice sheet mass loss and associated sea level rise during stadial-to-interstadial transitions. The imposed ocean forcing is able to force large-scale, abrupt grounding line retreat and associated high rates of ice discharge into the ocean. Conversely, atmospheric forcing (surface ablation) is not found to have a strong effect except in localized sectors, contributing little to overall ice discharge during abrupt climate transitions. They further find that temperature change in the surface ocean induces a stronger ice dynamic response in the ice sheet model than does the subsurface ocean, and that these change occur in an out-of-phase manner. They therefore suggest that ocean surface warming is the most effective forcing of EIS change during MIS3 stadial-to-interstadial transitions. Based on this and previous work (Alvarez-Solas et al. 2013), they argue that ocean-ice sheet interactions can account for "virtually all ice rafting events in the North Atlantic" during MIS3, as manifested in IRD records by Heinrich events from the Laurentide Ice Sheet during stadials, and by ice discharge from the EIS during interstadials.

Ice sheets are regarded as key players during abrupt climate change, but the underlying mechanisms, roles of oceanic versus atmospheric forcing, and involved ice sheet dynamics is far from resolved, as the authors rightly point out. This study is therefore a timely and exciting contribution to the community. The directed focus on MIS3 rather than the entire LGP allows for a more detailed comparison between different forcing, as well as some analysis of the transient ice dynamics using a 3-D ice sheet model, albeit with the model's inherent limitations in parameterizations and spatial resolution (see below). The manuscript is generally well written and nicely illustrated with figures. Some improvements can be made on the structure of the Results and Discussion sections since these are a little hard to follow, perhaps separating at least the Results into different subsections.

   While the ice sheet model dynamics used is fairly standard (hybrid SSA-SIA), the way climate forcing is implemented is more novel (albeit offline). Further, applying the idea of the EIS as a contributor to the North Atlantic IRD record in the framework of a dynamic, transient ice sheet model has not been done in this manner before. The study tests relative contributions of ocean and atmospheric forcing, and further subdivides ocean forcing into surface and subsurface changes, which has not been done for the MIS3 and for the EIS.

   Overall I am positive to the scientific focus and scope of the manuscript. I do however have a few major and a number of minor concerns that I'd like to see addressed. My concerns are mostly related to an incomplete description of the model dynamics and setup, and the need

for a discussion of related uncertainties. I would like to point out that the authors should be able to address most of these concerns without the need for additional model simulations.

**More substantial comments**
*Grid resolution and grounding line treatment*
Given the conclusion that the ocean plays a major role during abrupt ice sheet changes, the model treatment of grounding line dynamics is key. Several studies have shown that for many applications, a resolution of around 1 km often is needed to accurately capture grounding line migration. In addition, it has been shown (e.g. Gladstone et al 2017) that grounding line behaviour is sensitive to the choice of friction law and the physics of subshelf-melting.

Now, given the millennial-scale focus and large spatial scales involved in this study, I suspect that computational constraints do not allow for ice sheet flow to be resolved on such fine spatial scales, especially not with a 3-D finite difference model. Still, since changes to the marine boundary is an integral part of your conclusion, I feel that this point should be acknowledged and discussed in more detail; namely how your relatively coarse model resolution (40 km) affect your findings regarding the key role of the ocean and grounding line dynamics? Particular in light of Figure 9 where grounding line retreat is assessed in more detail.

The aims of the study are clearly described (response of the EIS to millennial-scale climate variability during MIS 3; ice sheet response to atmosphere vs ocean in abrupt glacial climate change). However, since quite some attention is given to stadial-to-interstadial conditions (i.e. abrupt glacial climate change), why not assess one specific DO event (for example DO 12 c. 47 ka, as shown in Figure 6) in more detail? One could for example do twin experiments over a particular DO event, with increased model grid resolution, to really pin down the conditions and dynamics involved, and assessing the uncertainty to model grid resolution in the process.

*Calving*
To me it's not entirely clear how calving is treated. Perhaps a naïve question, but since your grid resolution is 40 km, does this mean that blocks of ice 40x40 km are calved at once? If so, does this affect the ice dynamics in certain regions? Also, do you expect the model to be sensitive to the shelf thickness threshold H_calv you use? For example, in Banderas et al. 2018, where the climate forcing method used in the present study is explored, the same ice sheet model was used and H_calv = 200 m is employed. In contrast, the current study uses 150 m. Perhaps also give some references to the observational and/or theoretical basis of using such a threshold.
   To be clear, I do not suggest you to switch to another calving law; they all have their inherent flaws and uncertain parameters, especially for paleo-applications. Still, we know that model behaviour differs with the choice of calving law, so I think a more detailed discussion is warranted, also since calving is a key element of the EIS ice discharge that supposedly produces IRD during the modelled period MIS3. Also see my comment on p4, l13-17 below.

*Sensitivity to atmospheric forcing*

The sensitivity to the PDD parameters are tested thoroughly as shown in the Supplementary. Though it's becoming increasingly outdated, I can accept the use of PDD in this study. However, I'm missing some discussion regarding the underlying assumptions of the PDD model. In light of your aims and experimental setup, what's the rationale for using PDD, and not another parameterization, for example including changes in insolation (e.g. Robinson and Goelzer, 2014)? Would the use of PDD put any biases to the SMB fields? If so, in what regions? How would this influence mass loss and would it change your conclusion regarding the ocean vs the atmosphere? I suspect it won't but if this is what you expect, this should nevertheless be pointed out.

*Sensitivity to ocean forcing*

You find that the ocean has an important role in rapid ice sheet changes, and suggest based on your comparison of OCNsrf and OCNsub that surface ocean temperature is a more important driver than subsurface temperature. This appears at first glance counterintuitive, given present-day evidence from Greenland and Antarctica, where warming subsurface waters are regarded most important, since subsurface waters reach grounding lines and induce basal melting, and the properties (temp, salinity) of these subsurface waters would therefore control mass loss from basal melt, as you also point out in p.6, l.8-9. Now, if I understand your model setup correctly, you are not comparing the effect of **concurrent** surface **warming** with subsurface **warming**, but surface **warming** with subsurface **cooling** (opposite sign of anomalies in Fig. 2c and d) in your experiments OCNsrf and OCNsub. Even so, you do get much smaller amplitude ice volume changes (except c. 44 ka) with subsurface warming than with surface warming, and out-of-phase ice volume variations, as nicely illustrated in Fig. 4. I think this could be made more clear.

An explanation for the stronger response for OCNsrf than OCNsub is presented (ice sheet configuration with extensive shallow grounding lines more sensitive to surface than subsurface warming) but relies heavily on the model representation of grounding line dynamics, basal melting and model resolution along marine margins. You touch on these aspects in p. 10, l.29-34, but I think your finding that the experiment OCNsrf gives higher amplitude changes for the EIS than OCNsub would need to be explained and discussed further.

I agree with you that detailed assessments of the mechanisms of abrupt climate change is beyond the scope of this paper (as you point out at the end of the Discussion), and perhaps requires online coupled climate-ice sheet models. Nevertheless, I think you could briefly mention (in Discussion) what potential implications your found ocean-dominated regime have for abrupt climate change in general and MIS3 in particular.

You have nicely illustrated that whether the surface or subsurface dominates may be a question of the ice sheet configuration (e.g. p12, l20-22). Not only that, but you have attempted to link the rate of temperature change (e.g. p10, l29-30) to the question whether surface or subsurface ocean heat matter for the ice sheet, and also compared the impact on different regions. These are exciting findings and could be made even more visible than in the present manuscript. In this aspect, a more in-depth discussion of how you represent

grounding line dynamics (see above) and basal melt (see specific comments) seem all the more important.

*Contribution from different sectors*
The role of grounding line retreat and associated dynamic mass loss from Bjørnøyrenna ice stream is highlighted, along with a description of changes in other sectors (e.g. p9, l30-35). Perhaps some rough numbers could be given for mass fluxes for the different sectors. This would also be helpful for both future model and observational studies building on your study. See detailed comment in Results below.

*Ice sheets' role in abrupt climate events*
Are the time scales of modelled ice sheet change correct for the D-O type abrupt events? (decades from cold to warm). Does the ice sheet change fast enough in your model? Perhaps briefly comment on this in the Discussion.

**Specific comments (mostly minor)**

*Title*
The title is fine, but I'm not sure it gives enough credit to your finding that ocean forcing drives EIS change during MIS3. As it stands, the title could be interpreted as a study which only tests the influence of the ocean on the EIS (which I assume is not what you want). Also as it stands, we have no idea that this is a model study, but including this is personal preference.

*Abstract*
l8. Unclear what "its" refers to
l12. "provides a more realistic treatment of millennial-scale climatic variability than conventional methods" Not clear from the context what conventional methods you refer to here, and therefore why your model approach therefore is "novel"? Try to very briefly clarify this.

*Introduction*
p2, l.10. "its" – awkward phrasing given that you talk about both LIS and EIS in previous sentence
p.2, l19. Please state and provide a reference for why BKSIS is "often considered an analog" for the current WAIS.
p.2, l20-21. "Understanding the underlying mechanisms" **[of what?]** would provide insight into future evolution of the WAIS?
p.2, l23-24. This is true and important, but not unique for the EIS – other ice sheets advancing during the LGM would also have destroyed older evidence. Please rephrase.
p.2, l26. A detail but Finland would perhaps not be considered western Scandinavia, rather use just "Scandinavia".
p.2, l31. "The results" – imprecise wording; what results are you referring to?
p.2, l32-33. high co-variability of the BIIS volume, extent, ice discharge? Not clear what property of the BIIS that co-vary with ocean SSTs, without looking into the underlying reference.

p2-3, l35-1. Please specify that it is sediment cores/records that you refer to here.

p.3, l1. …was identified in [**records from**] the Irminger Sea…

p.3, l4. "as well" – awkward wording

p.3, l5 "just before interstadial transition" – do you mean "just before stadial-to-interstadial transitions"?

p.3, l17. part of **the** LGP. (missing "the"). Also a bit vague, maybe specify which part of the LGP that was modelled in detail in this study.

p3, l15-19. Recent studies by Patton et al 2017 QSR and Åkesson et al 2018 QSR may also be relevant in this literature overview (cf. l23-24 and l28-30).

p3, l20-23. Bassis et al 2017 Nature perhaps relevant.

p3, l.33. Nice overview of the paper – but what's in Section 4?

*Model and experimental setup*

p4, l5-6. Please mention briefly what the underlying assumptions of the SIA and SSA are. Any modeller will know this, but non-modellers might need a reminder.

p4, l7-8. Given the importance and uncertainty of basal drag on ice sheet dynamics, I think it would be helpful to briefly elaborate on how you represent basal drag and what the underlying assumptions are, e.g. type of sliding law, any non-linearity, treatment of sediments if any, spatial distribution of basal drag coefficient, if used, etc.

p4, l11-12. Are these arbitrary numbers or do they have some physical meaning? The criterion where you "activate" SSA could have an impact on your modelled ice velocities, grounding line retreat and ice discharge and should therefore be discussed.

p4, l14. criterium -> criterion

p4, l14. "its thickness" slightly awkward here; use "shelf thickness" to be precise

p4, l15. Please provide a reference for "typical thickness of observed ice-shelf fronts"

p4, l13-17. Please explain briefly the rationale behind using this double criterion, as opposed to, for example, a single ice thickness criterion, or using the Levermann calving law on its own. Also, what happens if there is no shelf in the model (e.g. vertical calving face) – is the calving rate in that case zero? What happens then to the basal melt rate? Given that many vertical termini we know from the present-day are grounded in fjords several hundred meters deep, the thickness criterion would not be reached in this case. Would this have any effect on EIS evolution? (you may include part of this in the Discussion if you wish to keep the model description short)

p4, l22. I know that your focus is not Greenland so this would not affect your conclusions at all, but I don't see the advantage of using the Bamber dataset from 2001, when more recent, more accurate datasets are available (e.g. Bamber et al 2013, Morlighem et al 2014; 2017). On this note, you do include Greenland in your model domain, which I think is indeed interesting and could've been a paper on its own. However, the modelled evolution of the Greenland Ice Sheet is not mentioned in the paper, except being shown in Fig. 1 and 6 and in the supplementary animation. What's the rationale of including Greenland, when the focus of the paper is the EIS? Is there a scientific motive or just a technical reason?

p4, l33. "inland" – would rather use "for grounded ice"

p5, l2. the abbreviation SMB has not been introduced yet (should be done at p4, l22)

Misc. regarding the model
- Please provide the model time step, both for ice flow and PDD. A table of model and forcing parameters along with their values/ranges would be useful.

- You mention that GRISLI-UCM is a thermomechanical model (p4, l4), but I can't find any information of the thermal part of model. Are thermomechanical feedbacks involved over the millennial time scales you focus on?
- Is Glacial Isostatic Adjustment included in the model, and if so, how is it accounted for?
- How is the calving rate defined (as plotted in Fig. 5) and how do you separate this from direct basal melt (also in Fig. 5)?

p5, l19-24. Are you here describing characteristics of the CLIMBER modelled climate in the North Atlantic, or reconstructions, or a combination? Please clarify.

p5, l4. parts -> ice

p5, l14-15. Note sure what you mean here by "purely floating ice shelves", please clarify.

p5, l15-19. You assume 10 times lower melt for "purely floating ice shelves" than at the grounding zone (what do you consider as the "grounding zone"?) and justify this with qualitative agreement with "some Greenland glaciers". OK, but given your previously claimed analogue between the Bjørnøyrenna basin (where most of the action in your model happens) and the WAIS, would it be more appropriate to compare your imposed melt rates with Antarctic melt rates? Also gamma = 0.1 seems a bit arbitrary as it stands; did you explore any other values for gamma and found 0.1 to be the "best" one, or did you settle on this directly based on present-day observations in the studies you cite? Note also that the cited Rignot and Jacobs (2002) covers basal melt in Antarctica, and not Greenland. In addition, the studies cited are for Greenland glaciers with floating tongues (Petermann and 79N), which is indeed more relevant than if you were referring to glaciers in Greenland with grounded termini; this should be mentioned.

p7, l14. Great that you're comparing with ice sheet reconstructions. I know that you're not trying to fit the model perfectly to reconstructions but rather to investigate the relative roles of forcings. Personally, I think that aggressive tuning of climate and model parameters to (over)fit the data perfectly will weaken the value of this kind of study, so I applaud you for not going too much down this route. Still, for transparency and to assess your slightly vague "to an extent that satisfactorily agrees with previous reconstructions", I think including a figure comparing with one or two ice sheet reconstructions (e.g. DATED-1 and ICE-5G) would be valuable. Perhaps you could add these reconstructed ice sheet margins in Fig. 1, or if this becomes too messy, add another figure.

p8, l5. applying (missing p)

p8, l10. You give a nice overview of your experiments. Would also be valuable with a table summarizing the experiments and their differences for easy reference (control run, constant vs time-varying atm forcing, surface vs subsurface ocean, sea level etc.)

p8, l13. Please specify that it is refreezing under ice shelves you're talking about here, since you do include refreezing in your SMB model.

*Results*

First off, I think this section would benefit from division into subsections.

p8, l22. "internal ice-sheet variability" – what is exactly in the ice sheet causes this internal variability?

p8, l23. "slight response" – please more specific, how many % variability or ice volume/sea level equivalent? Is this subdued response to sea level forcing what you expect, or surprising (you may link this to previous literature in the discussion)? Do you think your coarse model resolution dampens the response, making it "harder" for grounding lines to retreat, but once they retreat, the response is large since you "instantaneously" remove a big 40x40 km chunk of ice? Or is it something inherent to the sea level forcing? Is the subdued response to sea level the same everywhere, or does sea level forcing induce grounding line retreat in some sectors, related to the particular ice sheet configuration (e.g. deep vs shallow grounding lines)?

p8, l30. This is an exciting result. The anti-phase relationship is not perfectly in phase throughout the LGP, which perhaps should be mentioned. Given that your SST and subsurface anomalies (Fig. 2cd) are of opposite sign, though not with same spatial distribution, I don't think it's too surprising that the ice sheet responds in this anti-phase manner. Still I do think it's in interesting result with relevance both for abrupt climate change during the LGP and for present-day/future, but it requires a more thorough discussion. See also major comment above on ocean forcing.

p9, l5-13. Please check the manuscript to be consistent with the use of $yr^{-1}$ and $a^{-1}$ (as used at p7, l11).

p9, l20. mid panel -> b

p9, l20-35. A very interesting and nice paragraph where you break down EIS change into sectors and try to explain why. I think an additional figure (if feasible) showing ice volume through time for the different sectors you refer to (e.g. SW vs NE) for one or two forcings (for example ALLsub and ALLsrf), would be of great interest and also illustrate the spatial contrasts and their relation to the forcing you outline.

p10, l23. ...are representative of [**the ice sheet response**] during all other stadial-to-interstadial transitions.

p11, l8-12. Great that you're trying to quantify the grounding line retreat, I think this analysis strengthens the paper. Firstly, over what "fixed area" (line 11) do you define mikro? Is it the square highlighting the Bjørnøyrenna basin shown in Fig. 1c? Secondly, your definition of mikro appears to represent the percentage of non-grounded grid points in the Bjørnøyrenna basin, so that increasing mikro (more non-grounded grid points) corresponds to grounding line retreat. While there is nothing formally wrong with this definition, I wonder if it would be clearer to just use the grounded ice sheet area as your metric for grounding line retreat. Grounded ice area could be shown in Fig. 9 on two different y-axis, one in (%) and one in ($km^2$). See also comments below on Fig 9.

p11, l18-19. I think this an interesting point. For your experiment OCNsrf, you've found a quite close relation between ice thickness H and the number of non-grounded points in the Bjørnøyrenna basin (right panel in Fig. 9). Is this the same as saying that the grounded area and ice thickness in this basin scales linearly? I.e. that the more extensive grounding line retreat (higher mikro), the thinner ice sheet (lower H)? And conversely, a thickening ice sheet translates linearly into grounding line advance? Is this what we expect? Does this mean that ice sheet thinning and grounding line retreat occurs more or less at the same rate, i.e. are tightly coupled? There is also an "anomalous" branch of your H vs mikro plot, where grounding line retreat and thickness temporarily are decoupled. What stage of ice sheet change is this (stadial or interstadial)? What occurs first, grounding line retreat or

thinning? Is this what you expect, or counterintuitive? Just adding a brief discussion on this would be relevant both for both paleo-ice sheet changes and people working with present-day changes in Greenland and Antarctica.

  A related line or two about why v and mikro do not follow such close relationship would also improve the manuscript.

*Discussion*
p12, l2. "some authors" - need reference
p12. l2-3. I would like to congratulate the authors by making the link between the EIS during the LGP and the present-day/future of contemporary ice sheets. However, it's not entirely clear to me from this paragraph whether the authors' findings support or contradict the Kara-Barents complex as a "WAIS analogue". Here I think the relevance of the EIS for present/future changes of Greenland/Antarctica could be strengthened.
p12, l23. grounding line**s**
p12, l23-31. A very important paragraph where the authors outline uncertainties associated with linking calving (flux) and IRD. These uncertainties are outlined nicely, but presently they are not discussed in light of the findings in this study. I also feel that this paragraph would benefit from one or two additional references.
p13, l4. regarding initial ice sheet size – how does your initial ice sheet state entering MIS3 affect subsequent evolution? I don't expect any new simulations in this regard but a brief comment what you expect, particularly since you tuned your basal melt rates at 40 ka to obtain an ice sheet in reasonable agreement with reconstructions.
p13, l14. Rignot et al. 2002 -> Rignot and Jacobs, 2002. See also comment above (Section 2.2) on justifying your magnitudes of basal melt against data from Antarctica vs Greenland.

*Conclusions*
Well written. Consider including your finding about surface vs subsurface ocean. A brief statement on uncertainties in ice sheet dynamics/grounding line dynamics could also be included. I think you may also mention that you explicitly include calving in your model, and very briefly how oceanic basal melt is parameterized.

*Misc.*
  • check consistency of Bjørnøyrenna vs Bjørnøyrenn throughout text and figure captions.

**Figures**
Generally nice and clear figures. Some panels within the figures are missing abcd labels (Fig. 2, 7, 8, 9). To help the reader, make sure you make according changes in places within the text where you refer to different panels of these figures.
*Figure 6.* previous -> prior.
*Figure 7b.* ice velocities in the Bjørnøyrenna basin – how are these defined? Mean velocities over the entire basin? (same in Fig 8b)
*Figure 7c.* I like that you plot the calving rate in (Sv) for oceanographic relevance – also consider adding a second axis in mass loss per year (Gt/a) for the glaciologists reading this. (same in 8c)
*Figure 7d.* ice shelf extension – would rather use "ice shelf area" to emphasize you're showing area, not length. Check in text to be consistent. (same in Fig 8d)

*Figure 9.* A nicely plotted interesting figure. I would put "grounding line index \mikro (%)" as ylabel instead of just mikro (%) to help the reader, unless you follow my suggestion above to use the grounded area as a metric instead. In the caption, please also cross-reference where in the text the index mikro(t) is defined (Eq. 18). For ice thickness H, is this the mean ice thickness in the square shown in Fig. 1c? Ice stream velocities v, over what region are they defined? Finally, I would label this figure with abc, to more clearly refer to each panel in the text (e.g. p11, l13-21).

**Supplementary**
*Fig. S1 and S2.* Though it should be obvious to most readers, please spell out "S.l." in the yaxis label, as you've done in Fig. 3.

*Animation*. Should the units of time in the animation be changed ka -> a?
Also, unless I'm misinterpreting something, the model seems completely off when it comes to getting rid of ice in the Holocene (see screen dump from your animations below). You're modelling the evolution all the way to the present-day but northern Europe is still under ice in your model at 0.0 ka BP, so is northern Russia. Do you have an idea why? I know this is not the period you focus on, but people seeing the animation may take this large disagreement as a sign of something completely missing in your model. Given the severe mismatch, I think an explanation should be included in the manuscript.

[Figure]

*Figure*. Screen dumps from supplementary animation of modelled ice sheet state at 0 ka BP (present-day), for experiments ALLsub, ALLsrf and ATM.

**References**

Åkesson, H., Morlighem, M., Nisancioglu, K. H., Svendsen, J. I., & Mangerud, J. (2018). Atmosphere-driven ice sheet mass loss paced by topography: Insights from modelling the south-western Scandinavian Ice Sheet. *Quaternary Science Reviews*, *195*, 32-47.

Banderas, R., Alvarez-Solas, J., Robinson, A., & Montoya, M. (2018). A new approach for simulating the paleo-evolution of the Northern Hemisphere ice sheets. *Geoscientific Model Development*, *11*(6), 2299-2314.

Bamber, J. L., Layberry, R. L., & Gogineni, S. P. (2001). A new ice thickness and bed data set for the Greenland ice sheet: 1. Measurement, data reduction, and errors. *Journal of Geophysical Research: Atmospheres*, *106*(D24), 33773-33780.

Bamber, J. L., Griggs, J. A., Hurkmans, R. T. W. L., Dowdeswell, J. A., Gogineni, S. P., Howat, I., ... & Steinhage, D. (2013). A new bed elevation dataset for Greenland. *The Cryosphere*, *7*(2), 499-510.

Gladstone, R. M., Warner, R. C., Galton-Fenzi, B. K., Gagliardini, O., Zwinger, T., & Greve, R. (2017). Marine ice sheet model performance depends on basal sliding physics and sub-shelf melting. *The Cryosphere*, *11*, 319-329.

Morlighem, M., Rignot, E., Mouginot, J., Seroussi, H., & Larour, E. (2014). Deeply incised submarine glacial valleys beneath the Greenland ice sheet. *Nature Geoscience*, *7*(6), ngeo2167.

Morlighem, M., Williams, C. N., Rignot, E., An, L., Arndt, J. E., Bamber, J. L., ... & Fenty, I. (2017). BedMachine v3: Complete bed topography and ocean bathymetry mapping of Greenland from multibeam echo sounding combined with mass conservation. *Geophysical research letters*, *44*(21).

Patton, H., Hubbard, A., Andreassen, K., Auriac, A., Whitehouse, P. L., Stroeven, A. P., ... & Hall, A. M. (2017). Deglaciation of the Eurasian ice sheet complex. *Quaternary Science Reviews*, *169*, 148-172.

Robinson, A., & Goelzer, H. (2014). The importance of insolation changes for paleo ice sheet modeling. *The Cryosphere*, *8*(4), 1419-1428.

---

## Referee Comment (RC2) · Anonymous Referee #2 · 8 Nov 2018

Alvarez-Solas et al. investigate the millenial scale variability of the Eurasian ice sheet during the last glacial period. They use an ice sheet model forced offline by a combination of two glacial climatic snapshots, stadial and interstadial. The relative importance of the two snapshots is weighed by an index constructed from a Greenland temperature reconstruction. In their model framework, Alvarez-Solas et al. show that oceanic perturbations induce much greater ice volume changes compared to atmospheric perturbations. They discuss their ice volume variations with respect to IRD layers in marine sediments.

The paper tackles definitively very interesting questions regarding the role of the ocean

in the (in)stability of large marine ice sheets. Little has been done with this respect on the Eurasian ice sheet while a fair amount of geological constraints exist. I think the paper is well written and generally nicely illustrated but I have a few important comments that I would like to see addressed.

**General comments**

- Basal melting rate and ice volume. I am very happy to see that the authors have chosen to change their basal melting rate formulation compared to their previously submitted version of the manuscript (doi: 10.5194/cp-2017-143) so they no longer use a negative sub-shelf melting rate (ice accretion). However I am surprised that the change in setup, and subsequent change in results, does not relate to any change in conclusion nor discussion. In the previous version of the manuscript, during the transignt simulation, the ice volume was oscillating around the 40k spun-up ice volume. In the new version, the ice volume is now perpetually decreasing from 110k to 10k when using the oceanic forcing with kappa>1. As far as I understand your methodology, we expect the 40k ice sheet to be representative of a mean state of the MIS3 ice sheet and your millenial scale index should translate into waxing and waning of the ice sheet around the mean state. The fact that you have a negative trend in ice volume suggests that the model is unable to regrow ice after the imposed oceanic perturbation. I understand that is a complicated issue that cannot be resolved with such a simple index perturbation. However, it seems to me that it is not straightforward to draw robust conclusions on the physical mechanism for MIS3 ice volume oscillations when the model is currently unable to simulate an Eurasian ice sheet that survive to these oscillations. I might be missing something but I think this issue should be clarified and clearly discussed in the paper. As a side note: I could not find the volume your 40k spun-up ice sheet. This is needed to interpret the importance of the trend (8 to 12 m sle!).

- On the method, 1. Because CLIMBER3- $\alpha$  underestimate the stadial to interstadial temperature change at NGRIP, beta\* in the paper has been scaled to match the recorded amplitude. One can wonder if this scaling is appropriate for oceanic fields.
In the atmosphere the millenial anomaly simulated by CLIMBER at NGRIP is about 5-6 degrees, this is why you have roughly a beta that oscillates between -1.5 and 1.5 (amplitude 3) to reproduce a stadial to interstadial of about 15 degrees. In the ocean, CLIMBER also simulates SST anomalies of about 5-6 degrees around the British Isles, meaning that your oceanic temperature during certain DO events can increase by more than 15 degrees. Is this supported by any SST record? This makes me wonder about your experimental design that puts a critical weigh on the ocean...

- On the method, 2. Your base value for sub-shelf basal melting rate is 0.1 m/yr. Since you have a linear basal melting rate perturbation (Eq. 14), given your oceanic anomalies and a Kappa at 5 m/K/yr, for negative values of Beta (roughly half the time) you end up with B(t)

P1 L18-20 This is a strong assertion which seems overconfident to me based on the limitations of the experimental design. Please remove.

P2 L2 Do you mean BKIS?

P2 L19-21 This is arguable. Climatically speaking, the two ice sheets are in a very different context (latitude, AMOC, storm tracks...)

P2 L31-32 No direct evidence for ice volume but ice extent.

P3 L1-2 Since the Greenland ice sheet is included in your geographical domain, is this also reproduced in your simulations?

P4 L2-3 Perhaps you could include a section in the discussion on the limitation of the floatation criteria on a 40km grid resolution, as this is though to be inaccurate to compute grounding line migration. Do you think you would have different grounding line migration sensitivities with a much higher resolution at the grounding line or with a analytical flux at the grounding line?

P4 L26 When using the PDD method, you are discarding the role of insolation changes. Could you add a justification on why this is negligible?

P5 L23 Again, it could be nice to have a plot of the stadial to intersadial temperature change in the atmosphere and in the ocean from a "typical" DO event (beta from -1.5 to 1.5).

P6 L7 To facilitate the reading your standard value of Kappa can appear here.

P6 L17 Bgl not presented before.

P6 L21 Why 750m? It seems relatively low as we have ice shelves today at much greater depth in Antarctica.

P7 L11 See general comments. Justify/discuss the importance of the chosen value.

P7 L29-31 P8 L1-3 This is not clear to me why you did not use the 3D field computed
from CLIMBER3- $\alpha$ . Since the ice sheet model provides you the depth of the ice base you can easily read the temperature simulated by your climate model at this depth.

P7 L5 Section 5 is the conclusion.

P8 L25 What it the volume of your spun-up ice sheet? How small is 1.5 m sle relative to this volume? 10

P9 L21-25 This is unconvincing because a map of SMB changes from stadial to interstadial is missing. SMB is negative at the continental margins, from the BIIS to the BKIS. From your equations, it seems that you impose an important change in surface temperatures (please show as well annual and July temperature changes!) so it is hard to picture why melt is restricted to a narrow band in the South as you imply.

P9 L25-28 Does CLIMBER3- $\alpha$  provide oceanic temperature changes below the 40k ice sheet? How this is possible? If not, how do you compute the sub-shelf basal melting rate when the ice sheet retreats from its initial position?

P9 L25-28 It might be worth noting that if basal melting is more efficient than surface mass balance this is because you have calving in the ocean. Calving is a very efficient way to remove ice (confirmed by your Fig. 5).

P10 L8-9 The retreat pattern of Fennoscandian ice sheet is somewhat surprising. It seems that the ice sheet retreats increased basal melting in the Baltic sea (which is a lake in your setup right?)?

P14 L5 Alvarez-Solas et al. (2013) show that is the subsurface warming caused by AMOC slowdown is responsible for LIS H-events. When subsurface temperature is used here you end up basically with the same synchronisation for EIS and LIS. It is not really convincing to use subsurface temperature for one ice sheet and surface temperature for the other. Again, GRISLI gives you the depth of the ice shelf base so you can use the CLIMBER3- $\alpha$  layer corresponding to this depth, for the LIS and for the EIS. In this case, the study would have been more convincing. Consider reformulation
here.

Fig. 2 Please mention in the caption that these fields are later scaled to reproduce the NGRIP stadial to interstadial temperature change (temporally variable factor but roughly 3 times the changes simulated by CLIMBER3- $\alpha$ ). Otherwise this figure might be misleading.

Fig. 2 Around the coasts of Scandinavia you have a CLIMBER SAT anomaly of about 9 degrees which means that during certain DO events you have episodically a local temperature change of about 30 degrees (beta\* from -1.5 to 1.5). I am surprised that such a temperature change do not translate in large SMB perturbations. Any comment?

Fig. 5 What are the dashed grey lines? They do not seem to relate to the major tick marks.

Fig. 6 The southern edge of the BKIS (Taymyr peninsula / Ob river) seems almost not changed in ATM before and after the DO event. You have a beta\* change of almost 2.5 (roughly -1 to 1.5) meaning that you have a change in annual temperature of at least 4x2.5=9 degrees. The southern extension of the BKIS is limited by melt. With an additional 9 degrees in annual temperature (how many in July?), it is not obvious to me why you do not have any melt increase there.

Fig. 7 Episodically the ice shelf extension is abruptly rising (e.g. 45 kaBP) not necessarily linked to any significant change in beta\*, ice sheet velocity nor calving. What is the reason for that?

Supp. Mat. Fig 3 : the standard deviation in ice volume is not a good indication of the amplitude of millenial oscillations. You should correct from the background linear trend or simply compute the standard deviation of the dVdt variable. From the graph on the left, it seems that you do have oscillations of about 2 m sle for certain PDD parameter combination but maybe at a lower frequency. Could you comment on that?
**Technical corrections**

P6 L31 boundary

Fig. 5 Problem in the caption.

---

## Referee Comment (RC3) · Anonymous Referee #3 · 12 Nov 2018

Alvarez-Solas et al. present a modelling study that investigates the impacts of millennial-scale climatic and oceanic forcings on the Eurasian ice sheet during the last glacial period, and in particular during MIS 3. A 3D thermo-mechanical ice-sheet model is used, with an offline forcing that provides a more robust representation of millennial scale climate variabilities compared to traditional methods. Explicit treatment of sub-marine melting within the ice model allows the authors to consider the relative contributions of ice-surface melting (ablation) vs dynamic process related to ice-ocean interactions (ocean surface and subsurface melting).

Results show oceanic forcing plays a dominant role over surface melting in controlling

dynamic losses of the EIS over sub-millennial timescales, as well as its importance spatially. Of particular interest is the predicted role that subsurface ocean temperatures can play in enhancing ice discharge during stadial conditions in the Barents Sea/high latitudes, thus supporting empirical observations for the presence of Eurasian IRD in the North Atlantic during stadials. The approach of the manuscript, alongside sensitivity experiments, appears robust and the results provide an important contribution to further understanding ice-dynamical processes occurring in this understudied domain. I suggest minor revisions to the manuscript based on comments and questions below:

P2L22: In terms of underlying mechanisms contributing to collapse of the BSIS, there are additional papers to cite beyond Gudlaugsson. e.g., ice stream surging (Andreassen et al., 2014); subglacial meltwater (Esteves et al., 2017); subsurface melting/ocean warming (Ivanovic et al., 2018; Rasmussen and Thomsen, 2004).

P2L19: The acronym LGM is not defined

P2L25: It would be useful for readers not familiar with marine isotope stages to also state the timeframe in years BP

P2L32: I think this understates the uncertainty in marine sectors – minimum extents for ice in the Barents/Kara seas during MIS 3 are essentially unknown: Hughes et al. (2016) do not try to speculate on limits here prior to 32 ka BP. It would be appropriate to discuss the glacial history of the Eurasian Arctic during MIS3 within the context of long-term IRD records (e.g., Kleiber et al., 2001; Knies et al., 2001; Mangerud et al., 1998).

P3L10: Also should mention Petrini et al. (2018), which does have explicit treatment of ocean forcing for modelling retreat of the BSIS. Also possibly (Ivanovic et al., 2018) in terms of HS 1.

P3L17: And Patton et al. (2017).

P3L30: Missing section 4.

P4L12: What is the basis for these thresholds for SSA activation?

P4L14: criterium->criterion

P4L15: citation needed for the typical observed Hcalv.

P4L21: "Initial" topographic conditions infers GIA is accounted for, but is not described in the model description.

P4L26: It does not appear that any account has been taken for the contribution of insolation-induced melt during MIS 3 (e.g., Robinson and Goelzer, 2014).

P5L4: SMB not defined

P5L23: AMOC not defined

P6L8,9: Some citations here would be useful.

P6L18: The submarine melt rate for ice shelves appears somewhat arbitrary and does not appear to consider possible refreezing associated with supercooling (e.g., Jenkins and Doake, 1991). While Bgl is undoubtedly more important in terms of the glacial response, will modifying this coefficient of 0.1 likely introduce any major differences on the results?

P7L15: This statement on the agreement with previous reconstructions appears confusing – neither study cited shows reconstructed margins during MIS 3 at 40 ka BP. Are the authors instead referring to the glacial maximum of the Mid Weichselian (MIS 4/3) at ∼60 ka?

P8L5: wrong section cited.

P8L25: It would be useful to see this value in relation to total ice volume of the ice sheet.

P9L31: 'British-Irish'

P10L23: 'of all the other'

P12L3-5: The reason for linking these two statements is not clear unless it's mentioned also the susceptibility of the WAIS to oceanic warming.

P12L32: This is a useful section that discusses the major limitations of the present study and where future work is needed. The authors however do not mention the limitation of the grid resolution at the grounding line within the context of insights into the EIS responses across sub-millenial timescales. The use of an index to track grounding line dynamics is interesting and a very useful tool, although some mention of the simplifications on grounding line migration would be appropriate to mention given the main conclusions e.g., response time to abrupt forcing.

P12L15: Along the southwest EIS (Irish/Scottish margin) at least. This effect of increased IRD during stadials is not observed by Becker (2017) further north along the mid Norwegian margin during MIS3.

P12L17: Citation

P13L18: citation needed.

P13L34: Should mention here in the conclusions the anti-phase effects of the subsurface warming.

Figures: Bjørnøyrenna is misspelled among figure captions. Missing figure lettering on Fig 2 & 9.

Figure 6: It appears from the OCNsrf timeslices that the Baltic region of the FIS is dramatically affected by ocean surface temperature forcing even though this area was disconnected from the North Atlantic. Is there any provision in the model to distinguish freshwater vs. ocean?

Figure 9: Mean/max ice thickness?

References:

Andreassen, K., Winsborrow, M.C.M., Bjarnadóttir, L.R., Rüther, D.C., 2014. Ice stream

retreat dynamics inferred from an assemblage of landforms in the northern Barents Sea. Quat. Sci. Rev. 92, 246–257. doi:10.1016/j.quascirev.2013.09.015

Becker, L.W.M., Sejrup, H.P., Hjelstuen, B.O., Haflidason, H., Dokken, T.M., 2017. Ocean-ice sheet interaction along the SE Nordic Seas margin from 35 to 15ka BP. Mar. Geol. doi:10.1016/j.margeo.2017.09.003

Esteves, M., Bjarnadóttir, L.R., Winsborrow, M.C.M., Shackleton, C.S., Andreassen, K., 2017. Retreat patterns and dynamics of the Sentralbankrenna glacial system, central Barents Sea. Quat. Sci. Rev. 169, 131–147. doi:10.1016/j.quascirev.2017.06.004

Hughes, A.L.C., Gyllencreutz, R., Lohne, Ø.S., Mangerud, J., Svendsen, J.I., 2016. The last Eurasian ice sheets – a chronological database and time-slice reconstruction, DATED-1. Boreas 45, 1–45. doi:10.1111/bor.12142

Ivanovic, R.F., Gregoire, L.J., Burke, A., Wickert, A.D., Valdes, P.J., Ng, H.C., Robinson, L.F., McManus, J.F., Mitrovica, J.X., Lee, L., Dentith, J.E., 2018. Acceleration of Northern Ice Sheet Melt Induces AMOC Slowdown and Northern Cooling in Simulations of the Early Last Deglaciation. Paleoceanogr. Paleoclimatology 33, 807–824. doi:10.1029/2017PA003308

Jenkins, A., Doake, C.S.M., 1991. Ice-ocean interaction on Ronne Ice Shelf, Antarctica. J. Geophys. Res. Ocean. 96, 791–813. doi:10.1029/90JC01952

Kleiber, H.P., Niessen, F., Weiel, D., 2001. The Late Quaternary evolution of the western Laptev Sea continental margin, Arctic Siberiaâ̌ implications from sub-bottom profiling. Glob. Planet. Change 31, 105–124. doi:10.1016/S0921-8181(01)00115-1

Knies, J., Kleiber, H.-P., Matthiessen, J., Müller, C., Nowaczyk, N., 2001. Marine ice-rafted debris records constrain maximum extent of Saalian and Weichselian ice-sheets along the northern Eurasian margin. Glob. Planet. Change 31, 45–64. doi:10.1016/S0921-8181(01)00112-6

Mangerud, J., Dokken, T., Hebbeln, D., Heggen, B., Ingólfsson, Ó., Landvik, J.Y., Mejdahl, V., Svendsen, J.I., Vorren, T.O., 1998. Fluctuations of the Svalbard-Barents Sea Ice Sheet during the last 150 000 years. Quat. Sci. Rev. 17, 11–42. doi:10.1016/S0277-3791(97)00069-3

Patton, H., Hubbard, A., Andreassen, K., Auriac, A., Whitehouse, P., Stroeven, A.P., Shackleton, C., Winsborrow, M.C.M., Heyman, J., Hall, A.M., 2017. Deglaciation of the Eurasian ice sheet complex. Quat. Sci. Rev. 169, 148–172. doi:10.1016/j.quascirev.2017.05.019

Petrini, M., Colleoni, F., Kirchner, N., Hughes, A.L.C., Camerlenghi, A., Rebesco, M., Lucchi, R.G., Forte, E., Colucci, R.R., Noormets, R., 2018. Interplay of grounding-line dynamics and sub-shelf melting during retreat of the Bjørnøyrenna Ice Stream. Sci. Rep. 8, 7196. doi:10.1038/s41598-018-25664-6

Rasmussen, T.L., Thomsen, E., 2004. The role of the North Atlantic Drift in the millennial timescale glacial climate fluctuations. Palaeogeogr. Palaeoclimatol. Palaeoecol. 210, 101–116. doi:10.1016/J.PALAEO.2004.04.005

Robinson, A., Goelzer, H., 2014. The importance of insolation changes for paleo ice sheet modeling. Cryosph. 8, 1419–1428. doi:10.5194/tc-8-1419-2014

---

## Author Comment (AC1) · 8 Mar 2019

**General comments**

Alvarez-Solas et al. investigate the behaviour of the Eurasian Ice Sheet (EIS) during the Last Glacial Period (LGP), with a particular focus on the Marine Isotope Stage 3 (MIS3). They run a hybrid 3-D ice sheet model with explicit grounding line treatment and calving, using an offline climate forcing which separates orbital- and millennial-scale climate variability. After an initial control simulation providing the initial state for MIS3, the authors carry out a number of transient model experiments comparing the relative importance of the atmosphere and the ocean in driving ice sheet change over the MIS3 period. Particular attention is paid to the atmospheric and oceanic role in forcing ice sheet change during transitions from stadial (cold) to interstadial (warm) periods. Separate experiments are also carried out for how temperature changes in the surface and subsurface ocean affect the EIS.

The authors find a highly dynamic EIS during the LGP, and that ocean forcing dominates ice sheet mass loss and associated sea level rise during stadial-to-interstadial transitions. The imposed ocean forcing is able to force large-scale, abrupt grounding line retreat and associated high rates of ice discharge into the ocean. Conversely, atmospheric forcing (surface ablation) is not found to have a strong effect except in localized sectors, contributing little to overall ice discharge during abrupt climate transitions. They further find that temperature change in the surface ocean induces a stronger ice dynamic response in the ice sheet model than does the subsurface ocean, and that these change occur in an out- of-phase manner. They therefore suggest that ocean surface warming is the most effective forcing of EIS change during MIS3 stadial-to-interstadial transitions. Based on this and previous work (Alvarez-Solas et al. 2013), they argue that ocean-ice sheet interactions can account for "virtually all ice rafting events in the North Atlantic" during MIS3, as manifested in IRD records by Heinrich events from the Laurentide Ice Sheet during stadials, and by ice discharge from the EIS during interstadials.

Ice sheets are regarded as key players during abrupt climate change, but the underlying mechanisms, roles of oceanic versus atmospheric forcing, and involved ice sheet dynamics is far from resolved, as the authors rightly point out. This study is therefore a timely and exciting contribution to the community. The directed focus on MIS3 rather than the entire LGP allows for a more detailed comparison between different forcing, as well as some analysis of the transient ice dynamics using a 3-D ice sheet model, albeit with the model's inherent limitations in parameterizations and spatial resolution (see below). The manuscript is generally well written and nicely illustrated with figures. Some improvements can be made on the structure of the Results and Discussion sections since these are a little hard to follow, perhaps separating at least the Results into different subsections.

While the ice sheet model dynamics used is fairly standard (hybrid SSA-SIA), the way climate forcing is implemented is more novel (albeit offline). Further, applying the idea of the EIS as a contributor to the North Atlantic IRD record in the framework of a dynamic, transient ice sheet model has not been done in this manner before. The study tests relative contributions of ocean and atmospheric forcing, and further subdivides ocean forcing into surface and subsurface changes, which has not been done for the MIS3 and for the EIS.

Overall I am positive to the scientific focus and scope of the manuscript. I do however have a few major and a number of minor concerns that I'd like to see addressed. My concerns are mostly related to an incomplete description of the model dynamics and setup, and the need for a discussion of related uncertainties. I would like to point out that the authors should be able to address most of these concerns without the need for additional model simulations.

We are glad for the reviewer's overall positive attitude toward the focus and scope of our manuscript. We are also very grateful for their careful reading of the manuscript and the numerous suggestions that undoubtedly have improved it and contributed to make the results more clear. We have attempted to address their concerns by providing a more complete description of the model dynamics and setup. Below we give a detailed response to each of the comments.

**More substantial comments**

*Grid resolution and grounding line treatment*

*Given the conclusion that the ocean plays a major role during abrupt ice sheet changes, the model treatment of grounding line dynamics is key. Several studies have shown that for many applications, a resolution of around 1 km often is needed to accurately capture grounding line migration. In addition, it has been shown (e.g. Gladstone et al 2017) that grounding line behaviour is sensitive to the choice of friction law and the physics of sub shelf-melting.*

*Now, given the millennial-scale focus and large spatial scales involved in this study, I suspect that computational constraints do not allow for ice sheet flow to be resolved on such fine spatial scales, especially not with a 3-D finite difference model. Still, since changes to the marine boundary is an integral part of your conclusion, I feel that this point should be acknowledged and discussed in more detail; namely how your relatively coarse model resolution (40 km) affect your findings regarding the key role of the ocean and grounding line dynamics? Particular in light of Figure 9 where grounding line retreat is assessed in more detail.*

This is an important point raised by all three reviewers. We have attempted to acknowledge this in the Discussion section:

**R1 MC1, MC7 & R2 SC2e & R3 SC4b:** *Given the conclusion that the ocean plays a major role in abrupt ice sheet changes, the model's treatment of grounding-line dynamics is a key issue. Several studies have shown that for many applications, a resolution of around 1 km is needed to accurately determine the grounding-line position. In addition, it has been shown (e.g. Gladstone et al 2017) that the grounding-line behaviour is sensitive to the choice of friction law and the physics of submarine melting, and that these determine model-resolution requirements. In our case, the dependence of basal drag on effective pressure allows for the desirable property of basal drag going to zero at the grounding line. However, our basal-melt parameterisation does not provide a smooth transition from grounded to floating ice. Thus our results regarding the key role of the ocean on the grounding line position can be affected by the coarse model resolution. Computational constraints do not allow for the required high model resolution, especially with a 3D finite difference model on these long time scales. However, the potential inaccuracy of the grounding line position introduced by the coarse resolution, typically of ~100 km (Vieli and Payne, 2005; Gladstone et al., 2017) is one order of magnitude smaller than the grounding line migrations simulated here (more than 1000 km). This issue should be investigated in the future both at much higher resolution, as well as including different formulations of friction and submarine melting.*

*The aims of the study are clearly described (response of the EIS to millennial-scale climate variability during MIS 3; ice sheet response to atmosphere vs ocean in abrupt glacial climate change). However, since quite some attention is given to stadial-to-interstadial conditions (i.e. abrupt glacial climate change), why not assess one specific DO event (for example DO 12 c. 47 ka, as shown in Figure 6) in more detail? One could for example do twin experiments over a particular DO event, with increased model grid resolution, to really pin down the conditions and dynamics involved, and assessing the uncertainty to model grid resolution in the process.*

**R1 MC2:** This would indeed be a very interesting study and is actually something we want to assess in detail as a next step. However, it requires a significant amount of technical work. On the other hand, as acknowledged by the referee, our manuscript represents the first attempt to illustrate the EIS behaviour at millennial scale, which already makes the study quite novel. We believe more detailed studies, as the one suggested here, are in the scope of future work within our group.

*Calving*

To me it's not entirely clear how calving is treated. Perhaps a naïve question, but since your grid resolution is 40 km, does this mean that blocks of ice 40x40 km are calved at once? If so, does this affect the ice dynamics in certain regions? Also, do you expect the model to be sensitive to the shelf thickness threshold H_calv you use? For example, in Banderas et al. 2018, where the climate forcing method used in the present study is explored, the same ice sheet model was used and H_calv = 200 m is employed. In contrast, the current study uses 150 m. Perhaps also give some references to the observational and/or theoretical basis of using such a threshold.

To be clear, I do not suggest you to switch to another calving law; they all have their inherent flaws and uncertain parameters, especially for paleo-applications. Still, we know that model behaviour differs with the choice of calving law, so I think a more detailed discussion is warranted, also since calving is a key element of the EIS ice discharge that supposedly produces IRD during the modelled period MIS3. Also see my comment on p4, l13-17 below.

We understand the reviewer's concerns. Indeed, given that the resolution is 40 km, when ice is calved, it implies a block of 40 km x 40 km is removed at once. To assess the sensitivity of the ice dynamics to the value of the thickness threshold below which the ice is calved we have performed a new ensemble exploring a wide value range of this parameter's values, from 10 to 800 meters. These results together with a new figure (Figure S4) are now included in the Supplementary Information. Values of this threshold above 400 m favor a major (and likely unrealistic) disintegration of the Barents-Kara complex due to its relatively shallow bed. The typical value of this parameter is however 150 meters (see Peyaud et al, 2007). The overall effect of this sensitivity test around the preferred value is to modulate the amplitude of the response to the oceanic perturbations (see figure below) but this does not affect the main conclusions of this paper. A more detailed discussion of the calving law, in particular developing the rationale behind the double criterion is also included, explaining the former results together with more references to the observational and theoretical basis (see also comment SC2g below). A more thorough exploration of different calving laws is in the scope of future work that we are currently planning.

**R1 MC3** & R1 SC2g**:** *Calving takes place at the ice-shelf front when two conditions are met. First, the ice-shelf thickness must fall below a threshold $H_{calv}$. This is a semiempirical parameter reflecting the fact that this is the typical thickness of ice-shelf fronts currently observed in Antarctica (Grigg and Bamber 2011). Second, the upstream advection must fail to maintain the ice thickness above this threshold following a semi-Lagrangian approach (Peyaud et al. 2007) to account for the fact that ice-flux divergence fosters the formation of crevasses (Levermann et al. 2012). This method is standard in the GRISLI model. It was introduced after recognising that a systematic cutoff of ice shelves below a given threshold led to a realistic simulation of the present-day ice shelves in Antarctica, as is the case in many models, but prevents any development of new ice shelves (Peyaud 2006; 2007). When focusing on past climates, ice sheets should be able to evolve in response to climate changes, and in particular to allow the advance of ice shelves in cold climates. To this end, before calving ice in a certain point, we test whether advection allows for the growth of ice at the front, and therefore the ice-shelf advance. $H_{calv}$ was set to 150 m, in the standard setup. To assess the sensitivity of the ice dynamics to the value of the thickness threshold $H_{calv}$, we have performed a new ensemble exploring a wide value range of this parameter's values, from 10 to 800 meters (Fig. S4).*

*Values of this threshold above 400 m produce a drastic disintegration of the Barents-Kara complex due to its relative shallow bed. The overall effect of this sensitivity test around the preferred value is to modulate the amplitude of the response to the oceanic perturbations.*

*Sensitivity to atmospheric forcing*

The sensitivity to the PDD parameters are tested thoroughly as shown in the Supplementary. Though it's becoming increasingly outdated, I can accept the use of PDD in this study. However, I'm missing some discussion regarding the underlying assumptions of the PDD model. In light of your aims and experimental setup, what's the rationale for using PDD, and not another parameterization, for example including changes in insolation (e.g. Robinson and Goelzer, 2014)? Would the use of PDD put any biases to the SMB fields? If so, in what regions? How would this influence mass loss and would it change your conclusion regarding the ocean vs the atmosphere? I suspect it won't but if this is what you expect, this should nevertheless be pointed out.

The PDD scheme has limitations for paleo simulations essentially because it neglects insolation-driven melting, which can be very important for long-term simulations including variations at orbital timescales, especially in past warming periods. Nevertheless, because our focus is on the abrupt climate changes of the last glacial period, our reference climate is precisely a glacial climate where insolation variations are of much less importance when compared for example to glacial-interglacial transitions. We have now discussed this point with much detail in the Model description section by adding the following:

R3 SC2k & R2 SC2f & **R1 MC4:** *This melting scheme is admittedly too simple for fully transient paleo simulations, as it omits the contribution of insolation-induced effects on surface melting (Robinson and Goelzer, 2014). Nevertheless, insolation changes are most relevant in long-term simulations including variations at orbital timescales, especially in past warmer periods such as the Eemian. Since this study focuses on abrupt climate changes within a fixed glacial background climate, insolation changes are not important and the PDD melt model should be sufficient to give a good approximation of surface melt in response to interstadials in a reasonable manner.*

*Sensitivity to ocean forcing*

You find that the ocean has an important role in rapid ice sheet changes, and suggest based on your comparison of OCNsrf and OCNsub that surface ocean temperature is a more important driver than subsurface temperature. This appears at first glance counterintuitive, given present-day evidence from Greenland and Antarctica, where warming subsurface waters are regarded most important, since subsurface waters reach grounding lines and induce basal melting, and the properties (temp, salinity) of these subsurface waters would therefore control mass loss from basal melt, as you also point out in p.6, l.8-9. Now, if I understand your model setup correctly, you are not comparing the effect of **concurrent** surface **warming** with subsurface **warming**, but surface **warming** with subsurface **cooling** (opposite sign of anomalies in Fig. 2c and d) in your experiments OCNsrf and OCNsub. Even so, you do get much smaller amplitude ice volume changes (except c. 44 ka) with subsurface warming than with surface warming, and out-of-phase ice volume variations, as nicely illustrated in Fig. 4. I think this could be made more clear.

An explanation for the stronger response for OCNsrf than OCNsub is presented (ice sheet configuration with extensive shallow grounding lines more sensitive to surface than subsurface warming) but relies heavily on the model representation of grounding line dynamics, basal melting and model resolution along marine margins. You touch on these aspects in p. 10, l.29-34, but I think your

finding that the experiment OCNsrf gives higher amplitude changes for the EIS than OCNsub would need to be explained and discussed further.

We are indeed not comparing the effect of concurrent surface warming with subsurface warming, but (roughly) surface warming with subsurface cooling because of the expected anti-phase relationship between both. Smaller amplitude ice volume changes are generally produced with subsurface warming than with surface warming, and out-of-phase ice volume variations. We agree this is in part because OCNsrf gives higher amplitude changes for the EIS than OCNsub, but the rate of change also plays a role; we have explained and discussed this further by including the following:

**R1 MC5** & MC7**:** *This can be explained by the fact that, along the coast of Eurasia, the amplitude of the simulated SST anomalies used to compute basal melting in OCNsrf is larger than the subsurface temperature anomalies in OCNsub, since the basal melt was calculated by using ocean temperature at a fixed depth, either at the surface or at the subsurface. Also, transitions to stadials are usually more gradual than transitions to interstadials, thus the incursion of warmer (subsurface) waters happens in a smoother manner in this case.*

I agree with you that detailed assessments of the mechanisms of abrupt climate change is beyond the scope of this paper (as you point out at the end of the Discussion), and perhaps requires online coupled climate-ice sheet models. Nevertheless, I think you could briefly mention (in Discussion) what potential implications your found ocean-dominated regime have for abrupt climate change in general and MIS3 in particular.

We have added the following (see also comment SC1 below):

**R1 MC6** & SC1n**:** *Our results thus support the existence of a highly dynamic EIS during the LGP. They suggest an important role of oceanic melt forcing through changes in the ocean circulation in controlling the ice-stream activity. A number of studies have considered the interaction between ocean circulation changes and ice-sheet dynamics as a plausible mechanism to explain iceberg discharges from the LIS associated with H events. For example, subsurface oceanic warming during stadials in response to reduced North Atlantic deep water formation (Alvarez-Solas et al., 2010; Flückiger et al., 2006; Mignot et al., 2007; Shaffer et al., 2004) has been shown to be capable of producing large discharges from the LIS, induced by enhanced basal melting rates (Marcott et al., 2011; Alvarez-Solas et al., 2011b). The satisfactory agreement between the simulated calving and North Atlantic marine IRD records provides strong support for this mechanism (Alvarez-Solas et al., 2013), recently proposed to be modulated by isostatic adjustment (Bassis et al., 2017). The evaluation of the impact of these NH discharges on the oceanic circulation and their effects on the triggering mechanism of DO events require the use of a coupled climate-ice-sheet model. Nonetheless, it has recently been shown that the typical oceanic cooling registered in sediment cores of the North Atlantic during stadials occurs before the arrival of the icebergs to these same cores (Barker et al., 2015). In this sense, iceberg discharges from the Laurentide and the Eurasian ice sheets are seen as potential amplifiers but not as active elements in the triggering of millennial-scale variability.*

You have nicely illustrated that whether the surface or subsurface dominates may be a question of the ice sheet configuration (e.g. p12, l20-22). Not only that, but you have attempted to link the rate of temperature change (e.g. p10, l29-30) to the question whether surface or subsurface ocean heat matter for the ice sheet, and also compared the impact on different regions. These are exciting findings and could be made even more visible than in the present manuscript. In this aspect, a more in-depth discussion of how you represent grounding line dynamics (see above) and basal melt (see specific comments) seem all the more important.

We have extended our description of these two issues as follows, first concerning the role of the subsurface and then grounding-line dynamics and basal melt:

R1 MC5 & **MC7:** *Peaks in calving are recorded accordingly during pronounced stadial periods. These peaks are however of smaller amplitude than in OCNsrf. This can be explained by the fact that along the coast of Eurasia, the amplitude of the simulated SST anomalies used to compute basal melting in OCNsrf is larger than the subsurface temperature anomalies in OCNsub, since the basal melt was calculated by using ocean temperature at a fix depth, either at the surface or at the subsurface. Also, transitions to stadials are usually more gradual than transitions to interstadials, thus the incursion of warmer (subsurface) waters happens in this case in a smoother manner.*

R1 MC1, **R1 MC7** & R2 SC2e, R3 SC4b**:** *Given the conclusion that the ocean plays a major role in abrupt ice sheet changes, the model's treatment of grounding-line dynamics is a key issue. Several studies have shown that for many applications, a resolution of around 1 km is needed to accurately determine the grounding-line position. In addition, it has been shown (e.g. Gladstone et al 2017) that the grounding-line behaviour is sensitive to the choice of friction law and the physics of submarine melting, and that these determine model-resolution requirements. In our case, the dependence of basal drag on effective pressure allows for the desirable property of basal drag going to zero at the grounding line. However, our basal-melt parameterisation does not provide a smooth transition from grounded to floating ice. Thus our results regarding the key role of the ocean on the grounding line position can be affected by the coarse model resolution. Computational constraints do not allow for the required high model resolution, especially with a 3D finite difference model on these long time scales. However, the potential inaccuracy of the grounding line position introduced by the coarse resolution, typically of ~100 km (Vieli and Payne, 2005; Gladstone et al., 2017) is one order of magnitude smaller than the grounding line migrations simulated here (more than 1000 km). This issue should be investigated in the future both at much higher resolution, as well as including different formulations of friction and submarine melting..*

*Contribution from different sectors*

The role of grounding line retreat and associated dynamic mass loss from Bjørnøyrenna ice stream is highlighted, along with a description of changes in other sectors (e.g. p9, l30-35). Perhaps some rough numbers could be given for mass fluxes for the different sectors. This would also be helpful for both future model and observational studies building on your study. See detailed comment in Results below.

We have now included the following and have shown it in a new Figure (current Figure 8) where we decompose the contribution from different sectors (see **R1 SC3f**):

**R1 MC8**: *The corresponding mass fluxes reach up to about 0.05~Sv; of these, approximately 0.025 , 0.02 and 0.005~Sv originate in the Barents-Kara, Scandinavia and the British Islands, respectively.*

*Ice sheets' role in abrupt climate events*

Are the time scales of modelled ice sheet change correct for the D-O type abrupt events? (decades from cold to warm). Does the ice sheet change fast enough in your model? Perhaps briefly comment on this in the Discussion.

The DO response (in terms of atmospheric and oceanic patterns) is correct and it is as rapid as observed by construction of the forcing method (temporally scaled to Greenland isotopes). Of course,

the ice-sheet response is slower. However, we are not aware of the existence of proxies (other than global sea-level reconstructions) that show how fast the EIS response is expected to be.

**Specific comments (mostly minor)**

Specific comments (SC) are labeled according to the sections.

*Title*

The title is fine, but I'm not sure it gives enough credit to your finding that ocean forcing drives EIS change during MIS3. As it stands, the title could be interpreted as a study which only tests the influence of the ocean on the EIS (which I assume is not what you want). Also as it stands, we have no idea that this is a model study, but including this is personal preference.

We agree with the reviewer, therefore we have changed the title as follows:

**R1 SC Title:** *Ocean-driven, millennial-scale variability of the Eurasian Ice Sheet during the Last Glacial Period simulated with a hybrid ice-sheet-shelf model*

*Abstract*

l8. Unclear what "its" refers to □l12. "provides a more realistic treatment of millennial-scale climatic variability than conventional methods" Not clear from the context what conventional methods you refer to here, and therefore why your model approach therefore is "novel"? Try to very briefly clarify this.

We have rephrased this paragraph following the referee's suggestion.

**R1 SC Abstract, a:** *Thus, a better understanding of the evolution of the EIS during the LGP is important to understand its role in glacial abrupt climate changes.*

**R1 SC Abstract, b:** *The model is forced offline through a novel perturbative approach that, as opposed to conventional methods, clearly differentiates between the spatial patterns of millennial-scale and orbital-scale climate variability. Thus, it provides a more realistic treatment of the forcing at millennial timescale. The effect of both atmospheric and oceanic variations are included.*

*Introduction*

All points raised below by the reviewer were taken into account by rephrasing the corresponding parts of the paper following their suggestions:

p2, l.10. "its" – awkward phrasing given that you talk about both LIS and EIS in previous sentence□

**R1 SC1a:** *However, improving our understanding of the evolution of the EIS and its response to past climate changes is important for a number of reasons.*

p.2, l19. Please state and provide a reference for why BKSIS is "often considered an analog" for the current WAIS.

**R1 SC1b:** *From a broader perspective, the EIS, consisting of the Fennoscandian, the British Isles and the Barents-Kara ice sheets (FIS, BIIS and BKIS, respectively) contained a large marine-based sector at its maximum extension (Hughes et al., 2016) that was exposed to oceanic variations. The BKIS, in particular, was predominantly marine-based for much of the LGP. For this reason and because it had a similar size like the West Antarctic ice sheet (WAIS) during the Last Glacial Maximum (LGM)*

(Svendsen et al., 2004a; Anderson et al., 2002; Bentley et al. 2014; Denton and Hughes 2002; Evans et al., 2006; Hillenbrand et al. 2012; Whitehouse et al. 2012), it is sometimes considered as a geological analog of the current WAIS (e.g. Gudlaugsson et al., 2013). However, while the WAIS endured the deglaciation, the BKIS completely disappeared (Andreassen and Winsborrow, 2009).

p.2, l20-21. "Understanding the underlying mechanisms" **[of what?]** would provide insight into future evolution of the WAIS?☐

**R1 SC1c:** *Mechanisms contributing to the deglaciation of the BKIS include ice stream surging (Andreassen and Winsborrow, 2009); subglacial meltwater (Esteves et al., 2017) and subsurface melting through ocean warming (Ivanovic et al., 2018; Rasmussen and Thomsen, 2004). An improved understanding of these would provide important insights into the future evolution of the WAIS (Gudlaugsson et al., 2013, 2017).*

p.2, l23-24. This is true and important, but not unique for the EIS – other ice sheets advancing during the LGM would also have destroyed older evidence.

**R1 SC1d:** *Reconstructing the evolution of glacial ice sheets prior to the LGM has been difficult, in part because, in reaching their maximum extent, the ice sheets eroded and removed nearly all older deposits. This has hampered, in particular, the reconstruction of the EIS response to past glacial abrupt climate changes.*

Please rephrase. p.2, l26. A detail but Finland would perhaps not be considered western Scandinavia, rather use just "Scandinavia".

**R1 SC1e:** *In this line, records from Norway (Mangerud et al., 2003, 2010; Olsen et al., 2002), Finland (Helmens and Engels, 2010) and Sweden (Wohlfarth, 2010) indicate rapid and rhythmic ice-sheet variations in Scandinavia,[...]*

p.2, l31. "The results" – imprecise wording; what results are you referring to?☐

**R1 SC1f:** *The resolution and quality of geophysical data across marine sectors has improved considerably in the past decade (Hughes et al. (2016) and references therein). These data confirm substantial variations of the EIS volume, with the largest uncertainties in marine sectors of the ice sheets.*

p.2, l32-33. high co-variability of the BIIS volume, extent, ice discharge? Not clear what property of the BIIS that co-vary with ocean SSTs, without looking into the underlying reference.

**R1 SC1g:** *Strong variations in the deposition of IRD suggest high co-variability of BIIS-sourced iceberg calving events with changes in ocean sea surface temperature (Hall et al., 2011; Scourse et al., 2009) and variations in EIS ice streams (Becker et al., 2017).*

p2-3, l35-1. Please specify that it is sediment cores/records that you refer to here.☐

**R1 SC1h:** *Strong millennial-scale iceberg rafting variability of the BIIS has been documented in sediment records from the North Sea (Hall et al., 2011; Peck et al., 2007; Scourse et al., 2009),[...]*

p.3, l1. ...was identified in **[records from]** the Irminger Sea...☐

**R1 SC1i:** *A dominant periodicity equal to that of DO-events was identified in sediment records from the Irminger Sea,[...]*

We have suppressed "as well" in the sentence above:

**R1 SC1j:** *Strong millennial-scale iceberg rafting variability of the BIIS has been documented in sediment records from the North Sea (Hall et al., 2011; Peck et al., 2007; Scourse et al., 2009) [...]*

p.3, l5 "just before interstadial transition" – do you mean "just before stadial-to-interstadial transitions"?

**R1 SC1k:** *For the FIS, IRD records in the Norwegian Sea show the characteristic DO periodicity, with IRD discharge occurring just before stadial-to-interstadial transitions (Lekens et al., 2006).*

p.3, l17. part of **the** LGP. (missing "the"). Also a bit vague, maybe specify which part of the LGP that was modelled in detail in this study.☐

**R1 SC1l:** *Recently, an ice-sheet model constrained by data has been used to simulate the EIS evolution throughout part of the LGP, from 37 -19 ka BP (Patton et al., 2016).*

p3, l15-19. Recent studies by Patton et al 2017 QSR and Åkesson et al 2018 QSR may also be relevant in this literature overview (cf. l23-24 and l28-30).

**R1 SC1m:** *This study was subsequently extended throughout the last deglaciation, until 8 ka BP (Patton et al. 2017).*

**R1 SC1m (and R3 SC1e):** *Notable exceptions are the recent studies by Petrini et al. (2018) and Akesson et al. (2018). The latter used a high-resolution ice-sheet model with an accurate representation of the grounding-line dynamics to study the deglaciation of the marine-based southwestern section of the Scandinavian Ice sheet; however the model domain was limited to a very small region within southwest Norway.*

Bassis et al (2017) is a very interesting paper that provides a complementary view to some of our own work (e.g. Alvarez-Solas 2010; 2011; 2013), but it refers to Heinrich events, so we have opted to not cite it here. Instead, we have included a reference to these studies in the Conclusions section:

R1 MC6 and **SC1n:** *Our results thus support the existence of a highly dynamic EIS during the LGP. They suggest an important role of oceanic melt forcing through changes in the ocean circulation in controlling the ice-stream activity. A number of studies have considered the interaction between ocean circulation changes and ice-sheet dynamics as a plausible mechanism to explain iceberg discharges from the LIS associated with H events. For example, subsurface oceanic warming during stadials in response to reduced North Atlantic deep water formation (Alvarez-Solas et al., 2010; Flückiger et al., 2006; Mignot et al., 2007; Shaffer et al., 2004) has been shown to be capable of producing large discharges from the LIS, induced by enhanced basal melting rates (Marcott et al., 2011; Alvarez-Solas et al., 2011b). The satisfactory agreement between the simulated calving and North Atlantic marine IRD records provides strong support for this mechanism (Alvarez-Solas et al., 2013), recently proposed to be modulated by isostatic adjustment (Bassis et al., 2017).*

p3, l.33. Nice overview of the paper – but what's in Section 4?

We apologise for having left this out, this should have been as follows:

 *In Section 4 the implications of our study in glacial and future climate changes are discussed. Finally, the main conclusions are summarised in Section 5.*

*Model and experimental setup*

p4, l5-6. Please mention briefly what the underlying assumptions of the SIA and SSA are. Any modeller will know this, but non-modellers might need a reminder.

**R1 SC2a:** *The underlying assumption is that, for grounded ice, the flow is dominated by bed-parallel vertical shear (i.e., shear or deformational flow) [...]. [...] Ice shelves and ice streams are described following the Shallow Shelf Approximation (SSA, MacAyeal, 1989). In such fast flowing areas, bed-parallel shear is no longer dominant; instead, longitudinal and lateral stresses become important in such a way that the horizontal velocity is independent of depth (plug flow). Both approximations are valid when the spatial scale is much smaller in the vertical direction than in the horizontal, as is the case in large-scale ice-sheet modelling*

☐p4, l7-8. Given the importance and uncertainty of basal drag on ice sheet dynamics, I think it would be helpful to briefly elaborate on how you represent basal drag and what the underlying assumptions are, e.g. type of sliding law, any non-linearity, treatment of sediments if any, spatial distribution of basal drag coefficient, if used, etc.

**R1 SC2b:** *Basal stress under ice streams ($\tau_b$) is proportional to the ice velocity $u_b$ and to the effective pressure of ice $N_{eff}$ representing the balance between ice and water pressure:*

$\tau_b = - f u_b$

*where $f = c_f N_{eff}$. Here $c_f$ is an adjustable basal friction coefficient related to the bedrock topography that accounts for the basal type of material. For comparison, absolute values up to $c_f = 7 \times 10^{-4}$ a/m were inferred by Morlighem et al. (2013) in Antarctica, with a very heterogeneous distribution, with low coefficient values in areas of fast motion dominated by sliding. $N_{eff}$ is calculated as*

$N_{eff} = rho\ g\ H - rho_w\ g\ h$

*where rho and $rho_w$ are the densities of ice and water, H is the ice-sheet thickness, and h is the hydraulic head, which corresponds to the height that would be attained by water if it were not subject to confining pressure, calculated within the basal hydrology scheme implemented by Peyaud (2006). The first term on the right hand side thus represents the pressure due to the ice load; the second one, the subglacial water pressure. At the base of the ice shelves, friction and thus basal drag, is set to zero.*

p4, l11-12. Are these arbitrary numbers or do they have some physical meaning? The criterion where you "activate" SSA could have an impact on your modelled ice velocities, grounding line retreat and ice discharge and should therefore be discussed.

The presence of water at the base of the ice sheet implies that it is not frozen to the bedrock, i.e., sliding is physically possible. More water at the base facilitates sliding more by reducing the effective pressure, and sediments also facilitate sliding because they are deformable. The criteria of 1m water thickness over sediments reduces noise in the SSA activation. The 400 m criterion over hard bedrock is a tunable parameter, but it is also effectively used to reduce noise in the velocity calculation. Setting these criteria greater than zero ensures that ice streams are activated in regions that are robustly temperate. We think that changing these values over a reasonable range (including setting them both

to zero) would not change the conclusions of the work. We have included this explanation in the model description now.

**R1 SC2c:** *Setting these thresholds ensures that ice streams are activated in regions that are robustly temperate. The presence of water at the base of the ice sheet implies that it is not frozen to the bedrock, i.e., sliding is physically possible. More water at the base facilitates sliding more by reducing the effective pressure, and sediments also facilitate sliding because they are deformable. The criteria of 1 m water thickness over sediments reduces noise in the SSA activation. The 400 m criterion over hard bedrock is a more tunable parameter, also allowing for a more numerically robust calculation of velocities within the SSA.*

p4, l14. criterium -> criterion

R1 SC2h & **R1 SC2d:** Done

p4, l14. "its thickness" slightly awkward here; use "shelf thickness" to be precise

**R1 SC2e:** Done

p4, l15. Please provide a reference for "typical thickness of observed ice-shelf fronts"

**R1 SC2f:** *This is a semiempirical parameter reflecting the fact that this is the typical thickness of ice-shelf fronts currently observed in Antarctica (Griggs and Bamber, 2011).*

p4, l13-17. Please explain briefly the rationale behind using this double criterion, as opposed to, for example, a single ice thickness criterion, or using the Levermann calving law on its own. Also, what happens if there is no shelf in the model (e.g. vertical calving face) – is the calving rate in that case zero? What happens then to the basal melt rate? Given that many vertical termini we know from the present-day are grounded in fjords several hundred meters deep, the thickness criterion would not be reached in this case. Would this have any effect on EIS evolution? (you may include part of this in the Discussion if you wish to keep the model description short)

This double criterion is standard in the GRISLI model. It was introduced after recognising that calving ice shelves, when they are thinner than the threshold used here, lead to realistic simulation of the current ice shelves of Antarctica, but this method does not allow development of new ice shelves (see e.g. Peyaud 2006; 2007). When focusing on past climates, ice sheets should be able to evolve in response to climate changes, and in particular to allow the advance of ice shelves. This is enabled through an additional criterion on ice advection leading to sufficiently thick ice at the front. Concerning the vertical termini glaciers, it has been suggested (DeConto and Pollard 2016?) that the ice front can suffer dramatic calving in these places due to the so-called cliff instability mechanism. This process is not parameterized in our model. We believe its inclusion would, if something, amplify the simulated response of the EIS to the ocean forcing. Nonetheless, the necessity of including this phenomenon in ice-sheet models has recently been contested (Edwards et al., 2019). We have included this explanation in the model description together with a discussion of its implications in the Discussion section as follows:

R1 MC3 & **R1 SC2f:** *Calving takes place at the ice-shelf front when two conditions are met. First, the ice-shelf thickness must fall below a threshold $H_{calv}$. This is a semiempirical parameter reflecting the fact that this is the typical thickness of ice-shelf fronts currently observed in Antarctica (Grigg and Bamber 2011). Second, the upstream advection must fail to maintain the ice thickness above this threshold following a semi-Lagrangian approach (Peyaud et al. 2007) to account for the fact that ice-flux divergence fosters the formation of crevasses (Levermann et al. 2012). This method is*

standard in the GRISLI model. It was introduced after recognising that a systematic cutoff of ice shelves below a given threshold led to a realistic simulation of the present-day ice shelves in Antarctica, as is the case in many models, but prevents any development of new ice shelves (Peyaud 2006; 2007). When focusing on past climates, ice sheets should be able to evolve in response to climate changes, and in particular to allow the advance of ice shelves in cold climates. To this end, before calving ice in a certain point, we test whether advection allows for the growth of ice at the front, and therefore the ice-shelf advance. $H_{calv}$ was set to 150 m, in the standard setup. To assess the sensitivity of the ice dynamics to the value of the thickness threshold $H_{calv}$, we have performed a new ensemble exploring a wide value range of this parameter's values, from 10 to 800 meters (Fig. S4). Values of this threshold above 400 m produce a drastic disintegration of the Barents-Kara complex due to its relative shallow bed. The overall effect of this sensitivity test around the preferred value is to modulate the amplitude of the response to the oceanic perturbations.

**R1 S2g:** Furthermore, *it has been suggested (Pollard et al, 2015) that the ice front can suffer dramatic calving in vertical termini glaciers due to the so-called cliff instability mechanism. This process is not parameterized in our model. We believe its inclusion would, if something, amplify the simulated response of the EIS to the ocean forcing. Nonetheless, the necessity of including this phenomenon in ice-sheet models has recently been contested (Edwards et al., 2019)*

p4, l22. I know that your focus is not Greenland so this would not affect your conclusions at all, but I don't see the advantage of using the Bamber dataset from 2001, when more recent, more accurate datasets are available (e.g. Bamber et al 2013, Morlighem et al 2014; 2017). On this note, you do include Greenland in your model domain, which I think is indeed interesting and could've been a paper on its own. However, the modelled evolution of the Greenland Ice Sheet is not mentioned in the paper, except being shown in Fig. 1 and 6 and in the supplementary animation. What's the rationale of including Greenland, when the focus of the paper is the EIS? Is there a scientific motive or just a technical reason?

Greenland is included here for technical reasons but it is actually the focus of a separate paper that is currently under review where the domain is only Greenland (Tabone et al. 2018). We recognise that a more recent and thus accurate dataset should be used when considering Greenland, and we did so in the former study by using the recent dataset by Shaffer et al. (2016) that provides a global, high-resolution data set of ice sheet topography, cavity geometry, and ocean bathymetry. We have commented on this as follows:

**R1 SC2h** & R2 SC1d : *Note there are more recent datasets for Greenland topographic features (e.g. Bamber et al 2013, Morlighem et al 2014; 2017). However, since Greenland is not the focus of our study (see Tabone et al, 2018 for a more detailed study) this does not affect our results.*

p4, l33. "inland" – would rather use "for grounded ice"

**R1 SC2i:** Done.

p5, l2. the abbreviation SMB has not been introduced yet (should be done at p4, l22)

**R1 SC2j:** Done.

**Misc. regarding the model**

Please provide the model time step, both for ice flow and PDD. A table of model and forcing parameters along with their values/ranges would be useful.

We have now included these in the model description:

**R1 SC2k:** *Finally, the ice dynamics was calculated with a 1 year timestep while thermodynamics and boundary conditions (including PDD) were updated every 5 years.*

We have also included a new table (Table 1) with the main parameters and their standard and explored values. Please see the Tables section at the bottom of this document.

You mention that GRISLI-UCM is a thermomechanical model (p4, l4), but I can't find any information of the thermal part of model. Are thermomechanical feedbacks involved over the millennial time scales you focus on?

**R1 SC2l:** *A nonlinear viscous flow law (Glen's flow law) is used with an exponent n = 3. Viscosity depends on temperature through an Arrhenius law. A traditional enhancement factor, Ef, that decreases viscosity and accelerates inland flow is used in most ice-sheet models as a tuning parameter, in order to improve the agreement between modelled and measured ice thicknesses; here Ef= 3. Further details can be found in Ritz et al. (2001). Thermomechanical coupling is extended to the ice shelves and ice streams. Ice viscosity, dependent on the temperature field, is integrated over the thickness, as in Peyaud et al. (2007).*

Is Glacial Isostatic Adjustment included in the model, and if so, how is it accounted for?

Yes, glacial isostatic adjustment is included. We have now inserted a paragraph to explain how it is accounted for:

**R1 SC2m:** *The glacial isostatic adjustment (GIA) is described by the elastic lithosphere – relaxed asthenosphere method (Le Meur and Huybrechts, 1996), for which the viscous asthenosphere responds to the ice load with a characteristic relaxation time for the lithosphere of 3000 years. For the sake of simplicity, the isostatic adjustment is assumed here to be only due to local ice mass variations, as other works have done in the past (Greve and Blatter, 2009; Helsen et al., 2013; Huybrechts, 2002; Langebroek and Nisancioglu, 2016; Stone et al., 2013).*

How is the calving rate defined (as plotted in Fig. 5) and how do you separate this from direct basal melt (also in Fig. 5)?

Calving and basal melt are two distinct processes in the model. Calving is the result of the threshold criterion together with a semi-Lagrangian diagnosis of the advection on the ice shelf described above. The grounding line position is the result of applying the flotation criterium after the mass conservation equation is solved. Basal melt is dependent on the ocean temperature anomaly applied. There is no ambiguity in the model between these two terms. We have tried to make this clear now both in the text and in the caption of Figure 5:

**R1 SC2n:** *Note that there is no ambiguity in the model between calving and basal melt, which are two distinct processes in the model. Calving is the result of the threshold criterion described above; the calving rate at a given time is thus given by the amount of ice lost to the ocean through this process by unit of time, converted to mass-water equivalent. Basal melt is dependent on the applied ocean temperature anomaly.*

**R1 SC2n:** *The calving and basal-melt rates are given by the amount of ice lost to the ocean through the calving and basal-melting parameterisation per unit of time, converted to water-equivalent volume.*

p5, l19-24. Are you here describing characteristics of the CLIMBER modelled climate in the North Atlantic, or reconstructions, or a combination? Please clarify.

Yes, these are the results of simulations with the CLIMBER-3alpha model. To make this clear below we have modified this as follows:

**R1 SC2o:** *The key differences between these climatic modes as simulated by Montoya and Levermann (2008) with the CLIMBER-3-alpha model are that in the stadial, North Atlantic Deep Water (NADW) formation is relatively weak and takes place south of Iceland…*

**R1 SC2p:** Done.

p5, l14-15. Not sure what you mean here by "purely floating ice shelves", please clarify.

We mean below the ice shelves, away from the grounding line. We have reformulated this as follows:

**R1 SC2q:** *[...] it accounts separately for basal melting below the ice shelves (away from the grounding line) and at the grounding line. The basal melting rate of the ice shelves is given by [...]*

p6, l15-19. You assume 10 times lower melt for "purely floating ice shelves" than at the grounding zone (what do you consider as the "grounding zone"?) and justify this with qualitative agreement with "some Greenland glaciers". OK, but given your previously claimed analogue between the Bjørnøyrenna basin (where most of the action in your model happens) and the WAIS, would it be more appropriate to compare your imposed melt rates with Antarctic melt rates? Also gamma = 0.1 seems a bit arbitrary as it stands; did you explore any other values for gamma and found 0.1 to be the "best" one, or did you settle on this directly based on present-day observations in the studies you cite? Note also that the cited Rignot and Jacobs (2002) covers basal melt in Antarctica, and not Greenland. In addition, the studies cited are for Greenland glaciers with floating tongues (Petermann and 79N), which is indeed more relevant than if you were referring to glaciers in Greenland with grounded termini; this should be mentioned.

The value of 0.1 for the gamma coefficient was set based on present-day observations. Rignot and Jacobs (2002) indicated that melt rates near glacier grounding lines exceed the area average rates for the largest ice shelves by 1 to 2 orders of magnitude. Marsh et al. (2015) found a difference in one order of magnitude for the Ross Ice Shelf. Münchow et al., 2014 found values between 1-15 m/a in the Petermann glacier. Wilson et al., 2017 found basal melting values ranging from roughly 0 to 30-50 m/a in the 79 North Glacier, Ryder Glacier, and Petermann Glacier. We explored some other values of gamma, although not systematically. Changing these values to higher ones results in a reduction of the ice-shelf extension but it does not affect the general behavior of the marine basins since these react to changes at the grounding line. Smaller values than the mentioned 0.1 can favor the the occurrence of an unrealistic surface of the ice shelves. The reviewer is right concerning the citations and we are grateful for pointing this out. We have now rephrased the former paragraph as follows:

**R1 SC2r & R3 SC2o:** *Thus, we consider that the submarine melting rate for ice shelves is 10 times lower than that close to the grounding zone, which is in qualitative agreement with observations in some Greenland glaciers with floating tongues (Münchow et al., 2014; Wilson et al., 2017) as well as in Antarctic ice shelves (Rignot and Jacobs, 2002). Note that this value is subject to uncertainty. Although we did not explore any other values different from gamma = 0.1, we did consider a range of*

*kappa values between 1-10 m/a/K which accounts for a wide range of oceanic sensitivities (see section 2.3).*

p7, l14. Great that you're comparing with ice sheet reconstructions. I know that you're not trying to fit the model perfectly to reconstructions but rather to investigate the relative roles of forcings. Personally, I think that aggressive tuning of climate and model parameters to (over)fit the data perfectly will weaken the value of this kind of study, so I applaud you for not going too much down this route. Still, for transparency and to assess your slightly vague "to an extent that satisfactorily agrees with previous reconstructions", I think including a figure comparing with one or two ice sheet reconstructions (e.g. DATED-1 and ICE-5G) would be valuable. Perhaps you could add these reconstructed ice sheet margins in Fig. 1, or if this becomes too messy, add another figure.

**R1 SC2s:** Indeed this comparison was made in a previous paper where we focused on the methodology and we performed a detailed analysis of the ice sheet characteristics and their fit with reconstructions. Since the focus of the current paper is on the mechanisms producing millennial-scale variability, we believe it is not necessary to show this comparison again here (Note the current version of the manuscript has already been enlarged to 40 pages).

p8, l5. applying (missing p)

**R1 SC2t:** Done**.**

p8, l10. You give a nice overview of your experiments. Would also be valuable with a table summarizing the experiments and their differences for easy reference (control run, constant vs time-varying atm forcing, surface vs subsurface ocean, sea level etc.)

**R1 SC2u:** Done. We have also included an additional table (Table 2) summarizing the experiments. Please see the Tables section at the bottom of this document.

p8, l13. Please specify that it is refreezing under ice shelves you're talking about here, since you do include refreezing in your SMB model.

**R1 SC2v:** Done.

*Results*

First off, I think this section would benefit from division into subsections.

**R1 SC3w.** Done; this section has now been split into three subsections: Ice volume evolution, Mass-balance response and Grounding-line dynamics.

p8, l22. "internal ice-sheet variability" – what is exactly in the ice sheet causes this internal variability?

We have expanded on this and added the following to the manuscript:

**R1 SC3a:** *"a lower frequency SLE fluctuation is found as a result of internal ice-sheet variability (Figure 4) through a thermomechanical feedback. This slow variability appears only in the southernmost parts of the Eurasian ice sheet where ablation exists. It is due to an interplay between the available basal water favoring sliding and the EIS associated thinning due to an increase in*

*velocities. Since this phenomenon concerns only the ablative borders of the ice sheet and its frequency corresponds to more than 20 kyr, its governing dynamics is not detailed here.* "

p8, l23. "slight response" – please more specific, how many % variability or ice volume/sea level equivalent? Is this subdued response to sea level forcing what you expect, or surprising (you may link this to previous literature in the discussion)? Do you think your coarse model resolution dampens the response, making it "harder" for grounding lines to retreat, but once they retreat, the response is large since you "instantaneously" remove a big 40x40 km chunk of ice? Or is it something inherent to the sea level forcing? Is the subdued response to sea level the same everywhere, or does sea level forcing induce grounding line retreat in some sectors, related to the particular ice sheet configuration (e.g. deep vs shallow grounding lines)?

We have modified the mention to a "slight response" to its relative percentage w.r.t the ALLsrf simulation which is approximately 0.5 m in s.l.e. Since this response is of a significantly smaller amplitude, we did not carry out a detailed analysis of the response. However, it is noticeable that the part of the ice sheet reacting to sea-level variations corresponds to the Barents-Kara region. It is hard to have a clear picture of what to expect here, given this is a sensitivity study with no clear analogues in the literature. Therefore, we believe quantifying this particular result could be too speculative and unnecessary for the current manuscript.

**R1 SC3b:** When the model is forced only by changes in sea level (SL run), a small response of approximately 0.5 m SLE is observed on millennial-scales.

p8, l30. This is an exciting result. The anti-phase relationship is not perfectly in phase throughout the LGP, which perhaps should be mentioned. Given that your SST and subsurface anomalies (Fig. 2cd) are of opposite sign, though not with same spatial distribution, I don't think it's too surprising that the ice sheet responds in this anti-phase manner. Still I do think it's in interesting result with relevance both for abrupt climate change during the LGP and for present-day/future, but it requires a more thorough discussion. See also major comment above on ocean forcing.

We agree and have expanded accordingly:

**R1 SC3c & R1 SC5a:** *OCNsub shows an anti-phase relationship with respect to OCNsrf, with the largest reductions in ice volume occurring during prolonged stadial periods and regrowth during interstadials. This behavior can be explained by the fact that ocean waters at the subsurface warm (cool) during episodes of reduced (enhanced) convection at the Nordic Seas as a result of variations in the AMOC strength (Figure 3d-e). Note that the anti-phase relationship is, however, not perfect. At the surface, the largest anomalies are found off the North Atlantic, the British Isles and the Norwegian coast, and result from the intensification of Atlantic northward heat transport associated to the enhanced AMOC during interstadials; at the subsurface the concomitant cooling is largest in the Nordic Seas as a result of enhanced heat loss to the atmosphere associated with enhanced convection. Thus, the out-of-phase relationship found in the dynamic response of the EIS between these two oceanic experiments results from the opposed sign of their spatial forcing patterns (Figure 3). When considering the forcing at the subsurface of the ocean together with the atmosphere (ALLsub), slight reductions of the EIS volume (less than 1 m of s.l.e) during interstadials are superimposed onto the previous behavior (Figure 4).*

p9, l5-13. Please check the manuscript to be consistent with the use of yr-1 and a-1 (as used at p7, l11).

**R1 SC3d:** Done. We now consistently use $a^{-1}$ everywhere.

p9, l20. mid panel -> b

**R1 SC3e:** Done.

p9, l20-35. A very interesting and nice paragraph where you break down EIS change into sectors and try to explain why. I think an additional figure (if feasible) showing ice volume through time for the different sectors you refer to (e.g. SW vs NE) for one or two forcings (for example ALLsub and ALLsrf), would be of great interest and also illustrate the spatial contrasts and their relation to the forcing you outline.

**R1 SC3f:** We have followed the suggestion of the referee and included a new figure showing the evolution of the three main sectors of the EIS for ALLsub and ALLsrf.

p10, l23. ...are representative of [**the ice sheet response**] during all other stadial-to- interstadial transitions.

**R1 SC3g**: Done.

p11, l8-12. Great that you're trying to quantify the grounding line retreat, I think this analysis strengthens the paper. Firstly, over what "fixed area" (line 11) do you define mikro? Is it the square highlighting the Bjørnøyrenna basin shown in Fig. 1c? Secondly, your definition of mikro appears to represent the percentage of non-grounded grid points in the Bjørnøyrenna basin, so that increasing mikro (more non-grounded grid points) corresponds to grounding line retreat. While there is nothing formally wrong with this definition, I wonder if it would be clearer to just use the grounded ice sheet area as your metric for grounding line retreat. Grounded ice area could be shown in Fig. 9 on two different y-axis, one in (%) and one in (km2). See also comments below on Fig 9.

First, mu ("micro") is indeed defined over the square highlighting the Bjørnøyrenna basin shown in Fig. 2 (old Figure 1c). Secondly, using the grounded ice sheet area as your metric for grounding line retreat would be possible, but we have thought about different possible metrics and we finally concluded that an adimensional metric as the one used here would be preferable; we have actually used the same metric in other studies in different domains such as Antarctica (Blasco et al. 2018). We have explained both issues as follows:

**R1 SC3h:** *The migration of the grounding line through time has been characterized by means of an index ($\mu$) that weighs the proportion of non-grounded points in the region of the Bjørnøyrenna basin, defined over the black square highlighting the Bjørnøyrenna basin shown in Fig. 1c. Note that other metrics are also possible; the same metric has been used in other studies in different domains such as Antarctica* (Blasco et al. 2018).

p11, l18-19. I think this an interesting point. For your experiment OCNsrf, you've found a quite close relation between ice thickness H and the number of non-grounded points in the Bjørnøyrenna basin (right panel in Fig. 9). Is this the same as saying that the grounded area and ice thickness in this basin scales linearly? I.e. that the more extensive grounding line retreat (higher mikro), the thinner ice sheet (lower H)? And conversely, a thickening ice sheet translates linearly into grounding line advance? Is this what we expect? Does this mean that ice sheet thinning and grounding line retreat occurs more or less at the same rate, i.e. are tightly coupled? There is also an "anomalous" branch of your H vs mikro plot, where grounding line retreat and thickness temporarily are decoupled. What stage of ice sheet change is this (stadial or interstadial)? What occurs first, grounding line retreat or thinning? Is this what you expect, or counterintuitive? Just adding a brief discussion on this would be relevant both for both paleo-ice sheet changes and people working with present-day changes in Greenland and

Antarctica. A related line or two about why v and mikro do not follow such close relationship would also improve the manuscript.

Coming from a stadial state and an advanced grounding line position, the ocean warms facilitating a local negative mass balance at the grounding line. This thins the grounding line locally triggering its retreat and starting the propagation of the dynamic imbalance of the ice stream. The propagation of the surface slope changes happens almost instantaneously at these time scales (with a typical propagation speed of about 10 km a^(-1)). This chain of processes explains the tightened linear relationship between the Bjørnøyrenna basin thickness and the grounding line position ("mu"). Although a grounding line retreat (advance) of the grounding line in this region produces an acceleration (deceleration) of the ice streams, its linear relationship is less obvious than regarding ice thickness. This is partially explained by the fact that ice-stream velocities lag the grounding line imbalance due to the characteristic time for the kinematic wave to propagate along the ice streams of the whole basin (typically of ~1 km a^(-1)).

We have added the following:

**R1 SC3i:** *A local thinning of the grounding line produced by a warmer ocean triggers its retreat and starts the propagation of the dynamic imbalance of the ice stream. The propagation of a change in the surface slope happens almost instantaneously at these time scales (with a typical propagation speed of about 10 km a^(-1)). This chain of processes explains the tightened linear relationship between the Bjørnøyrenna basin thickness and the grounding line position ("mu"). Although a grounding line retreat (advance) of the grounding line in this region produces an acceleration (deceleration) of the ice streams, its linear relationship is less obvious than regarding ice thickness. This is explained by the fact that ice-stream velocities lag the grounding line imbalance due to the characteristic time for the kinematic wave to propagate along the ice streams of the whole basin (typically of ~1 km a^(-1)).*

**Discussion**

p12, l2. "some authors" - need reference

**R1 SC4a:** *Some authors (e.g. Gudlaugsson et al. (2013)) [...]*

p12. l2-3. I would like to congratulate the authors by making the link between the EIS during the LGP and the present-day/future of contemporary ice sheets. However, it's not entirely clear to me from this paragraph whether the authors' findings support or contradict the Kara-Barents complex as a "WAIS analogue". Here I think the relevance of the EIS for present/future changes of Greenland/Antarctica could be strengthened.

We have included a discussion on the role of oceanic temperature changes in the present and future as well for Greenland and Antarctica in the Discussion:

**R1 SC4b:** *Our results have implications not only for the study of past glacial abrupt climate changes, but also currently ongoing and future climate change. In Greenland, warmer North Atlantic waters penetrating into Greenland's fjords are currently thought to contribute to the recently enhanced discharge of ice into the ocean (e.g. Straneo and Heimbach, 2013). Warmer ocean temperatures enhance submarine melting at the calving front of tidewater glaciers, contributing to accelerate them, increasing the discharge of ice mass into the ocean and potentially leading to a retreat of their grounding lines. This mechanism has been observed in several of Greenland's marine-terminating glaciers (e.g. Hill et al. 2017; Wood et al. 2018). In Antarctica, the WAIS is losing mass at an*

*accelerated rate as a consequence of the enhanced submarine melting of floating ice shelves and calving processes at the ice front (Paolo et al., 2015; Rignot et al., 2013). The most rapid thinning and mass loss has occurred in the ice shelves of the Amundsen and Bellingshausen seas, in regions where Antarctic Continental Shelf Bottom Water have warmed through the intrusion of Circumpolar Deep Water onto the Amundsen and Bellingshausen Seas continental shelves (Schmidtko et al. 2014). Under future climate change, many climate models project a weakening of the AMOC and a regional cooling or minimum atmospheric warming around Greenland during the 21st century that constitutes a negative feedback that could reduce melting of the GrIS in a warming climate. However, a maximum in warming has also been found to occur in the subsurface ocean layer around Greenland as a consequence of AMOC reorganisations that could induce a year-round melting of polar ice sheets (Yin et al. 2011). Projections indeed indicate enhanced subsurface warming will lead to enhanced submarine melt rates of Greenland's outlet glaciers (e.g. Nick et al. 2013; Peano et al. 2016; Calov et al. 2018), even though models do not generally account for the dynamic response of these glaciers. In Antarctica, although processes that regulate ocean heat transport to the sub-ice-shelf cavities and their sensitivity to changes in forcing need to be understood (Rintoul et al. 2018), climate projections indicate that changes in stratification of the water column will enhance the intrusion of CDW in Antarctic ice-shelf cavities, and thereby submarine melting (Naughten et al. 2018). This mechanism is also found in a coupled climate model including an eddying ocean component (Goddard et al. 2017). Thus, changes in ocean water temperatures appear to be key in driving ice-sheet changes both in the past and in the future.*

p12, l23. grounding line**s**

**R1 SC4c:** Done**.**

p12, l23-31. A very important paragraph where the authors outline uncertainties associated with linking calving (flux) and IRD. These uncertainties are outlined nicely, but presently they are not discussed in light of the findings in this study. I also feel that this paragraph would benefit from one or two additional references.

**R1 SC4d:** We believe our Discussion section now deals with all the uncertainties in the literature concerning model-data comparison, including those regarding the implicit assumptions when interpreting IRDs and calving from ice sheets. (Please see the expansion on the Discussion section)

p13, l4. regarding initial ice sheet size – how does your initial ice sheet state entering MIS3 affect subsequent evolution? I don't expect any new simulations in this regard but a brief comment what you expect, particularly since you tuned your basal melt rates at 40 ka to obtain an ice sheet in reasonable agreement with reconstructions.

The initial ice-sheet size could indeed affect the ice-sheet response. Since the response to the ocean has been found to be dominant, a larger ice sheet, with more developed ice shelves and thus more exposed to the ocean would be prone to suffer stronger basal melting (since refreezing under ice shelves is not allowed); destabilisation of ice shelves could therefore result in a more dynamic ice sheet with larger calving peaks. A smaller ice sheet would therefore only be affected by atmospheric forcing.  We have thus included the following in the Discussion section:

**R1 SC4e:** *Also, our results depend somewhat on the particular SAT and oceanic temperature anomaly patterns simulated by our climate model, the magnitudes of the resulting forcing, and the initial size of the simulated EIS. Since the response to the ocean has been found to be dominant, a larger ice sheet, with more developed ice shelves and thus more exposed to the ocean would be prone to suffer stronger basal melting; destabilisation of ice shelves could therefore result in a more*

*dynamic ice sheet with larger calving peaks. A smaller ice sheet would therefore only be affected by atmospheric forcing.*

p13, l14. Rignot et al. 2002 -> Rignot and Jacobs, 2002. See also comment above (Section 2.2) on justifying your magnitudes of basal melt against data from Antarctica vs Greenland.

**R1 SC4f:** Done.

*Conclusions*

Well written. Consider including your finding about surface vs subsurface ocean. A brief statement on uncertainties in ice sheet dynamics/grounding line dynamics could also be included. I think you may also mention that you explicitly include calving in your model, and very briefly how oceanic basal melt is parameterized.

We have now extended in both issues as follows, first for the surface versus subsurface temperature issue, then for the uncertainties in ice sheet dynamics/grounding line dynamics and finally to include a brief explanation on calving and basal melt:

**R1 SC5a:** *An out-of-phase relationship is found in the dynamic response of the EIS when forcing the ice-sheet model with the millennial-scale simulated surface and subsurface temperature anomalies. This behaviour results from the roughly opposite sign of their spatial forcing patterns in the Nordic Seas. This pattern has been found to be robust in a number of models but its details could well be model dependent, and, in particular, dependent on the precise location of the convection sites affected (e.g. Brady and Otto-Bliesner, 2011; Mignot et al., 2007; Montoya and Levermann, 2008; Shaffer et al., 2004; Flückiger et al., 2006).*

**R1 SC5a:** *Since the ocean plays a major role during abrupt ice sheet changes, the model's treatment of grounding line dynamics is a key issue. Finally, this represents one of the first attempts to simulate both oceanic and atmospheric impacts on ice sheets associated to abrupt climate changes. Investigating this issue further with higher resolution in and exploring the effect of the underlying uncertainties in ice-sheet and grounding line dynamics is of uttermost interest and in the scope of future work.*

**R1 SC5a:** *The model includes as well an explicit grounding-line treatment, a simple basal melting parameterisation that depends linearly of the ocean temperature anomalies and calving through a double criterion on ice thickness and advection at the ice front.*

*Misc.*

• check consistency of Bjørnøyrenna vs Bjørnøyrenn throughout text and figure captions.

**R1 SC5b:** Done.

**Figures**

Generally nice and clear figures. Some panels within the figures are missing abcd labels (Fig. 2, 7, 8, 9). To help the reader, make sure you make according changes in places within the text where you refer to different panels of these figures.

**R1 SC Figures:** Done. Please see the New Figures section at the end of this document.

*Figure 6.* previous -> prior.

**R1 SC Figure 6:** Done.

*Figure 7b.* ice velocities in the Bjørnøyrenna basin – how are these defined? Mean velocities over the entire basin? (same in Fig 8b)

Ice velocities in the Bjørnøyrenna basin are indeed defined as mean values over the entire basin. This is now explained in the the figure captions:

**R1 SC Figure 7-8b:** Done: *Temporal component of the millennial-scale climatic forcing (beta index), ice velocities in the Bjørnøyrenna basin, calculated as mean values over the entire basin [...]*

*Figure 7c.* I like that you plot the calving rate in (Sv) for oceanographic relevance – also consider adding a second axis in mass loss per year (Gt/a) for the glaciologists reading this. (same in 8c)

**R1 SC Figure 7-8c:** We have tried to add a second axis but we were not satisfied with the aesthetics of the new figure. Thus, besides, adding the missing abcd labels we decided to keep it as it was.

*Figure 7d.* ice shelf extension – would rather use "ice shelf area" to emphasize you're showing area, not length. Check in text to be consistent. (same in Fig 8d)

**R1 SC Figure 7-8d:** Done.

*Figure 9.* A nicely plotted interesting figure. I would put "grounding line index \mikro (%)" as ylabel instead of just mikro (%) to help the reader, unless you follow my suggestion above to use the grounded area as a metric instead. In the caption, please also cross-reference where in the text the index mikro(t) is defined (Eq. 18). For ice thickness H, is this the mean ice thickness in the square shown in Fig. 1c? Ice stream velocities v, over what region are they defined? Finally, I would label this figure with abc, to more clearly refer to each panel in the text (e.g. p11, l13-21).

**R1 SC Figure 9:** Yes, the quantities correspond to the rectangle now illustrated in new Figure 2. Labeling done. Please see the New Figures section at the end of this document.

**Supplementary**

*Fig. S1 and S2.* Though it should be obvious to most readers, please spell out "S.I." in the yaxis label, as you've done in Fig. 3.

*Animation.* Should the units of time in the animation be changed ka -> a?

Yes, absolutely. Thank you for pointing this out.

Also, unless I'm misinterpreting something, the model seems completely off when it comes to getting rid of ice in the Holocene (see screen dump from your animations below). You're modelling the evolution all the way to the present-day but northern Europe is still under ice in your model at 0.0 ka BP, so is northern Russia. Do you have an idea why? I know this is not the period you focus on, but people seeing the animation may take this large disagreement as a sign of something completely missing in your model. Given the severe mismatch, I think an explanation should be included in the manuscript.

This has a clear explanation: we are not performing a transient simulation throughout the last glacial period and up to the present day, but investigating the effect of millennial-scale variability alone in a glacial background values (in practice this was done by fixing the orbital alpha index to its value at 40 ka BP). In this sense this is rather a sensitivity study. It means without the orbital variation of climate (provided by a time-varying alpha) deglaciation is simply not achieved. In order to avoid this potential confusion we have limited the animations now from 80 to 25 ka BP, also to allow focusing on the mos interesting part of the simulations.

**References**

Åkesson, H., Morlighem, M., Nisancioglu, K. H., Svendsen, J. I., & Mangerud, J. (2018). Atmosphere-driven ice sheet mass loss paced by topography: Insights from modelling the south-western Scandinavian Ice Sheet. *Quaternary Science Reviews*, *195*, 32-47.

Banderas, R., Alvarez-Solas, J., Robinson, A., & Montoya, M. (2018). A new approach for simulating the paleo-evolution of the Northern Hemisphere ice sheets. *Geoscientific Model Development*, *11*(6), 2299-2314.

Bamber, J. L., Layberry, R. L., & Gogineni, S. P. (2001). A new ice thickness and bed data set for the Greenland ice sheet: 1. Measurement, data reduction, and errors. *Journal of Geophysical Research: Atmospheres*, *106*(D24), 33773-33780.

Bamber, J. L., Griggs, J. A., Hurkmans, R. T. W. L., Dowdeswell, J. A., Gogineni, S. P., Howat, I., ... & Steinhage, D. (2013). A new bed elevation dataset for Greenland. *The Cryosphere*, *7*(2), 499-510.

Gladstone, R. M., Warner, R. C., Galton-Fenzi, B. K., Gagliardini, O., Zwinger, T., & Greve, R. (2017). Marine ice sheet model performance depends on basal sliding physics and sub- shelf melting. *The Cryosphere*, *11*, 319-329.

Morlighem, M., Rignot, E., Mouginot, J., Seroussi, H., & Larour, E. (2014). Deeply incised submarine glacial valleys beneath the Greenland ice sheet. *Nature Geoscience*, *7*(6), ngeo2167.

Morlighem, M., Williams, C. N., Rignot, E., An, L., Arndt, J. E., Bamber, J. L., ... & Fenty, I. (2017). BedMachine v3: Complete bed topography and ocean bathymetry mapping of Greenland from multibeam echo sounding combined with mass conservation. *Geophysical research letters*, *44*(21).

Patton, H., Hubbard, A., Andreassen, K., Auriac, A., Whitehouse, P. L., Stroeven, A. P., ... & Hall, A. M. (2017). Deglaciation of the Eurasian ice sheet complex. *Quaternary Science Reviews*, *169*, 148-172.

Robinson, A., & Goelzer, H. (2014). The importance of insolation changes for paleo ice sheet modeling. *The Cryosphere*, *8*(4), 1419-1428.

**Anonymous Referee #2**

Alvarez-Solas et al. investigate the millennial scale variability of the Eurasian ice sheet during the last glacial period. They use an ice sheet model forced offline by a combination of two glacial climatic snapshots, stadial and interstadial. The relative importance of the two snapshots is weighed by an index constructed from a Greenland temperature reconstruction. In their model framework, Alvarez-Solas et al. show that oceanic perturbations induce much greater ice volume changes

compared to atmospheric perturbations. They discuss their ice volume variations with respect to IRD layers in marine sediments.

The paper tackles definitively very interesting questions regarding the role of the ocean C1 in the (in)stability of large marine ice sheets. Little has been done with this respect on the Eurasian ice sheet while a fair amount of geological constraints exist. I think the paper is well written and generally nicely illustrated but I have a few important comments that I would like to see addressed.

We are grateful to the reviewer for their careful reading of the manuscript and the many suggestions made which undoubtedly have contributed to make the results more clear. We have attempted to address all concerns. Below we give a detailed response to each of the comments raised.

**General comments**

- Basal melting rate and ice volume. I am very happy to see that the authors have chosen to change their basal melting rate formulation compared to their previously submitted version of the manuscript (doi: 10.5194/cp-2017-143) so they no longer use a negative sub-shelf melting rate (ice accretion). However I am surprised that the change in setup, and subsequent change in results, does not relate to any change in conclusion nor discussion. In the previous version of the manuscript, during the transient simulation, the ice volume was oscillating around the 40k spun-up ice volume. In the new version, the ice volume is now perpetually decreasing from 110k to 10k when using the oceanic forcing with kappa>1. As far as I understand your methodology, we expect the 40k ice sheet to be representative of a mean state of the MIS3 ice sheet and your millenial scale index should translate into waxing and waning of the ice sheet around the mean state. The fact that you have a negative trend in ice volume suggests that the model is unable to regrow ice after the imposed oceanic perturbation. I understand that is a complicated issue that cannot be resolved with such a simple index perturbation. However, it seems to me that it is not straightforward to draw robust conclusions on the physical mechanism for MIS3 ice volume oscillations when the model is currently unable to simulate an Eurasian ice sheet that survive to these oscillations. I might be missing something but I think this issue should be clarified and clearly discussed in the paper. As a side note: I could not find the volume your 40k spun-up ice sheet. This is needed to interpret the importance of the trend (8 to 12 m sle!).

Indeed, we have now impeded refreezing by construction. We note, however, that reviewer 3 has now criticised this because they say this does not consider possible refreezing associated with supercooling, which seems to us to imply that the occurrence of refreezing is controversial. It is true that we have failed to discuss the negative trend in ice volume. We now discuss this issue and also give the initial ice-sheet volume in order to interpret the importance of the trend.

It is important to note, however, that waxing and waning around a mean state is not necessarily a robust expected result given our experimental design. Advancing and retreating ice sheets are known to suffer hysteresis. Thus, without any refreezing, grounding line retreat is expected to happen abruptly while re-advancing the grounding line to its former position takes much longer (see for example the results of the Úa finite element ice sheet model: https://sway.office.com/A3ihHbhXG59GwsYf -  in particular, the animation of the MISMIP experiment 3A nicely illustrates the mentioned transient hysteresis).

Summarizing this aspect: It is not "that the model is unable to regrow ice after the imposed oceanic perturbation", but that it does not have the time to do so between two oceanic perturbations. It is also important to stress that in the real world, millennial time-scale climate changes occur in a context of globally decreasing temperatures towards the LGM and that these can be counteracting the

mentioned ice-volume decline of our simulations. Because the focus of this paper is precisely on millennial timescales, this aspect is not explicitly addressed here.

**R2 MC1** & MC3**:**

As a consequence of the millennial-scale forcing, a  trend in ice volume from its initial value of 8.3 x 10^(15) km^3 (about 21 m SLE)  leading to a loss of 8-12 m SLE is found. This is a consequence of the fact that no refreezing is allowed and that a positive constant (and spatially uniform) basal melting of 0.1 m/a was imposed. As a consequence, accumulation is not able to compensate for ice loss through basal melt and calving after each ice-mass loss event. Note, however, that background conditions are fixed at 40 ka BP; in a more realistic setup, as time proceeds forward, orbital forcing leading to gradually colder conditions would be expected to aid in the ice regrowth, thereby helping to its growth throughout the LGP.

- On the method, 1. Because CLIMBER3-α underestimate the stadial to interstadial temperature change at NGRIP, beta* in the paper has been scaled to match the recorded amplitude. One can wonder if this scaling is appropriate for oceanic fields. C2 In the atmosphere the millenial anomaly simulated by CLIMBER at NGRIP is about 5-6 degrees, this is why you have roughly a beta that oscillates between -1.5 and 1.5 (amplitude 3) to reproduce a stadial to interstadial of about 15 degrees. In the ocean, CLIMBER also simulates SST anomalies of about 5-6 degrees around the British Isles, meaning that your oceanic temperature during certain DO events can increase by more than 15 degrees. Is this supported by any SST record? This makes me wonder about your experimental design that puts a critical weigh on the ocean. . .

Our experimental setup was conceived with the idea that the CLIMBER-3a model underestimates the SAT response in Greenland, and this was corrected for by scaling the simulated SAT anomalies. The same approach was indeed used for the ocean temperatures. It is conceivable that our synthetic temperature forcing is larger than that deduced from reconstructions, which range from 4-10 K (Dokken et al. 2013; Martrat et al. 2004; Rasmussen et al. 2016). However, the possible uncertainty in the temperature forcing is subsumed in the kappa index, which in our case varies between 1-10 m/K/a. These values encompass a broad range, which is at the lower end of present-day estimates in Antarctica (Rignot and Jacobs 2002). Thus, our forcing should not be strongly overestimating the past oceanic forcing. We have clarified this in detail as follows in the Discussion:

**R2 MC2:** Note the temporal index used is the same for the atmosphere and the ocean and the amplitude is given by an OGCM simulation of two different oceanic states mimicking stadial and interstadial periods. We then translate those fields into ablation (through PDD, whose uncertainty has been extensively explored) and into basal melting (through a linear equation). It is conceivable that in certain locations our synthetic oceanic temperature forcing is larger than that deduced from reconstructions, which range from 4-10 K (Dokken et al. 2013; Martrat et al. 2004; Rasmussen et al. 2016). However, the possible uncertainty in the temperature forcing is subsumed in the κ index which in our case varies between 1-10 m/a/K. These values are in the range of (or even below in most cases) those suggested by data in Antarctica (Rignot and Jacobs, 2002). Note, in particular, that from mid values of κ of 5 m/a/K the response to the ocean is already of greater amplitude than that to the atmosphere, making our main conclusions robust. Thus, our forcing should not be strongly overestimating the past oceanic forcing.

- On the method, 2. Your base value for sub-shelf basal melting rate is 0.1 m/yr. Since you have a linear basal melting rate perturbation (Eq. 14), given your oceanic anomalies and a Kappa at 5 m/K/yr, for negative values of Beta (roughly half the time) you end up with B(t) <0 (i.e. B(t) imposed to 0). Your perturbation is then mostly going towards one direction (more melt). This might explain why you have this negative trend in ice volume in OCN experiments. I think this base value of 0.1 m/yr play an

important role in your model setup but is not convincingly justified nor discussed. Also, why this parameter has be spatially homogeneous? Without knowing the actual value, we can expect very different sub-shelf basal melting rates in the Kara area compared to the British Isles area.

We agree that the constant background Bgl value of 0.1 m/a probably plays a role in the negative trend. And indeed, spatially non-uniform background melting is also conceivable. However, we have absolutely no information on what this background value would have been. Because our focus was the response of the EIS to millennial-scale climatic variability, we opted for the simplest experimental setup possible, meaning a spatially uniform and fix in time background value perturbed by a millennial-scale index. We have expanded on this as follows:

**R2** MC1 **& MC3:** As a consequence of the millennial-scale forcing, a  trend in ice volume from its initial value of 8.3 x 10^(15) km^3 (about 21 m SLE)  leading to a loss of 8-12 m SLE is found.  This is a consequence of the fact that no refreezing is allowed and that a positive constant (and spatially uniform) basal melting of 0.1 m/a was imposed, As a consequence accumulation is not able to compensate for ice loss through basal melt and calving after each ice-mass loss event. Note, however, that background conditions are fixed at 40 ka BP; in a more realistic setup, as time proceeds forward, orbital forcing leading to gradually colder conditions would be expected to aid in the ice regrowth, thereby helping to its growth throughout the LGP. Spatially non-uniform background melting is also conceivable. However, we have no information on what this background value would have been. Because our focus was the response of the EIS to millennial-scale climatic variability, we opted for the simplest experimental setup possible, meaning a spatially uniform and fixed-in-time background value perturbed by a millennial-scale index.

- Figure missing. It is hard to have a clear picture of what the actual forcing looks like as there is an important piece of information missing. I strongly suggest you to add an additional figure right after Fig. 2 in which you show the SMB and oceanic perturbations for a typical DO event (e.g. beta* from 1.5 to -1.5). I understand that there is a geometry feedback and that beta* is not constant but you can easily take your spun-up 40k ice sheet and show deltaSMB=SMB(beta=1.5)-SMB(beta=-1.5) (along with the equilibrium line in the stadial). And the same for Bmelt. This is a way to show the forcing that the ice sheet model is experiencing. I would ideally like to see the same kind of anomaly for the SAT, SST and sub-surface temperature.

**R2 MC4:** We here disagree with the referee in the sense that it should not be hard to have a clear picture of the forcings, given we clearly provide them in figures 1 and 3.  To make this even more clear we have now added the summer temperature fields (both MIS3 absolute and stadial-to-interstadial anomalies in new Figures 1 and 3 respectively).

**Specific comments**

P1 L18-20 This is a strong assertion which seems overconfident to me based on the limitations of the experimental design. Please remove.

Done. We have replaced that sentence by the following:

**R2 SC Abstract:** *Our results clearly show the capability of the EIS to react to glacial abrupt climate changes, and highlight the need for stronger constraints on the ice sheet's glacial dynamics and climate-ocean interactions.*

Introduction

P2 L2 Do you mean BKIS?

**R2 SC1a:** Indeed, we have corrected this.

P2 L19-21 This is arguable. Climatically speaking, the two ice sheets are in a very different context (latitude, AMOC, storm tracks...)

We understand the criticism. However this is a claim that has been found in the literature. We thus have now attempted to provide an explanation of this claim, also in order to include reviewer 1's suggestion SC1b:

**R2 SC1b:** *The BKIS, in particular, was predominantly marine-based for much of the LGP. For this reason, and because it had a similar size as the West Antarctic ice sheet (WAIS) during the LGM (Anderson et al., 2002; Bentley et al., 2014; Denton and Hughes, 2002; Evans et al., 2006; Hillenbrand et al., 2012; Svendsen et al., 2004; Whitehouse et al., 2012), it is sometimes considered as a geological analog of the current WAIS (e.g. Gudlaugsson et al., 2013).*

P2 L31-32 No direct evidence for ice volume but ice extent.

**R2 SC1c:** Indeed, we have replaced *volume* by *extent*.

P3 L1-2 Since the Greenland ice sheet is included in your geographical domain, is this also reproduced in your simulations?

R1 SC2h & **R2 SC1d:** This point was also raised by reviewer 1. Greenland is included here for technical reasons but it is actually the focus of a separate paper that is currently under review where the domain was only Greenland (Tabone et al. 2018).

2. Model and experimental setup

P4 L2-3 Perhaps you could include a section in the discussion on the limitation of the floatation criteria on a 40km grid resolution, as this is thought to be inaccurate to compute grounding line migration. Do you think you would have different grounding line migration sensitivities with a much higher resolution at the grounding line or with a analytical flux at the grounding line?

This is an important point raised by all three reviewers. We realise that with the coarse resolution it is not possible to accurately resolve the grounding-line dynamics. However at this stage it is not feasible to use the high resolutions required, and alternatives Thus our results should be considered as a first step toward investigating this problem and revisited in the future with higher-resolution models when this is feasible. We have attempted to acknowledge this in the Discussion section:

R1 MC1, R1MC7 & **R2 SC2e** & R3 SC4b: *Given the conclusion that the ocean plays a major role in abrupt ice sheet changes, the model's treatment of grounding-line dynamics is a key issue. Several studies have shown that for many applications, a resolution of around 1 km is needed to accurately determine the grounding-line position. In addition, it has been shown (e.g. Gladstone et al 2017) that the grounding-line behaviour is sensitive to the choice of friction law and the physics of submarine melting, and that these determine model-resolution requirements. In our case, the dependence of*

*basal drag on effective pressure allows for the desirable property of basal drag going to zero at the grounding line. However, our basal-melt parameterisation does not provide a smooth transition from grounded to floating ice. Thus our results regarding the key role of the ocean on the grounding line position can be affected by the coarse model resolution. Computational constraints do not allow for the required high model resolution, especially with a 3D finite difference model on these long time scales. However, the potential inaccuracy of the grounding line position introduced by the coarse resolution, typically of ~100 km (Vieli and Payne, 2005; Gladstone et al., 2017) is one order of magnitude smaller than the grounding line migrations simulated here (more than 1000 km). This issue should be investigated in the future both at much higher resolution, as well as including different formulations of friction and submarine melting.*

P4 L26 When using the PDD method, you are discarding the role of insolation changes. Could you add a justification on why this is negligible?

This issue was also raised by the other reviewers. The PDD scheme has limitations for paleo simulations essentially because it neglects insolation-driven melting, which can be very important for long-term simulations including variations at orbital timescales, especially in past warming periods. Nevertheless, because our focus is on abrupt climate changes of the last glacial period, our reference climate is precisely a glacial (stable) climate where insolation variations are not important. We have now discussed this point with much detail in the Model description section by adding the following:

R3 SC2k & **R2 SC2f &** R1 MC4**:** *This melting scheme is admittedly too simple for fully transient paleo simulations, as it omits the contribution of insolation-induced effects on surface melting (Robinson and Goelzer, 2014). Nevertheless, insolation changes are most relevant in long-term simulations including variations at orbital timescales, especially in past warmer periods such as the Eemian. Since this study focuses on glacial abrupt climate changes within a fixed background (glacial) climate, insolation changes are not important and the PDD melt model should be sufficient to give a good approximation of surface melt in response to interstadials in a reasonable manner.*

P5 L23 Again, it could be nice to have a plot of the stadial to interstadial temperature change in the atmosphere and in the ocean from a "typical" DO event (beta from -1.5 to 1.5).

**R2 SC2g:** We believe this information is provided in new figures 1 and 3, See also R2 MC4

P6 L7 To facilitate the reading your standard value of Kappa can appear here.

Done. We have included this as follows:

**R2 SC2h:** *kappa is the heat flux exchange coefficient between ocean water and ice at the ice-ocean interface; its standard value in the present study is kappa = 5 m a$^{-1}$ K$^{-1}$*

P6 L17 Bgl not presented before.

It was actually introduced just in the line before but to make this more clear we have now written B$_{gl}$ between parentheses, and also included a mention to the gamma factor:

**R2 SC2i:** *The basal melting rate for purely floating ice shelves (Bsh) is given by the grounding-line basal melt (Bgl) scaled by a constant factor gamma…,*

P6 L21 Why 750m? It seems relatively low as we have ice shelves today at much greater depth in Antarctica.

This value simply reflects an upper limit of the continental shelf in the area, and it is actually higher than the value of 450 m originally used by Peyaud et al (2007) for the same area. It roughly corresponds to the continental shelf depth of the Barents Sea. What allows a marine-terminating ice sheet to advance towards the continental-shelf break is a downstream grounding-line advance. This advance is  determined by local mass balance (whose terms are: accumulation, ablation  basal melting and calving) together with ice advection from the ice-sheet interior. At the continental-shelf break, the bedrock depth increases abruptly to much larger depths depth. A grounding-line advance beyond the continental-shelf break (at glacial times, either for the Antarctic or Eurasian ice sheets) would require an extremely large (unrealistic) ice flux from the ice-sheet interior.  In Antarctica there are indeed much larger ice shelves; this is partly because of the local temperature conditions but also because of the larger depth of the continental shelf. Imposing large basal melting values beyond this region is needed to limits the occurrence of unrealistically large ice shelves in open-ocean waters, which would likely be subject to basal melt rates above $B_{40K}$ = 0.1 m/a. We have modified this sentence to make this clear:

**R2 SC2j:** *As in Peyaud et al. (2007) in regions with ocean depths above 750 m, an artificially large melting rate (20 m~a$^{-1}$ is prescribed to avoid unrealistic growth of ice shelves beyond the continental-shelf break, where they would likely be subject to high melt rates in reality because of high heat exchanges with the ocean.*

P7 L11 See general comments. Justify/discuss the importance of the chosen value.

**R2 SC2k:** This line says "where $N_g$ (t) represents the evolution of the number of points of " ; it is unclear to us what the reviewer means here.

P7 L29-31 P8 L1-3 This is not clear to me why you did not use the 3D field computed from CLIMBER3-α. Since the ice sheet model provides you the depth of the ice base you can easily read the temperature simulated by your climate model at this depth.

The reviewer is totally right here. Our experimental design was done using either the surface or the subsurface for the sake of simplicity. We followed up from previous work (Álvarez Solas 2011; 2013) where we used temperature at a fixed depth, either the surface or the subsurface, to calculate the basal melting of ice shelves. We are actually planning to implement a basal melting parameterisation that takes into account the actual depth of the ice-shelf base that evolves in time. We have included a mention to this issue in the Discussion section.

**R2 SC2l:** *For the sake of simplicity, and following up from previous work (Alvarez Solas 2011; 2013) we herein calculated the basal melt by using ocean temperature at a fixed depth, either at the surface or at the subsurface. Using the three-dimensional temperature provided by the climate model at the local ice-shelf depth that can evolve in time as the ice-shelf thickness varies would have been more realistic and should be in the scope of future work.*

P8 L5 Section 5 is the conclusion.

**R2 SC2m:** Indeed. We have now corrected this to refer to section 4 (Discussion).

3. Results

P8 L25 What is the volume of your spun-up ice sheet? How small is 1.5 m sle relative to this volume?

The volume of the ice sheet was MISSING. We have included the following:

**R2 SC3a** & R2 SC3a: *In ATM, the atmospheric forcing alone causes a sequence of enhanced ablation episodes resulting in modest ice volume variations (up to 1.5 m SLE) during the most prominent stadial-interstadial transitions; this represents a change of approximately 7% with respect to the initial ice-sheet volume.*

P9 L21-25 This is unconvincing because a map of SMB changes from stadial to interstadial is missing. SMB is negative at the continental margins, from the BIIS to the BKIS. From your equations, it seems that you impose an important change in surface temperatures (please show as well annual and July temperature changes!) so it is hard to picture why melt is restricted to a narrow band in the South as you imply.

**R2 SC3b:** We have followed the suggestion and added summer temperatures. As it is now visible, these temperatures are between -1 and -5 ºC at the southern edge of the BKIS and Scandinavia, while the stadial to interstadial warming in summer is much reduced when compared to the annual anomaly and it is only slightly above 4 ºC, explaining the very limited ablation in this region. South and East of the British Islands summer temperatures are closer to 0ºC with a perturbation between 5 and 10 degrees which explains its ablative response.

We add here for further clarity a figure showing the surface mass balance changes experienced by the ice sheet during a stadial-to-interstadial transition:

[Figure]

ATM | Surface Mass Balance anomaly

We believe the manuscript is now sufficiently long and detailed with about 40 pages and 11 figures, thus we decided to not include this figure.

P9 L25-28 Does CLIMBER3-α provide oceanic temperature changes below the 40k ice sheet? How this is possible? If not, how do you compute the sub-shelf basal melting rate when the ice sheet retreats from its initial position?

No, it does not. Ocean temperatures were spatially extrapolated to allow them to be defined everywhere in case they are needed.

P9 L25-28 It might be worth noting that if basal melting is more efficient than surface mass balance this is because you have calving in the ocean. Calving is a very efficient way to remove ice (confirmed by your Fig. 5).

Done. This sentence has been included as follows:

**R2 SC3d:** *Note that basal melting together with calving is a very efficient method to remove ice; basal melting leads to thinning of the ice shelf which can subsequently undergo calving.*

P10 L8-9 The retreat pattern of Fennoscandian ice sheet is somewhat surprising. It seems that the ice sheet retreats increased basal melting in the Baltic sea (which is a lake in your setup right?)?

The referee is right. Because oceanic fields are extrapolated along the domain, the ice sheet model "sees" the mentioned lake as an ocean connected to the Atlantic so it can produce some basal melting. Nevertheless this overestimation represents a minor contribution and also the concerned area is subjected to surface ablation thus this effect is somehow cancelling out and attenuated. Note that avoiding this would require a considerable amount of technical work and we believe it will not affect the main conclusions of the paper but only slightly change the numbers of this particular region.

Conclusions

P14 L5 Alvarez-Solas et al. (2013) show that is the subsurface warming caused by AMOC slowdown is responsible for LIS H-events. When subsurface temperature is used here you end up basically with the same synchronisation for EIS and LIS. It is not really convincing to use subsurface temperature for one ice sheet and surface temperature for the other. Again, GRISLI gives you the depth of the ice shelf base so you can use the CLIMBER3-α layer corresponding to this depth, for the LIS and for the EIS. In this case, the study would have been more convincing. Consider reformulation here.

We understand the reviewer's concerns. As mentioned before, using the three-dimensional temperature provided by the climate model at the local ice-shelf depth that can evolve in time as the ice-shelf thickness varies would have been more realistic and should be in the scope of future work. We have reformulated this sentence as follows:

**R2 SC5a:** *Taken together with these studies, our results support the potential of NH ice sheets to react to glacial abrupt climate changes.*

Fig. 2 Please mention in the caption that these fields are later scaled to reproduce the NGRIP stadial to interstadial temperature change (temporally variable factor but roughly 3 times the changes simulated by CLIMBER3-α). Otherwise this figure might be misleading.

Done. This was modified as follows:

**R2 Fig2a:** *Note that to force the ice-sheet model these fields are scaled to reproduce the NGRIP interstadial minus stadial temperature change.*

Fig. 2 Around the coasts of Scandinavia you have a CLIMBER SAT anomaly of about 9 degrees which means that during certain DO events you have episodically a local temperature change of about 30 degrees (beta* from -1.5 to 1.5). I am surprised that such a temperature change do not translate in large SMB perturbations. Any comment?

**R2 Fig 2a: Please see response to R2 SC3b**

Fig. 5 What are the dashed grey lines? They do not seem to relate to the major tick marks.

**R2 Fig 5**: We have modified the dashed grey lines so they correspond to the major tick marks.

Fig. 6 The southern edge of the BKIS (Taymyr peninsula / Ob river) seems almost not changed in ATM before and after the DO event. You have a beta* change of almost 2.5 (roughly -1 to 1.5) meaning that you have a change in annual temperature of at least 4x2.5=9 degrees. The southern extension of the BKIS is limited by melt. With an additional 9 degrees in annual temperature (how many in July?), it is not obvious to me why you do not have any melt increase there.

**R2 Fig 6:** Absolute temperatures before the perturbation are well below -10ºC in this region, explaining why a warming of 5 to 10 degrees does not trigger large ablation. Please see also R2 SC3b and the surface mass balance figure we added for further clarity.

Fig. 7 Episodically the ice shelf extension is abruptly rising (e.g. 45 kaBP) not necessarily linked to any significant change in beta*, ice sheet velocity nor calving. What is the reason for that?

**R2 Fig7:** These episodes are explained by the existence of thresholds in the ice-shelf capability to expand as a result of decreasing basal melt. A gradual declining beta from positive values still allows for some melting that efficiently avoids the ice shelves to expand. When beta becomes slightly below 0 (to compensate the background 0.1 m a^(-1) of basal melting) ice shelves no longer experience any melting and are able to advance.

Supp. Mat. Fig 3 : the standard deviation in ice volume is not a good indication of the amplitude of millennial oscillations. You should correct from the background linear trend or simply compute the standard deviation of the dVdt variable. From the graph on the left, it seems that you do have oscillations of about 2 m sle for certain PDD parameter combination but maybe at a lower frequency. Could you comment on that?

Done. **R2 FigSM3:** We have now replaced the figure with its equivalent on dV/dt. See new Figure S3.

**Technical corrections**

P6 L31 boundary

**R2 Tech a:** Done

Fig. 5 Problem in the caption.

**R2 Tech b:** Corrected**.**

**Anonymous Referee #3**

Alvarez-Solas et al. present a modelling study that investigates the impacts of millennial-scale climatic and oceanic forcings on the Eurasian ice sheet during the last glacial period, and in particular during MIS 3. A 3D thermo-mechanical ice-sheet model is used, with an offline forcing that provides a more robust representation of millennial scale climate variabilities compared to traditional methods. Explicit treatment of submarine melting within the ice model allows the authors to consider the relative

contributions of ice-surface melting (ablation) vs dynamic process related to ice-ocean interactions (ocean surface and subsurface melting).

Results show oceanic forcing plays a dominant role over surface melting in controlling dynamic losses of the EIS over sub-millennial timescales, as well as its importance spatially. Of particular interest is the predicted role that subsurface ocean temperatures can play in enhancing ice discharge during stadial conditions in the Barents Sea/high latitudes, thus supporting empirical observations for the presence of Eurasian IRD in the North Atlantic during stadials. The approach of the manuscript, alongside sensitivity experiments, appears robust and the results provide an important contribution to further understanding ice-dynamical processes occurring in this understudied domain. I suggest minor revisions to the manuscript based on comments and questions below:

We are very grateful to the reviewer for their careful reading of the manuscript and the many suggestions made which undoubtedly have contributed to make the results more clear. We have attempted to address all concerns. Below we give a detailed response to each of the comments raised.

P2L22: In terms of underlying mechanisms contributing to collapse of the BSIS, there are additional papers to cite beyond Gudlaugsson. e.g., ice stream surging (Andreassen et al., 2014); subglacial meltwater (Esteves et al., 2017); subsurface melting/ocean warming (Ivanovic et al., 2018; Rasmussen and Thomsen, 2004).

We have now included the former references as follows:

**R3 SC1a:** Mechanisms contributing to the deglaciation of the BKIS include ice stream surging (Andreassen et al., 2014); subglacial meltwater (Esteves et al., 2017) and subsurface melting through ocean warming (Ivanovic et al., 2018; Rasmussen and Thomsen, 2004). An improved understanding of these would provide important insights into the future evolution of the WAIS (Gudlaugsson et al. 2013; 2017).

P2L19: The acronym LGM is not defined

**R3 SC1b**: This is right, we had defined in below (in page 5); we have now corrected this in both places.

P2L25: It would be useful for readers not familiar with marine isotope stages to also state the timeframe in years BP

Done, but in line 15, which reads now:

**R3 SC1c:** [...] MIS 3, ca. 60-25 ka BP [...]

P2L32: I think this understates the uncertainty in marine sectors – minimum extents for ice in the Barents/Kara seas during MIS 3 are essentially unknown: Hughes et al. (2016) do not try to speculate on limits here prior to 32 ka BP. It would be appropriate to discuss the glacial history of the Eurasian Arctic during MIS3 within the context of long-term IRD records (e.g., Kleiber et al., 2001; Knies et al., 2001; Mangerud et al., 1998).

This has been modified as follows:

**R3 SC1d: These data confirm substantial variations of the EIS extent, with the largest uncertainties in marine sectors of the ice sheets**; as a consequence trying to estimate its limits prior to 32 ka BP was not attempted by Hughes et al. (2016).

It would be appropriate to discuss the glacial history of the Eurasian Arctic during MIS3 within the context of long-term IRD records (e.g., Kleiber et al., 2001; Knies et al., 2001; Mangerud et al., 1998).:

P3L10: Also should mention Petrini et al. (2018), which does have explicit treatment of ocean forcing for modelling retreat of the BSIS. Also possibly (Ivanovic et al., 2018) in terms of HS1.

**R3 SC1e:** The work of Petrini et al, focuses on the BSIS retreat during the deglaciation and not before. We have nevertheless cited it now in the introduction.

P3L17: And Patton et al. (2017).

Done; this was also suggested by reviewer 1. It now reads as follows:

**R3 SC1f:** *This study was subsequently extended throughout the last deglaciation until 8 ka BP (Patton et al., 2017).*

P3L30: Missing section 4.

This has been corrected and now reads:

**R3 SC1g:** *In section 4 the implications of our study for glacial and future climate changes are discussed.*

2. Model description

P4L12: What is the basis for these thresholds for SSA activation?

This issue was also raised by reviewer 1. The presence of water at the base of the ice sheet implies that it is not frozen to the bedrock, i.e., sliding is physically possible. More water at the base facilitates sliding more by reducing the effective pressure, and sediments also facilitate sliding because they are deformable. The criteria of 1m water thickness over sediments reduces noise in the SSA activation. The 400 m criterion over hard bedrock is a more tuneable parameter, but is also effectively is used to reduce noise. Setting these criteria greater than zero ensures that ice streams are activated in regions that are robustly temperate. We think that changing these values over a reasonable range (including setting them both to zero) would not change the conclusions of the work. We have included this explanation in the model description now.

**R3 SC2g:** *Setting these thresholds ensures that ice streams are activated in regions that are robustly temperate. The presence of water at the base of the ice sheet implies that it is not frozen to the bedrock, i.e., sliding is physically possible. More water at the base facilitates sliding more by reducing the effective pressure, and sediments also facilitate sliding because they are deformable. The criteria of 1 m water thickness over sediments reduces noise in the SSA activation. The 400 m criterion over hard bedrock is a more tunable parameter, also allowing for a more numerically robust calculation of velocities within the SSA.*

P4L14: criterium->criterion

**R3 SC2h:** Done.

P4L15: citation needed for the typical observed Hcalv.

Done. This issue was also raised by reviewer 1. This now reads:

**R3 SC2i:** *This is a semiempirical parameter reflecting the fact that this is the typical thickness of ice-shelf fronts currently observed in Antarctica (Griggs and Bamber, 2011).*

P4L21: "Initial" topographic conditions infers GIA is accounted for, but is not described in the model description.

This issue was also raised by reviewer 1 (R1 SC2m). We have now included a description of the GIA as follows:

**R3 SC2j:** *The glacial isostatic adjustment (GIA) is described by the elastic lithosphere – relaxed asthenosphere method (Le Meur and Huybrechts, 1996), for which the viscous asthenosphere responds to the ice load with a characteristic relaxation time for the lithosphere of 3000 years. For the sake of simplicity, the isostatic adjustment is assumed here to be only due to local ice mass variations, as other works have done in the past (Greve and Blatter, 2009; Helsen et al., 2013; Huybrechts, 2002; Langebroek and Nisancioglu, 2016; Stone et al., 2013).*

P4L26: It does not appear that any account has been taken for the contribution of insolation-induced melt during MIS 3 (e.g., Robinson and Goelzer, 2014).

This issue was also raised by the other reviewers. Indeed, we have here resorted to the simplest approach possible to calculate ablation and thus the SMB, PDD. The PDD scheme indeed neglects insolation-driven melting, which can be very important for long-term simulations including variations at orbital timescales, especially in past warming periods. Nevertheless, because our focus is on abrupt climate changes of the last glacial period, our reference climate is precisely a glacial (stable) climate where insolation variations are not important. We have now discussed this point with much detail in the Model description section by adding the following:

R3 SC2k & **R2 SC2f** & R1 MC4**:** *This melting scheme is admittedly too simple for paleo simulations, as it omits the contribution of insolation-induced effects on surface melting  (Robinson and Goelzer, 2014). Nevertheless, insolation changes are most relevant in long-term simulations including variations at orbital timescales, especially in past warmer periods such as the Eemian. Since study focuses on glacial abrupt climate changes within a background fix (glacial) climate, insolation changes are not important and the PDD melt model should be sufficient to give a first approximation of surface melt in response to interstadials in a reasonable manner.*

P5L4: SMB not defined

**R3 SC2l:** This issue was also pointed out by reviewer 1 (R1 SC2j). We have now corrected this.

P5L23: AMOC not defined

**R3 SC2m:** Indeed, this has been corrected now.

P6L8,9: Some citations here would be useful.

Done, this now reads as follows:

**R3 SC2n:** *Several marine-shelf basal melting parameterisations can be found in the literature, as recently reviewed by Asay-Davis et al. (2017).*

P6L18: The submarine melt rate for ice shelves appears somewhat arbitrary and does not appear to consider possible refreezing associated with supercooling (e.g., Jenkins and Doake, 1991). While Bgl is undoubtedly more important in terms of the glacial response, will modifying this coefficient of 0.1 likely introduce any major differences on the results?

This issue was also raised by reviewer 1 (R1 SC2r). The value of 0.1 for the gamma coefficient was set based on present-day observations. *Rignot and Jacobs (2002) indicated that melt rates near glacier grounding lines exceed the area average rates for the largest ice shelves by 1 to 2 orders of magnitude. Marsh et al. (2015) found a difference in one order of magnitude for the Ross Ice Shelf. Münchow et al., 2014 found values between 1-15 m/a in the Petermann glacier. Wilson et al., 2017 found basal melting values ranging from roughly 0 to 30-50 m/a in the 79 North Glacier,, Ryder Glacier, and Petermann Glacier.* Results could show some sensitivity to this value, but we think this would not alter our conclusions. Although we did not explore any other values different from gamma = 0.1, we did consider a range of kappa values between 1-10 m/a/K, which accounts for a wide range of sensitivities. We have now rephrased the former paragraph as follows:

R1SC2r & **R3 SC2o:** *Thus, we consider that the submarine melting rate for ice shelves is 10 times lower than that close to the grounding zone, which is in qualitative agreement with observations in some Greenland glaciers with floating tongues (Münchow et al., 2014; Wilson et al., 2017) as well as in Antarctic ice shelves (Rignot and Jacobs, 2002; Marsh et al., 2016). Note that this value is subject to uncertainty. Although we did not explore any other values different from gamma = 0.1, we did consider a range of kappa values between 1-10 m/a/K which accounts for a wide range of oceanic sensitivities (see section 2.3).*

P7L15: This statement on the agreement with previous reconstructions appears confusing – neither study cited shows reconstructed margins during MIS 3 at 40 ka BP. Are the authors instead referring to the glacial maximum of the Mid Weichselian (MIS 4/3) at ~60 ka?

Indeed, these we have reformulated this to be more precise as follows:

**R3 SC2p:** *This procedure was found to facilitate the growth of European ice-sheets within the reconstructed limits for 60~ka BP and 20~ka BP (Svendsen et al. 2004; Kleman et al 2013).*

P8L5: wrong section cited.

**R3 SC2q:** This issue was also pointed out by reviewer 2 (**SC2k**). We have corrected it to refer to section 4.

P8L25: It would be useful to see this value in relation to total ice volume of the ice sheet.

This issue was also pointed out by reviewer 2 (**SC3a**). We have included this as follows:

**R3 SC3a** & R2 SC3a: *In ATM, the atmospheric forcing alone causes a sequence of enhanced ablation episodes resulting in modest ice volume variations (up to 1.5 m SLE) during the most*

*prominent stadial-interstadial transitions; this represents a change of about 7.2% of the initial ice-sheet volume.*

P9L31: 'British-Irish'

**R3 SC3b**: Done.

P10L23: 'of all the other

**R3 SC3c:** Done**.**

P12L3-5: The reason for linking these two statements is not clear unless it's mentioned also the susceptibility of the WAIS to oceanic warming.

**R3 SC4a:** This is right. We have moved the second sentence to the paragraph just below, where the evidence for the timing of IRD is discussed.

P12L32: This is a useful section that discusses the major limitations of the present study and where future work is needed. The authors however do not mention the limitation of the grid resolution at the grounding line within the context of insights into the EIS responses across sub-millenial timescales. The use of an index to track grounding line dynamics is interesting and a very useful tool, although some mention of the simplifications on grounding line migration would be appropriate to mention given the main conclusions e.g., response time to abrupt forcing.

This is an important point raised by all three reviewers. We realise that with the coarse resolution it is not possible to properly resolve the grounding-line dynamics. However at this stage it is not feasible to use the high resolutions required, and alternatives Thus our results should be considered as a first step toward investigating this problem and revisited in the future with higher-resolution models when this is feasible. We have attempted to acknowledge this in the Discussion section:

R1 MC1 & R1 SC5a & R2 SC2e & **R3 SC4b:** *Given the conclusion that the ocean plays a major role during abrupt ice sheet changes, the model treatment of grounding line dynamics is a key issue. Several studies have shown that for many applications, a resolution of around 1 km is often needed to accurately represent the grounding-line dynamics. In addition, it has been shown (e.g. Gladstone et al 2017) that the grounding-line behaviour is sensitive to the choice of friction law and the physics of submarine melting, and that these determine model-resolution requirements. An abrupt change across the grounding line in either basal drag or basal melting will require a high resolution. In our case the dependency of basal drag on effective pressure allows for the desirable property of basal drag going to zero at the grounding line. However, our basal-melt parameterisation does not provide a smooth transition from grounded to floating ice. Thus we cannot rule out that our results regarding the key role of the ocean and grounding line dynamics could be affected by the coarse model resolution. However, given the millennial-scale focus and large spatial scales involved in this study, computational constraints do not allow for the required high model resolution, especially with a 3D finite difference model. This issue should be investigated in the future both with a much higher resolution as well as including different formulations of friction and submarine melting.*

P12L15: Along the southwest EIS (Irish/Scottish margin) at least. This effect of increased IRD during stadials is not observed by Becker (2017) further north along the mid Norwegian margin during MIS3.

**R3 SC4c**: Done (we have added this comment).

P12L17: Citation P13L18: citation needed.

**R3 SC4d:** Done. Several references have been added: Broecker et al. (1985), Ganopolski and Rahmstorf (2001); Manabe and Stouffer (1995); Menviel et al. (2014); Rasmussen et al. (2013); Schmittner et al. (2003)

P13L34: Should mention here in the conclusions the anti-phase effects of the subsurface warming.

The following was added in the Conclusions section:

**R3 SC5a:** *Oceanic forcing was considered both at the surface and at the subsurface. The timing of the response with respect to changes registered in Greenland depends on whether the surface or the subsurface of the ocean is considered as the relevant forcing of the ice sheet. A quasi-anti-phase relationship is found in these two cases. This behavior can be explained by the fact that ocean waters at the subsurface warm (cool) during episodes of reduced (enhanced) convection at the Nordic Seas as a result of variations in the AMOC strength.*

Figures: Bjørnøyrenna is misspelled among figure captions.

This has been corrected now (see also comment by reviewer 1).

Missing figure lettering on Fig 2 & 9.

**R3 Fig 2, 9:** Done (see also comment by reviewer 1).

Figure 6: It appears from the OCNsrf timeslices that the Baltic region of the FIS is dramatically affected by ocean surface temperature forcing even though this area was disconnected from the North Atlantic. Is there any provision in the model to distinguish freshwater vs. ocean?

**R3 Fig 6:** No, we have to acknowledge that there is no distinction between the two in the model. Please see also the response to R2 SC3e

Figure 9: Mean/max ice thickness?

**R3 Fig 9**: Mean; we have included this in the figure caption now.

References:

Andreassen, K., Winsborrow, M.C.M., Bjarnadóttir, L.R., Rüther, D.C., 2014. Ice stream retreat dynamics inferred from an assemblage of landforms in the northern Barents Sea. Quat. Sci. Rev. 92, 246–257. doi:10.1016/j.quascirev.2013.09.015

Becker, L.W.M., Sejrup, H.P., Hjelstuen, B.O., Haflidason, H., Dokken, T.M., 2017. Ocean-ice sheet interaction along the SE Nordic Seas margin from 35 to 15ka BP. Mar. Geol. doi:10.1016/j.margeo.2017.09.003

Esteves, M., Bjarnadóttir, L.R., Winsborrow, M.C.M., Shackleton, C.S., Andreassen, K., 2017. Retreat patterns and dynamics of the Sentralbankrenna glacial system, central Barents Sea. Quat. Sci. Rev. 169, 131–147. doi:10.1016/j.quascirev.2017.06.004

Hughes, A.L.C., Gyllencreutz, R., Lohne, Ø.S., Mangerud, J., Svendsen, J.I., 2016. The last Eurasian ice sheets – a chronological database and time-slice reconstruction, DATED-1. Boreas 45, 1–45. doi:10.1111/bor.12142

Ivanovic, R.F., Gregoire, L.J., Burke, A., Wickert, A.D., Valdes, P.J., Ng, H.C., Robin- son, L.F., McManus, J.F., Mitrovica, J.X., Lee, L., Dentith, J.E., 2018. Acceleration of Northern Ice Sheet Melt Induces AMOC Slowdown and Northern Cooling in Simulations of the Early Last Deglaciation. Paleoceanogr. Paleoclimatology 33, 807–824. doi:10.1029/2017PA003308

Jenkins, A., Doake, C.S.M., 1991. Ice-ocean interaction on Ronne Ice Shelf, Antarc- tica. J. Geophys. Res. Ocean. 96, 791–813. doi:10.1029/90JC01952

Kleiber, H.P., Niessen, F., Weiel, D., 2001. The Late Quaternary evolution of the west-ernLaptevSeacontinentalmargin,ArcticSiberia Tˇimplicationsfromsub-bottompro- filing. Glob. Planet. Change 31, 105–124. doi:10.1016/S0921-8181(01)00115-1

Knies, J., Kleiber, H.-P., Matthiessen, J., Müller, C., Nowaczyk, N., 2001. Marine ice-rafted debris records constrain maximum extent of Saalian and Weichselian ice- sheets along the northern Eurasian margin. Glob. Planet. Change 31, 45–64. doi:10.1016/S0921-8181(01)00112-6

Mangerud, J., Dokken, T., Hebbeln, D., Heggen, B., Ingólfsson, Ó., Landvik, J.Y., Mejdahl, V., Svendsen, J.I., Vorren, T.O., 1998. Fluctuations of the Svalbard-Barents Sea Ice Sheet during the last 150 000 years. Quat. Sci. Rev. 17, 11–42. doi:10.1016/S0277-3791(97)00069-3

Patton, H., Hubbard, A., Andreassen, K., Auriac, A., Whitehouse, P., Stroeven, A.P., Shackleton, C., Winsborrow, M.C.M., Heyman, J., Hall, A.M., 2017. Deglaciation of the Eurasian ice sheet complex. Quat. Sci. Rev. 169, 148–172. doi:10.1016/j.quascirev.2017.05.019

Petrini, M., Colleoni, F., Kirchner, N., Hughes, A.L.C., Camerlenghi, A., Rebesco, M., Lucchi, R.G., Forte, E., Colucci, R.R., Noormets, R., 2018. Interplay of grounding-line dynamics and sub-shelf melting during retreat of the Bjørnøyrenna Ice Stream. Sci. Rep. 8, 7196. doi:10.1038/s41598-018-25664-6

Rasmussen, T.L., Thomsen, E., 2004. The role of the North Atlantic Drift in the millennial timescale glacial climate fluctuations. Palaeogeogr. Palaeoclimatol. Palaeoecol. 210, 101–116. doi:10.1016/J.PALAEO.2004.04.005

Robinson, A., Goelzer, H., 2014. The importance of insolation changes for paleo ice sheet modeling. Cryosph. 8, 1419–1428. doi:10.5194/tc-8-1419-2014

**Figures:**

[Figure]

**New Figure 1 |** (old Figure 1 with labeling and Summer Temperature)

[Figure]

**New Figure 2 |** (former panel c on old Figure 1, with new indications for the different regions of the EIS decomposed in new Figure 8)

[Figure]

**New Figure 3 |** (old figure 2 with a new panel for summer temperature anomalies and labeling)

[Figure]

**New Figure 4 |** (former Figure 3)

[Figure]

**New Figure 5 |** (former Figure 4)

[Figure]

**New Figure 6 |** (former Figure 5)

[Figure]

**New Figure 7 |** (former Figure 6 with labeling)

[Figure]

**R1 SC3f: New Figure 8 |** Regional decomposition of volume changes

[Figure]

**New Figure 9 |** (former Figure 7 with labeling and text corrections)

[Figure]

**New Figure 10 |** (former Figure 8 with labeling and text corrections)

[Figure]

**New Figure 11 |** (former Figure 9 with labeling)

[Figure]

**Figure S1 |** (no changes)

[Figure]

**Figure S2 |** (no changes)

[Figure]

**Figure S3 |** (We have replaced the former ice volume time series by their derivatives, thus the standard deviation is now calculated on dV/dt)

[Figure]

**Figure S4 |** (New figure showing the dependency of the volume evolution on Hcalv)

**Tables:**

| Variable \| Parameter | Identifier name | Standard value | Explored range | Units |
|---|---|---|---|---|
| Basal friction coefficient on sediments | $c_f$ | $2\times 10^{-5}$ | – | a m$^{-1}$ |
| Basal friction coefficient on bedrock | $c_f$ | $20\times 10^{-5}$ | – | a m$^{-1}$ |
| Standard deviation of daily temperature | $\sigma$ | 5 | [4 - 6] | K |
| Snow conversion factor from PDDs to melt | $f_{PDD_{snow}}$ | 0.003 | [0.0015 - 0.006] | mwe PDD$^{-1}$ |
| Ice conversion factor from PDDs to melt | $f_{PDD_{snow}}$ | 0.008 | [0.004 - 0.016] | mwe PDD$^{-1}$ |
| Standard deviation of daily temperature | $\sigma$ | 5 | [4 - 6] | K |
| Ice thickness threshold for calving | $H_{calv}$ | 150 | [10 - 500] | m |
| Oceanic sensitivty for ice-shelf melting | $\kappa$ | 5 | [0 - 100] | m a$^{-1}K^{-1}$ |

**Table 1.** Model parameters used in this study with their standard and explored values

| | Millennial-scale forcing component | | | |
|---|---|---|---|---|
| Experiment name | Atmosphere | Surface ocean | Subsurface ocean | Sea level |
| $CTRL$ | · | · | · | · |
| $SL$ | · | · | · | ✓ |
| $ATM$ | ✓ | · | · | · |
| $OCN_{srf}$ | · | ✓ | · | · |
| $OCN_{sub}$ | · | · | ✓ | · |
| $ALL_{srf}$ | ✓ | ✓ | · | ✓ |
| $ALL_{sub}$ | ✓ | · | ✓ | ✓ |

**Table 2.** Millennial-scale components used to force the ice-sheet model in the different experiments shown in this study.

---

## Author Response (AR2)

I thank the authors for their thorough answers. They have notably included a few important points in the discussion. I still have some minor suggestions to make the paper more convincing.

- 1) The simulated ice sheets in the course of the simulations show a negative trend in ice volume, leading to a ~40-50% ice loss from their initial value. I previously pointed out that such transient response may raise doubt in the experiment setup: is the imposed forcing too strong to maintain an ice sheet?
The authors put forward three points that can explain this trend: i- ice volume hysteresis; ii-orbital variability and iii- refreezing / sub-shelf ice accretion.
The two first points are indeed expected. A simple way to close the debate, and thus to reinforce your conclusions, would be to present the results of your ice reconstructions when the orbital forcing is also included, following the setup of a previous paper in your group (Banderas et al., 2018). In doing so and showing that your setup can broadly reproduce the expected Eurasian ice sheet volume change (e.g. a maximum ice volume between 26 to 21 k), readers will be definitively convinced of the validity of your setup, strengthening your conclusions.

We have followed the referee's suggestion by performing two additional simulations and made a new accompanying figure. Figure S6 shows the ice volume evolution when the orbital forcing is also included. This figure clearly shows that we can reproduce "the expected Eurasian ice sheet volume change (e.g. a maximum ice volume between 26 to 21 k)." Therefore, as the referee points out, it proves the validity of our setup.

Please see new Figure S6. A reference to this new figure has been added in the Discussion section of the manuscript.

Finally, refreezing does occur below the grounded and floating ice. Refreezing is by definition melt water that turns back to ice under certain conditions. I agree that locally this refreezing can lead to ice accretion. On a larger scale however, refreezing leads, at best, to a net zero mass balance. In order to have model grid-size scale (40km) ice accretion, it means that ocean waters turn into ice. This might happen but I doubt that a substantial ice-sheet scale ice growth (i.e. a few metres of SLE per millenia) can be due to this phenomenon.

We appreciate this final nuancing of the role of ice accretion by the referee. Indeed, we already removed refreezing under the ice shelves in the previous version of the manuscript. Thus, the current experimental set up totally satisfies the referee's view on refreezing.

- 2) I appreciate the discussion on the limitations of the model on the grounding line migration, although you only discuss the role of resolution. You should also mention that there are ways to cope with coarse resolution models such as flux adjustment derived from analytical formulations (e.g. Schoof, 2007) or subgrid grounding line computations and basal friction scaling (e.g. Winkelmann et al., 2011).

We agree with the referee and have included a sentence in the Discussion section, where we discuss these other methods of treating the grounding line. It reads: "Sub-grid parameterizations (e.g. Feldmann et al., 2014; Winkelmann et al., 2011) or flux adjustment derived from analytic formulations (e.g. Schoof, 2007) have been proposed as methods to treat the grounding line in coarse resolution models"
We have also included the references to the mentioned studies.

- 3) I previously asked for a 2D map of the imposed forcing changes, i.e. SMB and sub-shelf basal melting rate, around an idealised DO event (for example a change from beta=-1. to beta=1. On top of your initial topography). This allows the reader to have a grasp of what the ice sheet is actually experiencing. In your answer you point me to Fig. 1 and 3 which show the absolute temperature and precipitation and their changes. However: i- Fig. 1 and 3 are the direct outputs from CLIMBER3 (not scaled to NGRIP) and thus it is not what you actually use as a forcing; and ii- it is not straightforward to convert a summer temperature to a SMB.
I agree that the paper is already long but this kind of figure seems to me even more important than Fig. 3 which shows climatic fields that are not directly used by the ice sheet model.

As a side note: I do not understand what is depicted in the figure of your answer R2 SC3b. I assume it is interstadial minus stadial since you have a more negative SMB at the margins. Also, there is no indication on which beta change this figure is based on, nor if it is computed after the atmospheric scaling. In any case, this kind of figure along with the sub-shelf melting anomaly (or potential melt if there is no ice shelf) would be an important addition to the paper.

We have now included a new figure (Figure S5) which shows the SMB and sub-shelf basal melting (basal mass balance; BMB) anomaly fields as requested by the referee. This new figure is built by choosing the periods around DO12 as we did for figure 7. The stadial period previous to DO12 has a value of beta of ca. - 0.5 and the posterior interstadial a value of ca. 1.5. We have referenced the new figure in the Results section.

As previously stated in the paper: *"The rate of ice loss by basal melting is similar to that resulting from the increase in ablation (as reflected in the SMB) during the peak of a stadial-to-interstadial period. However, basal melting is much more efficient than surface mass balance in decreasing volume along the whole duration of an interstadial. This is due to the fact that ablation is restricted to the southern borders of the EIS. Thus, when the ice sheet has retreated to areas of no ablation, in spite of a slight further loss provided by the elevation feedback it rapidly equilibrates and a negative surface mass balance cannot propagate further inland. In contrast, when enhanced basal melting from higher oceanic temperatures is applied, the associated retreat can propagate further inland occupying a large proportion of the Bjørnøyrenna basin and facilitating high rates of volume loss (although similar in amplitude with respect to SMB) during the whole interstadial period (see the animation corresponding to ALLsrf and Figure S5 in the Supplementary Information)".*

This description is now furtherly supported by the inclusion of Figure S5.

[Figure]

Figure S5 | Surface (left) and basal (right) mass balance anomalies during DO12 (i.e. SMB and BMB fields at 47.5 ka BP minus those fields evaluated at 45.6 ka BP).

[Figure]

Figure S6 | Eurasian ice volume evolution for the last glacial period when including the orbital forcing alone (grey) and the orbital and millennial components together (black). The inset shows the evolution during MIS3.